# Using GNSS-based vegetation optical depth, tree sway motion, and eddy-covariance to examine evaporation of canopy-intercepted rainfall in a subalpine forest

Sean P. Burns<sup>1,2</sup>, Vincent Humphrey<sup>3,4</sup>, Ethan D. Gutmann<sup>2</sup>, Mark S. Raleigh<sup>5</sup>, David R. Bowling<sup>6</sup>, and Peter D. Blanken<sup>1</sup>

Correspondence: Sean P. Burns (sean@ucar.edu)

### Abstract.

Recent advances in the measurement of water content within a forest, have led to new possibilities to study canopy evaporation. We used a pair of Global Navigation Satellite System (GNSS) receivers (one above the canopy and one near the forest floor) to calculate the vegetation optical depth (VOD) during the warm season in a Colorado subalpine forest. One goal in our study was to compare VOD to the concurrent tree sway motion and subcanopy/above-canopy eddy-covariance evapotranspiration ET measurements. We found that VOD increased and tree sway frequency decreased during wet periods; furthermore, both measurements exhibited a linear relationship between each other and suggested that it took around 14 h after rainfall ceased for the intercepted rainwater to fully evaporate from the canopy. On dry days, we found that tree sway was more sensitive to diel changes in internal tree-water content than VOD. The ET measurements provided quantitative estimates of canopy evaporation (0.02 mm h<sup>-1</sup> at night, to 0.08 mm h<sup>-1</sup> during mid-day). Following rainfall, nighttime VOD, tree sway and ET all showed a steady (nearly constant) drying of the canopy. Variability in the VOD and tree sway measurements, comparisons with water content from the CLM4.5 land-surface model, and challenges with ET measurements, are also discussed.

# 1 Introduction

As warm-season precipitation falls into a forest, it first alights upon the leaves/needles and branches of the canopy, a process called interception. Depending on the forest type, tree structure, rainfall intensity, and wind gustiness, interception can primarily occur in the upper or lower canopy. As rain continues to fall, a thin film of water covers the leaves/needles until they reach a saturation point and begin to drip (or gusts of wind can shake the rainwater off the needles). From there, the inter-

<sup>&</sup>lt;sup>1</sup>Department of Geography, University of Colorado, Boulder, CO, USA

<sup>&</sup>lt;sup>2</sup>NSF National Center for Atmospheric Research, Boulder, CO, USA

<sup>&</sup>lt;sup>3</sup>Federal Office of Meteorology and Climatology MeteoSwiss, Zürich Airport, Switzerland

<sup>&</sup>lt;sup>4</sup>Department of Geography, University of Zürich, Zürich, Switzerland

<sup>&</sup>lt;sup>5</sup>College of Earth, Ocean, and Atmospheric Sciences, Oregon State University, Corvallis, OR, USA

<sup>&</sup>lt;sup>6</sup>School of Biological Sciences, University of Utah, UT, USA

cepted rainwater can drip onto lower branches (eventually reaching the ground, i.e., throughfall), travel down the trunk or stem (stemflow), or be evaporated (canopy evaporation). For a detailed description of canopy interception and related processes see Horton (1919); McNaughton and Jarvis (1983); Calder (1990); Klaassen et al. (1998); Crockford and Richardson (2000); van Dijk et al. (2015); Sheil (2018); Van Stan, II et al. (2020), and references therein. Within this paper we use the term *canopy evaporation* to refer to the evaporation of canopy-intercepted precipitation back to the atmosphere. The throughfall that reaches the ground can also evaporate back to the atmosphere (soil evaporation) or infiltrate deeper into the soil where it becomes part of the subsurface groundwater. Once in the soil, the water can be absorbed by tree roots and eventually transpired back to the atmosphere via leaf stomata (transpiration). Evapotranspiration ET is the sum of transpiration, net soil evaporation/condensation, and net canopy evaporation/condensation (e.g., Stoy et al., 2019; Coenders-Gerrits et al., 2020; Miralles et al., 2020). For our purposes, we neglect condensation and assume that net evaporation is dominated by evaporation (i.e., the transport of water vapor from the soil/canopy surfaces to the atmosphere). Discussion about condensation (i.e., negative ET) over land surfaces can be found elsewhere (e.g., El-Madany et al., 2016; Allen et al., 2020; Paulus et al., 2024).

Of the above-mentioned processes, canopy evaporation is difficult to measure and poorly understood. In general, anywhere from 10-50% of intercepted rainwater can be evaporated back to the atmosphere (e.g., Klaassen et al., 1998). However, for light rainfall over a dense forest, nearly 100% of the intercepted rainfall will evaporate. And it has been suggested that canopy evaporation rates can be as high as 39.4 mm day<sup>-1</sup> (an average of 1.64 mm h<sup>-1</sup>) during extreme rain events (Page et al., 2020). Vertical and horizontal differences in the spatial structure of a forest as well as variations in rainfall intensity, subcanopy turbulence, and humidity are part of the challenges of measuring canopy evaporation. Our study combines several disparate measurement techniques (vegetation optical depth, tree sway motion, and eddy covariance) to try and improve our understanding of canopy evaporation in a high-elevation subalpine forest.

30

Forest ET can be considered from three different perspectives: (i) at the leaf-level scale which considers transport of water vapor through the leaf boundary layer (e.g., Jarvis, 1985), (ii) from energy-balance considerations (e.g., McNaughton and Black, 1973), or (iii) by a water-balance approach (e.g., Gutmann, 2020; Brutsaert, 2023). For (ii), models of canopy ET typically use the surface energy budget to derive the Penman-Monteith equation which requires determining the aerodynamic and canopy (controlled by plant stomata) resistances (e.g., Monteith, 1965; Rutter et al., 1971; Shuttleworth, 1991; Martin et al., 1999; Gash et al., 1999; Muzylo et al., 2009; Moene and van Dam, 2014). When the canopy is completely wet, transpiration becomes small and ET is dominated by evaporation of canopy-intercepted water. In this situation, the canopy resistance (closed stomata) in the Penman-Monteith equation can be set to zero (Jarvis and Stewart, 1979; Klaassen, 2001; Bonan, 2019) and canopy evaporation  $\lambda E_i$  with units of [W m<sup>-2</sup>] becomes

$$\lambda E_i = \frac{s(R_{\text{net}} - G - S_t) + \left(\frac{\rho c_p}{r_a}\right) (q_{\text{sat}}(T_a) - q_a)}{s + c_p/\lambda},\tag{1}$$

where  $R_{\rm net}$  is net radiation [W m<sup>-2</sup>], G is soil heat heat flux [W m<sup>-2</sup>],  $S_t$  is the heat and water storage terms in the biomass and airspace between the ground and flux measurement height as well as the energy consumed by photosynthesis [W m<sup>-2</sup>],  $\rho$ 

is air density [kg m<sup>-3</sup>],  $c_p$  is the specific heat of moist air at constant pressure [J kg<sup>-1</sup> K<sup>-1</sup>],  $r_a$  is the aerodynamic resistance to scalar transport [s m<sup>-1</sup>],  $q_a$  is specific humidity [kg kg<sup>-1</sup>],  $q_{\rm sat}$  is the saturated specific humidity at air temperature  $T_a$  [K], s is the slope of the saturated specific humidity versus temperature curve [K<sup>-1</sup>] evaluated at  $T_a$ , and  $\lambda$  is the latent heat of vaporization of water [J kg<sup>-1</sup>]. In Eq. 1, vapor pressure e is often used rather than q, and aerodynamic conductance  $g_a$  is often used instead of  $r_a$  ( $g_a$  is the reciprocal of the aerodynamic resistance,  $g_a = r_a^{-1}$ ). Eq. 1 was originally formulated by Penman (1948) to estimate evaporation from an open water surface. When vegetation is partially wet, the wet and dry canopy regions can be considered independently of each other (e.g., Bosveld and Bouten, 2003; Bonan, 2019).

The atmospheric variables that drive canopy evaporation are thought to be some combination of available energy (i.e.,  $R_{\rm net}-G-S_t$  in Eq. 1), the dryness of the atmosphere (i.e., vapor pressure deficit or the  $(q_{\rm sat}-q_a)$  term in Eq. 1), advection of warmer/drier air that provides sensible heat (e.g., McNaughton and Jarvis, 1983; Calder, 1998; Schellekens et al., 1999; van Dijk et al., 2015) and the turbulent mixing of drier air into the canopy airspace (e.g., Pearce et al., 1980; Massman, 1983). The amount of turbulent mixing is determined by  $r_a$  and depends on the interaction between the wind profile and surface roughness (which depends on canopy structure). In general, it has been found that canopy evaporation models underpredict the measured canopy evaporation rates (e.g., Klaassen et al., 1998; van Dijk et al., 2015; Van Stan, II et al., 2016; Iida et al., 2017).

Vegetation optical depth (VOD) is a direct measure of the opacity of the vegetation to microwave radiation, with higher VOD values indicating denser and water-rich canopies (Frappart et al., 2020; Konings et al., 2021). It has been measured over large spatial regions (order of 150 km²) by satellites for decades (e.g., Owe et al., 2001; Liu et al., 2011). Initial work attributed changes in VOD to changes in internal tree water content and above-ground biomass (e.g., Rodriguez-Alvarez et al., 2012; Momen et al., 2017; Frappart et al., 2020). However, more recent studies have avoided using VOD during or just-following rainfall (e.g., Holtzman et al., 2021; Yao et al., 2024). This is because intercepted water forms a water layer which attenuates and reflects microwave signals more than internal water found within the xylem tubular structures; for conifers, tree water is primarily found within xylem tracheids that are on the order of 5 to 80 μm in diameter and less than 5 mm long (e.g., Tyree and Ewers, 1991; Hacke et al., 2015). It has also been shown that VOD is sensitive to dew and fog-related deposits of water on vegetation surfaces (e.g., Gerlein-Safdi, 2021). Our subalpine forest site is relatively dry, and dew formation is infrequent, though dew formation is highly variable and depends upon specific microclimates and location (Brewer and Smith, 1997).

While attributing changes in VOD to changes in biomass versus vegetation water content remains challenging, owing to a lack of consensus on retrieval algorithms and possible GNSS depolarization effects due to the canopy and topography (e.g., Camps et al., 2020; Dente et al., 2024), there has been growing interest in using these globally available observations to monitor above ground biomass and vegetation response to drought (Frappart et al., 2020). The extent to which these satellite observations are sensitive to canopy interception remains largely unknown. Such knowledge would have important implications for the interpretation of diurnal signals in satellite-based VOD (Xu et al., 2021). Recently, Global Navigation Satellite System (GNSS) receivers have been used to measure VOD at the tree- or forest-level scale (i.e., 10-100 m<sup>2</sup>) over time periods of an hour or less; with such fine-scale spatial and temporal measurements, it has been suggested that VOD can provide an estimate of precipitation intercepted by a forest canopy (e.g., Humphrey and Frankenberg, 2023).

Independently, tree sway frequency changes have also been shown to be related to canopy interception and evaporation of rain (van Emmerik et al., 2018; Ciruzzi and Loheide II, 2021), as well as interception of snow (Raleigh et al., 2022). As precipitation accumulates on the vegetation, the increased weight of the water on the branches and leaves/needles leads to a decrease in the overall tree sway frequency because the tree's intrinsic frequency mode changes with the altered mass distribution (e.g., Selker et al., 2011; Jackson et al., 2021; Ciruzzi and Loheide II, 2021; Raleigh et al., 2022).

One goal of our study is to evaluate how well VOD and tree sway frequency estimate the amount of liquid precipitation that is retained by the subalpine canopy during and after rainfall at the Niwot Ridge Forest US-NR1 AmeriFlux site. We take this analysis a step further, by relating these measurements to two levels of eddy-covariance ET which provides a quantitative measure of canopy evaporation. Our novel analysis of the diel cycle, provides additional insight into the canopy evaporation dynamics at our US-NR1 forest site. To help evaluate the VOD, tree sway frequency, and ET measurements, we supplement our analysis with other US-NR1 measurements (surface wetness and bole water content).

Though the focus of our study is on observations, we also want to compare the VOD and tree sway observations with land-surface model results. In Burns et al. (2018) the Community Land Model CLM4.5 (Oleson et al., 2013) components of the latent heat flux (transpiration, canopy evaporation, and soil evaporation) were compared with the measured ecosystem-scale latent heat flux at the US-NR1 site. Since the VOD and tree sway frequency measurements are related to canopy water content (and thus canopy evaporation), these observations are compared with CLM4.5-modeled canopy surface water content.

The paper is organized as follows: Section 2 has the site description (Sect. 2.1), the measurements of VOD (Sect. 2.2.1), tree sway (Sect. 2.2.2), and ET (Sect. 2.2.3), the CLM4.5 model (Sect. 2.3) and our data analysis techniques (Sect. 2.4). Section 3 has the results and discussion showing: time series (Sect. 3.1), diel cycle composites (Sects. 3.2 and 3.3), comparisons with CLM4.5 (Sect. 3.4), variations in the VOD and tree sway time series (Sect. 3.5), and a few thoughts about limitations and possible improvements to our study (Sect. 3.6). Section 4 has our final conclusions.

# 105 2 Materials and Methods





### 2.1 Site Description

The Niwot Ridge Subalpine Forest AmeriFlux site (US-NR1; Blanken et al. (1998-present)) is located on a remnant glacial moraine in the Colorado Rocky Mountains about 8 km east of the Continental Divide. The forest near the 26-m US-NR1 main tower is primarily composed of subalpine fir (*Abies lasiocarpa* var. *bifolia*), lodgepole pine (*Pinus contorta*), and Englemann spruce (*Picea engelmannii*) with a tree density of around 4000 trees ha<sup>-1</sup> (0.4 trees m<sup>-2</sup>), a leaf area index (LAI) of 3.8–4.2 m<sup>2</sup>m<sup>-2</sup>, and tree heights of 13–15 m (Monson et al., 2010; Burns, 2018). The trees near the tower (as-identified by lidar) are shown in Fig. 1. The mean annual temperature at US-NR1 is around 2°C. According to the Köppen–Geiger climate classification system (Kottek et al., 2006) the site is type Dfc which corresponds to a cold, snowy/moist continental climate with precipitation spread fairly evenly throughout the year; the long-term mean annual precipitation at the site is around 800 mm and snow typically covers the ground from mid-November until late May (Burns et al., 2015). During the warm-season, precipitation primarily occurs during afternoon convective storms generated by upslope mountain-plain wind patterns

(e.g., Turnipseed et al., 2004; Zardi and Whiteman, 2013). More information about the US-NR1 site can be found in Burns et al. (2015) and references therein or on-line at http://ameriflux.lbl.gov.

Frozen tree boles have been shown to limit sap flow within the US-NR1 trees and control seasonal transpiration onset and dormancy (Bowling et al., 2018) as well as changing the bole flexural properties which modifies the tree sway frequency (e.g., Granucci et al., 2013; Gutmann et al., 2017; Raleigh et al., 2022). The freezing of internal tree water also changes the water permittivity which affects VOD (e.g., Schwank et al., 2021). To avoid the complications of frozen boles and align with previous work done during the warm-season at the US-NR1 site (e.g., Hu et al., 2010; Burns et al., 2015), we restrict our analysis to the "warm" season, which we define as starting when the boles thaw in spring and ending when they freeze in the fall.

### 125 **2.2** Measurements



The measurements used in our study are summarized in Table 1. Most of these measurements have been described in previous publications (e.g., Burns et al., 2015; Raleigh et al., 2022); in the Appendices we provide additional information on the measurements of ET (Appendix A1), turbulence (Appendix A2), net radiation (Appendix A3), surface wetness (Appendix A4), dielectric permittivity  $\varepsilon_a$  and tree bole water content (Appendix A5), and airborne lidar (Appendix A6). Precipitation measurements were collected at the nearby NOAA U.S. Climate Reference Network (USCRN) site (see Table 1). Below, we describe the VOD and tree sway measurements and our use of the above-canopy and subcanopy ET measurements.

# 2.2.1 Vegetation Optical Depth (VOD)

Two identical Septentrio PolaRx5 GNSS receivers, each connected to a Trimble Zephyr 2 antenna, were used to derive a 30-min time series of VOD measurement (Table 1). One GNSS (gnssA) was located above the forest canopy and one (gnssB) near the ground. The above-canopy gnssA antenna was mounted on the southwest corner of the 26-m US-NR1 tower and extended to just-above the top of the tower (see photo in the supplement, Fig. S1a). The subcanopy gnssB antenna was located on a tripod 1.27 m above the ground (Fig. S1b). The gnssA antenna was around 25 m east and 10 m south of the gnssB antenna (Fig. 1) and photos of gnssB taken from the top of the US-NR1 tower and on-the-ground looking at the subcanopy south and east of gnssB are shown in the supplemental material (Fig. S2). With GNSS signals there is a data gap to the north; in addition, the northwest and southeast corners of the tower had lightning dissipators that extended above the gnssA antenna. Therefore, GNSS signals from the north and areas affected by the dissipators were excluded from the VOD calculation (Fig. S3). The footprint of the VOD measurement is shown in Fig. 1 and further discussed in Appendix B.

The two Septentrio receivers collected signals from all GNSS constellations (e.g., GPS, GLONASS, Galileo, BeiDou) which are needed to make sub-hourly VOD measurements (Humphrey and Frankenberg, 2023). The GNSS uses L-band frequencies (1000–2000 MHz, wavelengths of 15–30 cm) because this frequency range is minimally affected by weather phenomena such as rain, snow, and clouds (Ogaja, 2011). The effect of ground reflections is negligible for the purpose of calculating VOD from the direct (line-of-sight) signals (Ghosh et al., 2024). The data processing and calculation of hourly VOD time series used the

**Table 1.** Instrumentation and measurements relevant to our study. Sensor heights are the distance above the ground. Sample rate refers to the highest rate of archived data samples [in Hz]; for data sampled less frequently than 1 Hz, the sample rate is shown as  $1/(\Delta T)$  where  $\Delta T$  is the time difference between samples [in seconds]. Deployment dates refer to a particular measurement at this location. Dimensionless measurements are indicated with units of "[-]".

| Measured Variables                                        | Symbol                | Units                                                                              | Sensor<br>type                           | Manufacturer <sup>a</sup> make/model                   | Sensor<br>Height(s)<br>[m] | Sample<br>Rate<br>[Hz]                    | Deployment Dates                                                | Additional<br>Comments |
|-----------------------------------------------------------|-----------------------|------------------------------------------------------------------------------------|------------------------------------------|--------------------------------------------------------|----------------------------|-------------------------------------------|-----------------------------------------------------------------|------------------------|
| Vegetation Optical<br>Depth                               | VOD                   | -                                                                                  | GNSS                                     | Septentrio,<br>model PolaRx5                           | 26.0<br>1.27               | $\begin{array}{c} 1/5 \\ 1/5 \end{array}$ | Aug 2022 – Aug 2023<br>Aug 2022 – Aug 2023                      | A                      |
| Tree Sway Frequency                                       | $f_{sway}$            | Hz                                                                                 | Accelerometer                            | GCDC,<br>model X16-1D                                  | 12.0                       | ≈12                                       | May 2016 - Oct 2023                                             | В                      |
| Air Temperature                                           | $T_a$                 | °C                                                                                 | Platinum resistance<br>thermometer (PRT) | Vaisala, HMP-35D<br>REBS, model THP-1                  | 21.5<br>2.0                | 1<br>1                                    | Nov 1998 – present<br>Dec 2014 – present                        | C                      |
| Bole Temperature                                          | $T_{bole}$            | °C                                                                                 | Type E<br>Thermocouple                   | Omega,<br>FF-E-20-TWSH-SLE                             | 1.5                        | 1/900                                     | Jul 2017 – present                                              | D                      |
| Bole Apparent<br>Dielectric Permittivity                  | $\varepsilon_a$       | -                                                                                  | Capacitance/<br>Frequency Domain         | Meter,<br>model GS3                                    | 1.5                        | 1/900                                     | Jul 2017 – present                                              | E                      |
| Wetness                                                   |                       | 0=dry, 1=wet                                                                       | Circuit Board                            | CSI,<br>model 237                                      | 13.5                       | 1                                         | Nov 1998 – present                                              | F                      |
| Precipitation                                             | Precip                | $\mathrm{mm}\mathrm{h}^{-1}$                                                       | Weighing Gauge                           | Geonor,<br>model T-200B                                | 1.5                        | 1/3600                                    | Oct 2003 – present                                              | G                      |
| Evapotranspiration  ⇒ 3-D Wind Components, and Turbulence | ET $u, v, w, (u_*)_z$ | $\begin{array}{c} \operatorname{mm} h^{-1} \\ \operatorname{m} s^{-1} \end{array}$ | Sonic<br>Anemometer                      | CSI, CSAT3<br>CSI, CSAT3<br>CSI, CSAT3                 | 21.5,<br>5.7,<br>2.5       | 10,<br>10,<br>10                          | Nov 1998 – present<br>Aug 2003 – present<br>Aug 2003 – present  | Н                      |
| ⇒ Water Vapor Density                                     | $ ho_v$               | ${\rm mgm^{-3}}$                                                                   | Infrared Gas<br>Analyzer (IRGA)          | LI-COR, LI-7200<br>LI-COR, LI-7500A<br>LI-COR, LI-7500 | 21.5,<br>2.5,<br>2.5       | 20,<br>10,<br>10                          | May 2014 – present<br>Sep 2014 – present<br>Jul 2003 – Sep 2023 |                        |

<sup>&</sup>lt;sup>a</sup> Company websites and acronyms used: CSI: Campbell Scientific, Inc., Logan, UT 84321 (https://www.campbellsci.com); GCDC: Gulf Coast Data Concepts, LLC., Waveland, MS 39576 (https://gulfcoastdataconcepts.com); Geonor: Augusta, NJ 07822 (https://www.geonor.com); LI-COR: LI-COR Biosciences, Lincoln, NE 68504 (https://www.licor.com); Omega: Omega Engineering Inc., Norwalk, CT 06854 (https://www.omega.com); REBS: Radiation and Energy Balance Systems, Inc. Seattle, WA 98115 (206-624-7221); Septentrio: 3001 Leuven, Belgium (https://www.septentrio.com).

A: See Septentrio (2001); each PolaRx5 GNSS receiver is connected to a Trimble model Zephyr 2 antenna. Both GNSS systems were co-located at the top of the US-NR1 main tower between 17 June and 31 July, 2022. The 5-sec sampling interval refers to the GNSS BINary EXchange (BINEX) data files.

**B**: See Sect. 2.2.2 and Raleigh et al. (2022) for additional details; the accelerometer is connected to a single spruce tree. The detrended tree sway frequency uses the symbol  $f_d$ .

C: See Burns et al. (2015) for additional details.

**D**: See Bowling et al. (2018) for additional details.

E: See Appendix A5 for additional details. When purchased, the GS3 sensor was manufactured by Decagon Devices.

**F**: See Appendix A4 and Burns et al. (2015) for additional details; the CSI 237 sensor was oriented horizontally just-below canopy-top. The output from the sensor has been normalized so that a value of zero corresponds to dry conditions while a value of one corresponds to completely wet conditions. Values between 0 and 1 correspond to "slightly wet" conditions.

G: See Diamond et al. (2013) for additional details; Precipitation is from the NOAA U.S. Climate Reference Network (USCRN) Hills Mills site which uses a Small Double Fence Intercomparison Reference (SDFIR) type of wind shield around the precipitation gauge, and is located in a clearing about 500 m northeast the US-NR1 main tower.

H: The planar-fit wind components are in the streamwise u, crosswind v, and vertical w directions. Evapotranspiration ET is calculated from the covariance between vertical wind w' and water vapor  $\rho'_v$  fluctuations. See Appendix A1 and Burns et al. (2015) for additional details. The sonic anemometers were also used to calculate a local friction velocity  $(u_*)_z = ((\overline{u'w'})_z^2 + (\overline{v'w'})_z^2)^{0.25}$  at each sonic height, z.

Figure 1. The location of the two GNSS antennas (gnssA and gnssB) are shown as stars on a 90 m x 90 m map relative to the location of the US-NR1 26-m tower. The gnssA antenna is at the top of the US-NR1 tower and gnssB is located in the forest 1.27 m above the ground on a tripod. The background colors show vegetation heights calculated from airborne lidar data within 0.5 m x 0.5 m regions. The filled circles indicate tree locations that are shorter than 12 m and open circles indicate trees taller than 12 m. The outer black circle around gnssB shows the VOD footprint area with a radius of  $\approx$  30 m, while the inner black circle shows an alternate VOD footprint based on examination of the skyplot shown in Fig. S3. The VOD footprint is further discussed in Appendix B.

L1 band (1575 MHz) and followed the procedures in Humphrey and Frankenberg (2023). A brief summary of the VOD data-processing steps is as follows: (1) calculate the GNSS signal attenuation due to the forest by comparing the GNSS signal strength from the forest receiver (gnssB) to the one above the canopy (gnssA), (2) use the signal-strength difference to estimate the forest transmissivity  $\gamma$ , and (3) calculated an initial estimate of VOD from VOD =  $-\ln(\gamma)\cos(\theta)$  (where  $\gamma$  and  $\theta$  are the canopy transmissivity and incidence angle, respectively). Because the GNSS samples the forest at irregular temporal intervals and angles, the long-term mean as a function of azimuth and elevation angle is used to improve the precision of the hourly measurements (Humphrey and Frankenberg, 2023). Linear interpolation over time was used to convert the hourly data to a 30-min time series. For the two months prior to deploying gnssB on the ground, both GNSS were mounted side-by-side at the top of the tower to determine the appropriate system settings, ensure good inter-system agreement, and exclude any systematic biases. For our study period, the VOD data were restricted to the warm season, split between 2022 (Sept–Oct) and 2023 (Jun–Aug).

# 2.2.2 Tree sway motion



The tree sway motion was measured with a single three-axis accelerometer (Gulf Coast Data Concepts, model X16-1D) which is the same as used by Raleigh et al. (2022). The X16-1D sensor was located at a height of around 9 m on a 13 m tall spruce tree on the northwest side of the US-NR1 main tower. Processing of the raw tree sway accelerations [m s<sup>-2</sup>] into frequency [Hz] used the method described by Raleigh et al. (2022), except that a 30-minute analysis window was used. A brief summary of the data-processing steps is as follows: (1) spectral analysis of the 12-Hz acceleration data to determine a primary frequency of the tree-swaying motion  $f_{sway}$  for each 30-min period, (2) calculate a sliding 72-hr mean-filtered  $f_{sway}$  time series, (3) remove any 30-min  $f_{sway}$  outliers relative to the 72-hr mean-filtered data or values with low spectral power (i.e., due to low wind speeds), and (4) gap-fill any missing or removed 30-min time periods in the  $f_{sway}$  time series using splines. A detailed description of these steps can be found in Sect. 3.2 of Raleigh et al. (2022), with raw accelerometer data available from Raleigh (2021).

The natural sway frequency  $f_{sway}$  of a coniferous tree acts like a damped harmonic oscillator; therefore, it does not depend on wind speed. This is highlighted in Sect. 3 of Raleigh et al. (2022) as well as many other studies (e.g., Moore and Maguire, 2004; Van Emmerik et al., 2017; Jackson et al., 2021), who show that  $f_{sway}$  can be described by the cantilever model:

$$f_{sway} \propto \frac{1}{2\pi} \left(\frac{K}{m}\right)^{0.5},$$
 (2)

where K is the flexural rigidity of the tree and m is the mass of the tree, including the branches and needles. As precipitation accumulates on the needles and branches, m changes which alters  $f_{sway}$ . Therefore,  $f_{sway}$  is primarily a structural property of the tree that depends on mass (tree biomass plus water in/on the tree), elasticity (which varies with tree temperature, thermal state, and water content), and tree geometry (tree height and diameter). Changes to the magnitude of horizontal wind speed will modify the amplitude of the tree motion, not the frequency.

Tree sway frequency  $f_{sway}$  data have been collected at the site starting in fall of 2014 until summer 2023 with some missing periods. During spring there was a gradual decrease in the sway frequency which reached a minimum sometime in July and then increased into late summer and fall (Fig. S5). This was presumably due to a combination of changes to the temperature and/or moisture content of the tree, both of which have well-known effects on wood elasticity (e.g., Gerhards, 1982; Gao et al., 2015). Because we are interested in short-term (hourly) changes in tree sway due to precipitation we have used a sliding 10-day median filter to remove the low-frequency trend, as shown for 2019 in Fig. S5. For the detrended tree sway frequency  $f_d$ , we chose to set the mean to a value near 1, which is only slightly larger than the typical mean value of around 0.94 Hz. We have left the units of  $f_d$  in Hz, but this only applies to deviations from the mean. As we show in supplemental Fig. S6 (and will further discuss in Sect. 3.2), the diel cycle analysis using either  $f_{sway}$  or  $f_d$  produced similar results.

# 2.2.3 Above-canopy and subcanopy ET




Technical details about the 21.5 m and 2.5 m ET measurements are in Table 1 and Appendix A1. The US-NR1 subcanopy turbulent fluxes have been used previously in winter to estimate snow interception (Molotch et al., 2007; Harvey et al., 2025) and examine changes in snowpack temperature (Burns et al., 2014). Here, we use assume that the above-canopy flux represents total ecosystem ET (i.e., tree transpiration + canopy evaporation + ground evaporation), whereas the subcanopy flux represents only the ground evaporation and any transpiration from low vegetation (i.e., below the 2.5 m subcanopy sensor). Therefore, the difference between the above-canopy and subcanopy fluxes isolates the transpiration and intercepted water evaporation (e.g., Blanken et al., 1997; Wilson et al., 2001; Molotch et al., 2007; Paul-Limoges et al., 2020; Wolf et al., 2024). Possible issues with this technique are further discussed by Wolf et al. (2024) and in Sect. 3.6. We will categorize ET by different wetness conditions (Sect. 2.4) to get a quantitative estimate of canopy evaporation from the forest.

### 2.3 Modeling with CLM4.5

Because the magnitudes of VOD and tree sway frequency are related to water content within the canopy, we decided it was worthwhile comparing these observations with canopy water content determined by a land-surface model. Previous studies of canopy evaporation at US-NR1 by Burns et al. (2018) used the big-leaf land-surface model CLM4.5, making these results readily available for inclusion in our current study.

When liquid water is present on the canopy, CLM4.5 uses the leaf boundary layer resistance  $r_b$  scaled by the potential evaporation from wet foliage to evaporate the canopy water. The expression for  $r_b$  is,

$$r_b = \frac{1}{C_v} \left( \frac{U_{av}}{d_{leaf}} \right)^{-0.5}$$
, (3)

where  $C_v$  is the turbulent transfer coefficient between the canopy surface and canopy air ( $C_v = 0.01 \text{ m s}^{-0.5}$ ),  $U_{av}$  is an estimate of the wind speed within the subcanopy, and  $d_{leaf}$  is the characteristic dimension of the leaves in the direction of wind flow (with a default value of 0.04 m). The big-leaf of CLM4.5 is divided into areas where evaporation (wet areas) and transpiration (dry areas) occur and scaled accordingly. The water and energy budgets within CLM4.5 must be balanced and a 30-step iterative

process is used to simultaneously solve the system of equations. For full details on these iterative steps see Chapter 5 in Oleson et al. (2013).

The CLM4.5 parameter maximum leaf wetted fraction  $f_{wet}^{max}$  limits the area of the leaf surface that is wet. A larger  $f_{wet}^{max}$  value decreases the dry portion of the canopy undergoing transpiration and increases the leaf area for evaporation of canopy water. For  $f_{wet}^{max} = 1$ , the entire leaf is covered in water, which is the CLM4.5 default value (Oleson et al., 2013; Burns et al., 2018). In Burns et al. (2018), 14 different CLM4.5 configurations were examined; they concluded that  $f_{wet}^{max} = 0.02$  was more appropriate for a needleleaf forest than  $f_{wet}^{max} = 1$ . One of the CLM4.5 variables ("H2OCAN") represents the amount of intercepted water on the canopy surfaces. Because changes in H2OCAN should be similar to changes in VOD, we use four of the CLM4.5 cases from Burns et al. (2018) to examine, "CLM Canopy H2O". The cases we chose to include are A1 and F2 ( $f_{wet}^{max} = 1$ ) and B0 and G1 ( $f_{wet}^{max} = 0.02$ ). Other differences between the A1, F2, B0, and G1 configurations are related to the subcanopy turbulent transfer coefficient, stability criteria, and use of friction velocity; anyone curious for additional details can refer to Table 1 in Burns et al. (2018). The mechanisms that control the evaporation of warm-season intercepted water in CLM4.5 are the same as those in CLM5 (Lawrence et al., 2019).

### 2.4 Data Analysis




To study the impact of rainfall interception on the evaporative fluxes and VOD, we followed the methodology of Burns et al. (2015) and label days when the daily rainfall exceeded 3 mm as "Wet" days. The choice to use 3 mm as the wet-day criterion was a balance between effectively capturing the effect of precipitation and providing enough wet periods to improve the wet-day statistics. To examine the effect of precipitation on VOD, ET, and the other variables we use the following strategy: first, we designate the precipitation state of the day of interest as either "Wet" or "Dry", we then indicate the preceding-day precipitation state with a lower-case letter ("w" for wet or "d" for dry) and create composite diel cycles for dDry (dry day followed by a dry day), dWet (dry day followed by a wet day), wWet (wet day followed by a wet day), and wDry (wet day followed by a dry day) days. Only 30-min periods when all variables of interest were available were used to create the diel cycle composites.

It is important to realize that eddy-covariance fluxes are sensitive to rain accumulating on sonic anemometer transducers and gas-analyzer windows so therefore on dWet and wWet days either gap-filling of ET is needed or periods with heavy rain are excluded from the analysis. Therefore, our results focus on wDry days where there is zero to little precipitation (daily total < 3 mm) and the flux instruments have a chance to dry out. In order to improve the statistics for certain variables, we use additional years of data. For example, for the ET diel-cycle analysis, we used ET data from the warm season between years 2004 and 2022. Whenever this occurs, it will be described in the figure caption.

Unless otherwise noted, ET and all other statistics are calculated over 30-min periods. The sampling rate of the primary variables are listed in Table 1. To ease comparison with other studies (e.g., Klaassen et al., 1998; Humphrey and Frankenberg, 2023), we have expressed ET and the rate of precipitation in units of mm of  $H_2O$  per hour (mm h<sup>-1</sup>).

### 3 Results and Discussion

### 3.1 Time series





Figure 2 demonstrates how VOD, tree sway frequency  $f_{sway}$ , and the dielectric permittivity  $\varepsilon_a$  (our proxy for tree water content) were modified by precipitation in fall 2022. The "wet" days are highlighted by thin vertical black lines and the blue squares in Fig. 2c. Similar time series for the 2023 spring/summer period are in the supplement (Fig. S7). As mentioned in Sect. 2.4, our analysis focused on the period after the boles had thawed in spring and before they froze in fall, and the precipitation was rain.

As one would expect, bole temperature varied between the daytime and nighttime air temperatures (Fig. 2a). We ended our 2022 analysis period when the bole temperature went below  $0^{\circ}$ C on 4 October 2022 (DOY 277). The freezing of tree sap-water and dependence of water permittivity on temperature has been shown to have a large effect on VOD (e.g., Schwank et al., 2021) as well as changes to the tree flexural stiffness which affects tree sway (e.g., Granucci et al., 2013; Gutmann et al., 2017). This change in behavior in VOD,  $f_{sway}$ , and  $\varepsilon_a$  occurred just-before DOY 300 in Fig. 2 when the air temperature dipped below  $-5^{\circ}$ C and the boles became frozen.

Variations in VOD were primarily due to changes in the subcanopy GNSS (gnssB signal strength anomaly) whereas the gnssA anomaly at the top of the tower was stable over the entire period (Fig. 2b), including in colder conditions. This indicates that the method used to calculate VOD worked as expected, and it was gnssB in the forest that was affected by the canopy-intercepted precipitation. The bias in the signal strength anomalies between the two GNSS receivers is also indicative of the effect of the canopy.

During precipitation tree sway frequency dropped on the order of 0.05 Hz (Fig. 2b) which is consistent with previous results (e.g., Ciruzzi and Loheide II, 2021). In addition to the frequency drops due to precipitation, there was a more gradual low-frequency change over time which we have removed as described in Sect. 2.2.2. The VOD time series also has a very weak low-frequency trend that we did not remove (because it's removal would not affect our results). The dielectric permittivity of the sap-water in the tree boles increased in magnitude following precipitation events and decreased sharply as the boles froze (Fig. 2d).

# 265 3.2 Diel cycle during dDry, dWet, wWet, and wDry days

To summarize how precipitation affected our observations, net radiation  $R_{net}$ , ET, VOD, detrended tree sway frequency  $f_d$ , surface wetness, and precipitation from our analysis period (fall 2022 and spring/summer 2023) are composited in Fig. 3 based on the dDry, dWet, wWet, and wDry precipitation states described in Sect. 2.4. During wet periods  $R_{net}$  (Fig. 3a) and ET (Fig. 3b) were both reduced below dDry conditions; however, on a wDry day there was an increase in ET (relative to dDry conditions) of around  $0.1 \,\mathrm{mm}\,\mathrm{h}^{-1}$  at mid-day, that is due to evaporation of recently-fallen rainwater on the vegetation and ground (Burns et al., 2015, 2018).

On a dDry day, VOD was around 0.37 and fairly constant over the diel cycle (Fig. 3d). As precipitation occurred (i.e., dWet and wWet days), VOD increased to between 0.55 and 0.61. The maximum VOD value of 0.61 occurred on the after-

Figure 2. Time series from 2022 of (a) mid-day (10–14 MST) air temperature  $T_a$  at 21.5 m and 2 m heights; nighttime (22-2 MST) 2 m  $T_a$ ; and 30-min bole temperature (at 12 mm depth), (b) GNSS-based vegetation optical depth VOD and signal strength anomalies, (c) tree sway frequency  $f_{sway}$ , (d) apparent dielectric permittivity  $\varepsilon_a$ , and (e) precipitation and surface wetness. VOD and  $\varepsilon_a$  are both dimensionless as indicated by "[-]". The thin vertical black lines and filled blue periods in panel (c) indicate when wet days occurred. The thicker vertical black line at day of year 277 indicates the cut-off date of our study.

Figure 3. The warm-season mean composite diel cycle of (a) net radiation  $R_{net}$ , (b) evapotranspiration ET, (c) detrended tree sway frequency  $f_d$ , (d) vegetation optical depth VOD and wetness, and (e) precipitation for each precipitation state (dDry, dWet, wWet, and wDry) where the precipitation state for each diel cycle is identified above panel (a) and separated by thin vertical black lines. In (a) and (b), the dDry diel cycle is repeated in the dWet, wWet and wDry states as a red line. The right-axis of (b) shows ET as latent heat flux with units of [W m<sup>-2</sup>].  $\lambda$ E. The diel cycle is calculated from 30 min measurements during the warm-season for August-October 2022 and June-August 2023 and the approximate number of days (N) used to create each composite is shown in panel (a). More information on the measurements, precipitation state, and data compositing are within the text.

Figure 4. The relationship between vegetation optical depth VOD and detrended tree sway frequency  $f_d$  from the composite diel cycle shown in Fig. 3 is shown. For comparison, we have included VOD versus non-detrended tree sway frequency  $f_{sway}$  (see legend); the red lines are linear-fits with the fit coefficients using  $f_d$  and  $f_{sway}$  listed near each line.

noon/evening of a wWet day which matches the timing of the highest precipitation amounts (Fig. 3e). On a wDry day, VOD was initially high (at 0 MST), but by the afternoon had recovered to near the dDry value.




The impact of precipitation on the tree sway frequency (Fig. 3c) is almost the mirror image of the VOD pattern (Fig. 3d), and the linear relationship between VOD and sway frequency is shown in Fig. 4. Here, it is clear that the low frequency detrending of the tree sway data (see Sect. 2.2.2) did not significantly modify the results (i.e., the slope of VOD vs  $f_{sway}$  was -4.26 compared to -4.34 for VOD vs  $f_d$ ). It is highly encouraging that these two completely independent methods (VOD and tree sway) are showing consistent changes as the canopy becomes wet and dries.

The surface wetness data closely matched the VOD trends with an important exception: the wetness sensor rapidly dried in the early morning (at around 8:00 MST, i.e., when  $R_{net}$  increased sharply) for the dWet, wWet, and wDry days. We describe some of the limitations with the wetness sensor in Appendix A4, and summarize our thoughts here: (i) the wetness sensor was located near the top of the canopy and therefore had increased exposure to radiation and wind, (ii) the wetness sensor is a small flat plate that does not properly represent the water-holding capacity of an evergreen branch/needles, and (iii) the mountain-plain wind pattern at the site leads to clear mornings (which encourage evaporation), while clouds/precipitation typically occurred in the afternoon or evening (Fig. 3e). Therefore, wetness data should be considered an estimate of how long it takes the water to evaporate from a flat, hard, open surface near canopy-top. At best, it could represent leaf wetness in the

upper part of the forest canopy, but it is not indicative of the entire forest drying (and therefore reasonable to see the wetness sensor drying faster compared to what can be inferred from VOD or tree sway motion).

The time series of dielectric permittivity (Fig. 2d) suggests that  $\varepsilon_a$  would provide valuable information about how the internal sap-water of the trees was modified by precipitation. However, the dielectric permittivity included many large jumps and shifts that we could not explain and were corrected as described in Appendix A5. Over the diel cycle, it is expected that morning increases in transpiration will reduce the trunk water content and trunk diameter (e.g., Zweifel and Häsler, 2001; Oberleitner et al., 2022). As shown in Fig. S8c, on dDry, dWet, and wDry days,  $\varepsilon_a$  reached a maximum just-after sunrise (at around 7:00 MST) followed by an afternoon minimum (after the mid-day transpiration maximum). On wWet days, when transpiration was reduced, the diel cycle of  $\varepsilon_a$  was fairly constant. In general,  $\varepsilon_a$  on wet days was slightly elevated compared to dry days, but the dDry and wDry days were not different enough to extract insightful information, as we had for VOD and tree sway (Fig. 3c, d).

There is an important implication from the dDry periods (when most of the ET is transpiration, not evaporation). Though  $\varepsilon_a$  had a dDry diel cycle with a clear early-morning maximum and late afternoon minimum, VOD had relatively small variation without any apparent pattern (Fig. 5). This suggests that, at our site, VOD changes were largely controlled by water on the canopy *surfaces*, not the the internal water content of the trees. In contrast, tree sway frequency had a diel pattern on dDry days (Fig. 5c) that was similar to that of  $\varepsilon_a$  (Fig. 5e), suggesting that tree sway frequency was more affected by internal tree water content changes than VOD. Ciruzzi and Loheide II (2019) has also shown that tree sway motion followed changes in internal tree water content.

The lack of a clear diel pattern in VOD during dDry conditions was a surprising result because previous studies (e.g., Holtzman et al., 2021; Humphrey and Frankenberg, 2023; Yao et al., 2024) have shown a VOD diel pattern that they related to the changes in internal water content of the forest/trees. As we looked closer at this, we realized that US-NR1 VOD in dDry conditions was much lower than VOD from the other sites (both the mean value and the diel range in dry conditions). For example, the US-NR1 dDry VOD diel range is 0.36 to 0.38 (Fig. 5d) whereas the VOD diel range in the study by Holtzman et al. (2021) was on the order of 0.85 to 1.1 (their Fig. 4) and that of Yao et al. (2024) had a range of 0.62 to 0.65 (their Fig. 2). Both of these studies are from deciduous forests in the eastern USA. Though we cannot definitively explain the reason for low VOD at the US-NR1 site, we offer a few possible explanations: (i) the US-NR1 forest has a lower tree density, (ii) the internal water content of the coniferous US-NR1 trees is lower and stored differently within the tree bole than broadleaf trees in the more humid locations (Hacke et al., 2015; Luo et al., 2020), or (iii) there is too much noise in the VOD measurements to properly capture the true diel cycle in internal tree water content during dDry conditions (which is when the VOD signal is weakest).

# 3.3 Diel cycle during a wDry day







We now take a closer look at the wDry diel cycle in Fig. 3 which is re-plotted in Fig. 6 with a few important changes. First, for ET we used the warm-seasons between 2004–2022 to create the wDry ET composite and include the subcanopy (2.5 m) ET fluxes (Fig. 6a; the other precipitation states for 2004–2022 are shown in supplemental Fig. S9). Second, we use the ET

Figure 5. The warm-season mean composite diel cycle of (a) net radiation  $R_{net}$ , (b) evapotranspiration ET (and latent heat flux  $\lambda$ E), (c) detrended tree sway frequency  $f_d$ , (d) vegetation optical depth VOD and wetness, and (e) apparent dielectric permittivity  $\varepsilon_a$  for dDry conditions. In (a)–(b) and (e), the results are from years 2018–2021 (N = 322 dDry days) while in (c) and (d) the results are from years 2022–2023. For clarity, each panel lists the average number days (N) that are used to create the composite.

diel cycle difference between dDry and wDry conditions to estimate the 2.5 m ground evaporation and 21.5 m canopy and ground evaporation (Fig. 6b). Third, we add a panel of CLM4.5 wDry Canopy H2O results for comparison to the observations (Fig. 6e). We also note that, wDry days have much less actual rain than wWet and dWet days (Fig. 3e), so ET on a wDry day has fewer missing or gap-filled data caused by instrument issues due to heavy rain (Sect. 2.4).

Figure 6. The warm-season mean composite diel cycle of (a) evapotranspiration ET on wDry and dDry days at 21.5 m and 2.5 m, (b) the ET difference between wDry and dDry conditions, (c) detrended tree sway frequency  $f_d$ , (d) vegetation optical depth VOD and wetness, and (e) CLM4.5 values of the canopy intercepted water content for the A1, B0, F2, and G1 cases (see Sect. 2.3 for case details). Panels (c)—(e) are for wDry conditions. Linear fits to  $f_d$  and VOD before and after 14:00 MST are shown as red lines (the pre-14:00 MST slope is S1, and the post-14:00 slope is S2). In (a)—(b), the results are from years 2004–2022 (N = 176 wDry days) while in (e) the CLM4.5 results are from years 1999–2003 and 2006–2014 (as in Burns et al. (2018)). For clarity, each panel lists the average number of days (N) that are used to create the composite.

When we examine the VOD and tree sway data over the 24-hour wDry diel cycle, a fairly obvious change in slope versus time occurred at around 14:00 MST (Fig. 6c, d). We highlighted this by a linear fit through each period, as shown by the red lines in Fig. 6c, d. For tree sway (Fig. 6c), the slope changed from  $0.052 \,\mathrm{Hz} \,(\mathrm{day})^{-1}$  to  $0.008 \,\mathrm{Hz} \,(\mathrm{day})^{-1}$  (a slope decrease by a factor of 6.5). For VOD (Fig. 6d), the slope changed from  $-0.193 \,(\mathrm{day})^{-1}$  to  $-0.034 \,(\mathrm{day})^{-1}$  (a slope decrease by a factor of 5.6). For both tree sway and VOD the smaller slopes after 14:00 MST were indicative of the intercepted water being removed from the canopy by evaporation. The day-to-day variability in VOD and sway frequency was also smaller in the afternoon/evening of a wDry day compared to the early morning and midday values (Fig. S10c,d). As discussed in Sect. 3.2, the wetness sensor rapidly dried between 6:00–8:00 MST (Fig. 6d), due to the sensor characteristics and location.






Above-canopy and subcanopy ET on dDry days both peaked around an hour before noon MST at around  $0.25 \,\mathrm{mm}\,h^{-1}$  and  $0.06 \,\mathrm{mm}\,h^{-1}$ , respectively (Fig. 6a). In contrast, the corresponding ET values on wDry days had a similar pattern, but with peak mid-day values of  $0.33 \,\mathrm{mm}\,h^{-1}$  and  $0.08 \,\mathrm{mm}\,h^{-1}$ . The higher magnitude ET on wDry days (relative to dDry) was due to the presence of liquid water on the trees and ground that enhanced evaporation; we show the enhancement amount as  $\Delta$ ET in Fig. 6b. Between 0–6 MST on a wDry day,  $21.5 \,\mathrm{m}\,\Delta$ ET was fairly constant suggesting a canopy evaporation rate of around  $0.02 \,\mathrm{mm}\,h^{-1}$ . It is important to note that  $2.5 \,\mathrm{m}\,E$ T during the nocturnal hours was very small ( $< 0.01 \,\mathrm{mm}\,h^{-1}$ ) which suggests that nocturnal evaporation was primarily from the canopy, not the ground. By the end of a wDry day (between  $18-24 \,\mathrm{MST}$ ), there was no more liquid water to evaporate, and  $21.5 \,\mathrm{m}\,\Delta$ ET became small ( $< 0.01 \,\mathrm{mm}\,h^{-1}$ ) (Fig. 6b). This ET result is consistent with the VOD and tree sway observations which also suggest that the intercepted canopy water was fully evaporated by the afternoon or early evening on a wDry day.

If we only consider the nocturnal data on a wDry day, then the VOD, tree sway and ET results are in good agreement (Fig. 6). VOD and tree sway data suggest that the evaporation rate of canopy-intercepted rain is fairly constant (Fig. 6c, d), while the ET data tell us that nocturnal ground evaporation was small/negligible and the evaporation of canopy-intercepted rain was fairly constant at around 0.02 mm h<sup>-1</sup> (Fig. 6b).

For the wDry daytime data, the comparison between VOD and  $f_d$  with ET is less clear. The VOD and  $f_d$  results imply that the canopy evaporation of intercepted water was nearly linear with time from the early morning (nocturnal) hours up until the time that the intercepted water was completely (or mostly) evaporated, at around 14:00 MST. This would mean that the morning sunrise (i.e., increases in  $R_{net}$ , transpiration, and air temperature) had a minimal effect on canopy evaporation. In contrast, between around 6:00–10:00 MST, 21.5 m  $\Delta$ ET increased from a value of around 0.02 mm h<sup>-1</sup> to around 0.08 mm h<sup>-1</sup> (Fig. 6b). A portion of this increase can be explained by increased ground evaporation starting just-after 8:00 MST. However, the increase in 21.5 m  $\Delta$ ET between 6:00–8:00 MST is inconsistent with the VOD/tree sway measurements which suggest the canopy evaporation was fairly constant during that period. One possible explanation for the increase in  $\Delta$ ET between 6:00–8:00 MST is that transpiration between dDry and wDry days are not the same. However, previous work at the US-NR1 site has suggested that transpiration is similar on dDry and wDry days (Burns et al., 2015) and tree-to-tree variability from the 2007 sap flow data of Hu et al. (2010) was small (Fig. S11b-d).

Overall, our ET-based canopy evaporation estimates of  $0.02 \text{ mm h}^{-1}$  to  $0.08 \text{ mm h}^{-1}$  are on the lower side of other studies canopy evaporation rates, which typically range between  $0.03 \text{ mm h}^{-1}$  to  $0.45 \text{ mm h}^{-1}$  (Table 2). The ratio of mid-day canopy

**Table 2.** Examples of the range of canopy evaporation values sampled by other studies within evergreen forests. For a similar list of canopy evaporation rates from other studies see Table 2 in Klaassen et al. (1998) and Table II in McNaughton and Jarvis (1983).

| Study/Reference           | Forest Type                  | Study Methodology                     | Canopy<br>Evaporation<br>[mm h <sup>-1</sup> ] | Additional<br>Comments |
|---------------------------|------------------------------|---------------------------------------|------------------------------------------------|------------------------|
| Rutter et al. (1971)      | Corsican pine                | Estimated/Measured                    | 0.03-0.24                                      | A                      |
| Pearce et al. (1980)      | Evergreen-mixed              | Rain Gauges<br>(regression)           | 0.37                                           | В                      |
| Massman (1983)            | Douglas fir                  | Modeling                              | 0.03-0.45                                      | C                      |
| Grelle et al. (1997)      | Norway spruce,<br>Scots pine | Eddy-covariance                       | 0.05-0.1                                       | D                      |
| van der Tol et al. (2003) | Sitka Spruce                 | From the residual from Energy Balance | 0.12-0.2                                       | E                      |

A: Range of canopy evaporation was from 4 individual storms using Penman-Monteith and a water balance equation.

**B**: Found similar evaporation rates for daytime and nighttime.

C: The range of canopy evaporation rates listed is from 20 individual storms in Table I of Massman (1983).

**D**: The values of  $0.05 \text{ mm h}^{-1}$  and  $0.1 \text{ mm h}^{-1}$  are during wet periods when transpiration was assumed to be zero. For other periods maximum ET for this forest was on the order of  $0.6 \text{ mm h}^{-1}$ .

E: Measured sensible heat flux and net radiation so evaporation was determined from the energy balance residual. The mean daytime evaporation was  $0.174 \, \text{mm h}^{-1}$  while nighttime was  $0.076 \, \text{mm h}^{-1}$ .

evaporation to total ET on a wDry day seems reasonable [i.e.,  $EET^{-1} \approx 0.2$ ]. Furthermore, previous studies have suggested that canopy evaporation measurements tend to overestimate modeling results (e.g., van Dijk et al., 2015).

The lack of a change in VOD and sway frequency during the morning transition is consistent with previous canopy evaporation estimates in evergreen forests (e.g., Pearce et al., 1980), but counter to the general consensus that net radiation plays an important role in evaporation of canopy water (e.g., Klaassen, 2001). Our results suggest that turbulent mixing of dry air into the canopy airspace has a primary control on canopy evaporation, whereas the early-morning increases in net radiation and air temperature are of secondary importance. The US-NR1 site is in sloping terrain ( $\approx 6\%$  slope) that is subject to nocturnal drainage flows which provides a mechanism/energy for turbulent mixing (and horizontal advection) to occur. If the turbulence within the canopy is controlling canopy-water evaporation, one would expect it to be approximately constant from early morning to past sunrise. Previous studies have shown that above-canopy friction velocity  $u_*$  typically increases from a nocturnal value of around  $0.4 \text{ m s}^{-1}$  to over  $0.6 \text{ m s}^{-1}$  at mid-day (Burns et al., 2015). We found that subcanopy local  $(u_*)_z$  at 5.7 m, on a wDry day was fairly constant through the morning transition period with a slight increase at mid-day (Fig. 7). In addition,

Figure 7. The mean warm-season composite diel cycle of (a) net radiation  $R_{net}$ , (b) local friction velocity  $(u_*)_z$  at 21.5 m, 5.7 m, and 2.5 m (see legend), and (c) only showing subcanopy  $(u_*)_z$ . The lines with solid circles are the diel cycle in wDry conditions, whereas the red lines are the corresponding diel cycle in dDry conditions. These results are from years 2004–2022 where N is the number of wDry days, shown in the upper-right corner.

between 00:00 to 8:00 MST, 5.7 m  $(u_*)_z$  on a wDry day is elevated over that of a dDry day (Fig. 7c). This topic goes beyond the goals of the current study, but suggests that turbulence within the canopy could be playing a primary role in the evaporation of canopy-intercepted rainwater.

### 3.4 Comparison with the CLM4.5 model

As one would expect, CLM Canopy H2O increases for the dWet and wWet precipitation states depending on the value of the maximum leaf wetted fraction  $f_{wet}^{max}$  (Fig. 8c). Furthermore, the B0 and G1 cases ( $f_{wet}^{max} = 0.02$ ) retain more water within the

**Figure 8.** The warm-season mean composite diel cycle for dDry, dWet, wWet, and wDry conditions of (a) observed and CLM4.5 net radiation  $R_{net}$ , (b) observed and CLM4.5 evapotranspiration ET, (c) CLM4.5 canopy water content, and (d) precipitation. In (b) and (c) the CLM4.5 A1, B0, F2, and G1 cases are shown (see legend in panel (b) as well as Sect. 2.3 for case details). These results are from years 1999–2003 and 2006–2014 and use the same periods as Burns et al. (2018). The number of days (N) used for each diel cycle are shown in panel (a).

**Table 3.** Information about each of the 17 wDry periods analyzed in our study. The columns are: the wDry year and day of year (DOY); the daily precipitation amount on the preceding wet day and the wDry day; the time the precipitation started/stopped on the wet day; and the number of hours needed to shift the time series to align the ends of the precipitation events. The two precipitation amounts shown in bold are the minimum and maximum values used in Figs. 9 and 10.

|      |      | Daily Precip | itation | Precipita | tion Times |            |
|------|------|--------------|---------|-----------|------------|------------|
|      | wDry | wWet/dWet    | wDry    | Start     | End        | Time Shift |
| Year | DOÝ  | [mm]         | [mm]    | [MST]     | [MST]      | [h]        |
|      |      |              |         |           |            |            |
| 2022 | 229  | 8.50         | 0.00    | 06:30     | 15:30      | 8.5        |
|      | 234  | 10.90        | 0.00    | 10:00     | 19:00      | 5.0        |
|      | 239  | 4.10         | 0.00    | 14:30     | 19:30      | 4.5        |
|      | 254  | 4.53         | 0.00    | 00:00     | 19:30      | 4.5        |
|      | 265  | 7.70         | 0.37    | 10:30     | 24:00      | 0.0        |
|      | 275  | 3.40         | 0.00    | 12:00     | 15:30      | 8.5        |
|      |      |              |         |           |            |            |
| 2023 | 139  | 8.90         | 0.30    | 08:00     | 21:30      | 2.5        |
|      | 141  | 4.40         | 0.00    | 13:00     | 16:30      | 7.5        |
|      | 147  | 7.80         | 0.30    | 09:30     | 21:30      | 2.5        |
|      | 159  | 29.00        | 0.00    | 12:00     | 22:30      | 1.5        |
|      | 164  | 3.60         | 0.90    | 01:30     | 14:30      | 9.5        |
|      | 168  | 7.20         | 2.70    | 14:00     | 22:30      | 1.5        |
|      | 187  | 15.13        | 1.40    | 00:00     | 22:30      | 1.5        |
|      | 189  | 3.80         | 0.40    | 15:30     | 24:00      | 0.0        |
|      | 202  | 16.10        | 1.80    | 00:00     | 22:00      | 2.0        |
|      | 214  | 13.25        | 1.97    | 00:00     | 24:00      | 0.0        |
|      | 219  | 3.70         | 0.60    | 12:00     | 22:30      | 1.5        |
|      |      |              |         |           |            |            |

canopy than the A1 and F2 cases ( $f_{wet}^{max} = 1$ ; see Sect. 2.3 for details about each case), which are reflected in the differences in CLM4.5 ET between cases (Fig. 8b). Also note that on a wWet day, there is a sharp decrease in CLM Canopy H2O following sunrise on a wWet day, which is similar to what we observed with the tower wetness sensor (Fig. 3d and Fig. S9c).



If we focus on the wDry conditions, Canopy H2O shows a distinct decrease in the intercepted canopy water content between 6:00 and 11:00 MST (Fig. 6e); where the A1 and F2 cases ( $f_{wet}^{max} = 1$ ) had rapid drying between 6:00–8:00 MST, and the B0 and G1 cases ( $f_{wet}^{max} = 0.02$ ) between 6:00–11:00 MST. It appears that in the B0 and G1 cases, a small amount of water is retained by the canopy in the wDry afternoon and evening (the reason for this behavior is unknown). Relevant to our study, the important result is that all four cases show that CLM4.5 canopy evaporation increased at sunrise which is inconsistent with the VOD and tree sway frequency results. In fact, the CLM Canopy H2O behavior at sunrise is strikingly similar to that of the wetness sensor (Fig. 6d). This suggests that both the CLM4.5 model and the wetness sensor do not properly represent the water-holding capacity of an evergreen forest (as discussed in Sect. 3.2). If the nearly linear change in intercepted water content found with the VOD and tree sway observations are also found in other ecosystem types, this could be an area to improve the CLM model.

# 3.5 Variations in VOD and tree sway





Up to this point, we have focused on the composite mean results from our measurements (i.e., Fig. 6). The variability of tree sway frequency and VOD is also an important consideration, and Table 3 lists the precipitation amounts and timing for each of the 17 wDry days. Figure 9 shows each of the 17 wDry day time series of VOD and sway frequency. Here, we observe: (i) VOD has more rapid temporal variations than sway frequency, (ii) case-by-case variability in VOD and sway frequency during the drier conditions are smaller than during wet conditions (this is shown quantitatively in Fig. S10c,d), (iii) rainfall occurred on wDry days, but is less than our  $3 \text{ mm} (\text{day})^{-1}$  criteria, and (iv) rainfall occurred at all times of day during wWet and dWet days (Table 3). By highlighting the wet days with the highest and lowest precipitation by bold lines in Figure 9, it is clear that the largest wet-day precipitation amount (red bold line) induces a significant change to VOD and  $f_d$  whereas the lowest precipitation amount (olive dashed line) results in a rather small change to VOD and  $f_d$ .

We considered how the timing of the rainfall affected our results by shifting the precipitation periods so they align at the end of the wWet or dWet day (Fig. 10). The maximum shift to the time series was 9.5 hours, but most were less than 4 hours (Table 3). For wet days with multiple precipitation events, the final precipitation event of the day was used to align the time series. The choice of aligning the time series to the end of the wet days was arbitrary, but useful for comparison purposes. In Fig. 10a,b, the mean composite value of  $f_d$  and VOD is shown as a light blue line (with filled circles) in the "Post-Precip" period, which can be compared with the black line that is the composite of VOD and tree sway from Fig. 9. While the line from the shifted data is slightly smoother, the trend that we observed in the change of slope in VOD and sway frequency at around 14:00 MST are similar.

As an attempt to determine the key variables driving the changes in VOD and  $f_d$ , we created scatter plots for VOD (Fig. 11) and  $f_d$  (Fig. 12, where we have multiplied  $f_d$  by -1 to make the pattern similar to that of VOD). First, we confirmed that the fit of VOD vs  $f_d$  shown in Fig. 4 was robust over all 30-min data (Fig. 11a). Because of the close relationship between VOD and  $f_d$ , the scatter plots in Fig. 11 and Fig. 12 show similar patterns and descriptions with either VOD and  $f_d$  are interchangeable. We note the following: (i) though there is a lot of scatter, the largest rainfall amounts occurred during low-wind periods (i.e., 415 21.5 m WS < 3 m s<sup>-1</sup>, see Fig. 12a), (ii) as one would expect, there was a tendency for higher VOD values (smaller  $f_d$ ) with larger precipitation amounts (Fig. 11b, Fig. 12b), and (iii) drier air (i.e., larger VPD) led to smaller VOD (larger  $f_d$ ) values (Fig. 11c, Fig. 12c). The relationship between VOD and horizontal wind speed and friction velocity  $(u_*)_z$  was less clear (Figs. 11d,e,f), but there was a slight tendency for lower  $(u_*)_z$  to result in higher values of VOD; this relationship is confounded 420 by the tendency for higher precipitation to occur at lower wind speeds (Fig. 12a). The lack of a relationship between  $f_d$  and wind speed is especially apparent at moderate to higher WS values in Fig. 12d (e.g., 21.5 m WS > 2.5 m s<sup>-1</sup>), and consistent with the mechanical theory discussed in Sect. 2.2.2 and shown in Eq. 2. This supports our interpretation that variations in the  $f_d$  time series are primarily influenced by canopy interception and subsequent evaporation. Finally, it should be noted that some of the short-term variability in VOD is inherent to the uneven and constantly varying distribution of satellites and orbits through the canopy. This noise is only partially addressed by the data-processing algorithms. 425

Figure 9. The 2-day time series of the 17 wDry days from the warm-season of 2022 and 2023 for (a) detrended tree sway frequency  $f_d$ , (b) vegetation optical depth VOD, and (c) precipitation. The time series of 30-min samples are centered on the start of the wDry day and include the time series from the wet day that precedes the wDry day. The color of each line is shown by year and day of year (DOY) of the wDry day to the right of the panels. The periods from 2022 are shown as dashed lines. The time series with the minimum and maximum wet day total precipitation are in bold and highlighted by the text "(Min)" and "(Max)" in the right-hand list (see Table 3 for details). In (a) and (b), the composite wDry diel cycle from Fig. 6 is shown by the black line with filled circles.

**Figure 10.** As in Fig. 9, but with time shifted so the elapsed time is from the end of the precipitation event. In the Post-Precip side of the plot, the green line with open circles is the composite mean of the individual shifted time series, while the line with filled black circles are the mean values versus time-of-day as shown in Fig. 9a, b.

Figure 11. Scatter plots between vegetation optical depth VOD and (a) detrended tree sway frequency  $f_d$ , (b) the total precipitation amount from the wet day preceding the wDry day, (c) vapor pressure deficit VPD, (d) above-canopy mean horizontal wind speed WS, (e) above-canopy friction velocity  $(u_*)_z$ , and (f) subcanopy  $(u_*)_z$ . The solid black points are the 30-min mean values from the entire warm season period of our study, the red points are the mean values calculated between 4 and 8 hours after precipitation ended (see Fig. 10). In (a), the green line is the linear fit from Fig. 4. In (d), data with  $(u_*)_z > 0.2 \text{ m s}^{-1}$  are not shown to highlight the lower  $(u_*)_z$  results (this removes 1.8% of the data in (d)). Also, that there are fewer subcanopy  $(u_*)_z$  points because of missing data due to precipitation-effects on the sonic anemometer.

Figure 12. Similar to Fig. 11, but using detrended tree sway frequency  $f_d$  rather than VOD (we multiplied  $f_d$  by -1 to create patterns consistent with VOD shown in Fig. 11). In (a), the relationship between above-canopy mean horizontal wind speed WS and total precipitation amount from the wet day preceding the wDry day is shown. The solid black points are the 30-min mean values from the warm season periods for years 2016 to 2023, the red points are the mean values calculated between 4 and 8 hours after precipitation ended. See Fig. 11 for additional details.

# 3.6 Final thoughts





We found that the timing of evaporation of canopy-intercepted rainwater in a subalpine forest was consistently measured by two very different techniques, GNSS-based VOD and accelerometer-based tree sway frequency. Neither technique directly measures evaporation, but are proxy variables that are affected by the amount of liquid water within the canopy. It is encouraging to see good agreement between these techniques, especially considering that tree sway measurements were from a single tree, whereas the VOD footprint area was around 2,000 to 3,000 m<sup>2</sup>. In contrast, eddy-covariance ET measures the amount of vertically-transported water vapor, but it is subject to theoretical and practical challenges, especially in the subcanopy (e.g., Wolf et al., 2024). A few of these challenges are: (i) separating the total water vapor (ET) flux into the component fluxes of transpiration and canopy/ground evaporation, (ii) decoupling between the true surface flux and the flux measured by the eddy-covariance sensor (e.g., Thomas et al., 2013), (iii) taking into account non-steady flow and horizontal water-vapor transport (which could be taking place below our 2.5 m subcanopy sensor), (iv) eddy-covariance instruments do not work properly during active rainfall, and (v) footprint issues that are difficult to evaluate. Most of these concerns do not affect the VOD and tree sway measurements, which make them appealing to combine with eddy-covariance ET.

We end this section with a few limitations to our study and recommendations for future studies:

- 1. The VOD results are based on only 17 wDry days, while the ET results are based on 176 wDry days. So, there are significant statistical differences (and time periods covered) between these two data sets. If we had a multi-year record of VOD data that matched ET, then we could better examine other effects on the canopy evaporation, such as the dryness of the air or the rainfall amounts and intensity.
  - 2. We did not explicitly consider water flowing down the stems of the trees in our study. In general, this is a small term in the forest water balance (e.g., Carlyle-Moses et al., 2014). Nor did we consider fog/dew as a source of intercepted water.
    - 3. It would be useful to have a leaf-wetness sensor that mimics the actual water-holding capacity of the needles in an evergreen tree (and place them at different heights within the canopy airspace). There has been recent progress in creating more realistic needle-like structures (e.g., Wan et al., 2019; Li et al., 2022); to the best of our knowledge such sensors have yet to be deployed in real-world/field conditions.
- 4. The evaporation rate in a forest has large spatial variability. For example, the upper canopy typically dries out much faster than the middle and lower canopy due to higher exposure to radiation and wind. We could not consider such fine-scale features in our study.
  - 5. Tree sway motion was measured with a single accelerometer connected to a single tree. Without too much additional effort, the number of accelerometer sensors could be scaled-up to sample more trees (e.g., Granucci et al., 2013). The tree-sway technique also requires a minimal wind speed value to initiate tree motion. Video can also be used to simultaneously measure tree sway frequency from many trees within a forest (e.g., Ammatelli et al., 2025).

- 6. Our study used a single pair of GNSS receivers; it would be relatively easy to deploy additional GNSS ground receivers to get a better idea of VOD spatial variability. Perhaps having one receiver located mid-canopy could separate out the upper and lower-canopy interception amounts.
- 7. Our results are specific for coniferous forests which are more efficient at retaining tree-surface water than deciduous trees (e.g., Xiao and McPherson, 2016; Pflug et al., 2021); canopy evaporation from other forest types (such as broadleaf forests) could behave quite differently (e.g., Levia et al., 2019).
  - 8. If the VOD and tree sway data can be accurately related to the amount of liquid water within a forest canopy, such information could improve land-surface models (e.g., Lundquist et al., 2021) and help validate satellite-based canopy evaporation measurements (e.g., Stoy et al., 2025; Ranjbar et al., 2025).

### 4 Conclusions





We used the 2022/2023 warm-season observations from the Niwot Ridge Forest US-NR1 AmeriFlux site to examine how vegetation optical depth VOD (measured with a pair of GNSS receivers) and tree sway frequency (measured with a 3-axis accelerometer) behaved on a day-following-rain (a "wDry" day). Based on 17 wDry days, VOD and tree sway frequency correlated very well with each other and showed a nearly linear change with time from the start of a wDry day till around 14:00 local standard time (Fig. 6). We postulate that the near-linear change with time was due to a steady rate of intercepted-rainwater evaporation from the forest canopy. Over this 14-hour period, there was a substantial mean decrease in VOD from about 0.55 to 0.42 (Fig. 6d). After 14:00 LST, the rate of change of VOD and tree sway (versus time) decreased by around a factor of 5, suggesting that the evaporation of canopy-intercepted rainwater was complete (or greatly diminished). The variation in VOD and sway frequency between the 17 different wDry days was large (Fig. 9), which we attributed to differing precipitation amounts, mean wind speed and turbulence, and atmospheric humidity conditions. While both VOD and tree sway were strongly affected by liquid water on the vegetation *surfaces*, only tree sway seemed to be affected by transpiration-depleted internal tree water on the afternoon of a dry day. How internal and external water affect VOD has been discussed (without conclusion) by Holtzman et al. (2021). Our results suggest that satellite-based VOD measurements during wet periods should be interpreted with caution because, for our particular site, surface water on vegetation dominated the VOD signal compared to internal changes in tree water content. This is especially true when other changes, such as forest biomass differences or freeze/thaw transitions, are minimal.

As an independent, quantitative check, above-canopy and subcanopy eddy covariance evapotranspiration ET measurements were compared with the VOD and tree sway results. To the best of our knowledge, this is the first time VOD, tree sway and eddy-covariance measurements have been used together at a single site. Similar to VOD and tree sway, the ET measurements showed the cessation of canopy evaporation in the late afternoon following a wet day. The ET results suggest that the evaporation rate of canopy-intercepted rainwater from the US-NR1 subalpine forest was on the order of  $0.02 \text{ mm h}^{-1}$  at night and  $0.08 \text{ mm h}^{-1}$  at mid-day. This canopy evaporation rate is on the low side compared with observations from other studies (Ta-

ble 2), but there are are several challenges with the eddy-covariance ET observations which are discussed in Sect. 3.6. During the morning transition, the VOD and tree sway measurements both suggest that increased net radiation (at sunrise) had a small effect on the rate of canopy evaporation; however, the ET results showed an increase in canopy evaporation by over a factor of 3 during the sunrise period (Fig. 6). Friction velocity measurements within the canopy layer suggest that turbulence could be a more important variable than net radiation in controlling the canopy evaporation (Fig. 7).

For the modeling component in our study, we found that the Community Land Model CLM4.5 canopy surface water content increased appropriately as rain fell (Fig. 8). However, our results suggest that evaporation of the CLM canopy surface water content during a wDry day mimicked the evaporation from a flat-plate wetness sensor near canopy-top. Both the flat-plate sensor and CLM4.5 surface water content showed a rapid drying of the canopy following sunrise presumably due to increased net radiation. As noted above, the VOD and tree sway measurements were *insensitive* to changes in net radiation at sunrise; we suspect that the lack of sensitivity of VOD and tree sway to net radiation is due to the ability of the clumped needles within the US-NR1 subalpine forest to retain water droplets (relative to a flat-plate wetness sensor). We suggest that future studies of canopy evaporation focus attention on the sunrise period to determine if there are clear signals of increased canopy evaporation during this critical time of day.

Overall, we conclude that VOD from a pair of GNSS receivers and tree sway are both capable of measuring warm-season canopy interception (as well as the canopy-rainwater-retention time following rainfall). The VOD footprint is around 95% smaller and qualitatively different than the ET flux footprint which depends on wind direction and atmospheric conditions. The analysis techniques we have presented herein could be applied to other forest ecosystems and used to inform and improve big-leaf type land-surface models, such as CLM, as well as multi-layer canopy models which are becoming more prevalent within the scientific community (e.g., Bonan et al., 2021; Jiang et al., 2025).

Code and data availability. Data used in the study are available from Burns et al. (2025b) and/or the Assets tab on the paper website;

Data-processing software are available from the Assets tab on the paper website.

# **Appendix A: Additional Measurement Information**

### A1 Water vapor flux/evapotranspiration






Ecosystem eddy covariance flux measurements at the US-NR1 site started in fall of 1998 (e.g., Monson et al., 2002; Burns et al., 2015). Here we focus on the eddy-covariance instrumentation used in 2022–2023 (i.e., during the period of VOD data collection). The sensible heat and CO<sub>2</sub> flux measurements categorized by the precipitation state have been presented in earlier studies (e.g., Burns et al., 2015, 2018). As a departure from these earlier studies, the latent heat flux measurements are represented with units of water vapor flux or ET [mm of H<sub>2</sub>O per hour] though our plots also include the energy units [W m<sup>-2</sup>] to ease comparison with previous work. In our study, "evapotranspiration" includes transpiration from trees as well as the evaporation of liquid water from the canopy surfaces and the ground (Miralles et al., 2020).

The primary above-canopy water vapor flux instrument was a closed-path infrared gas analyzer IRGA (LI-COR, model LI-7200) at 21.5 m height, co-located with a Campbell Scientific CSAT3 sonic anemometer (Table 1). In the subcanopy, water vapor fluctuations were measured at 2.5 m height with an open-path IRGA (LI-COR, model LI-7500A) and co-located CSAT3 on a 6-m subcanopy flux tower about 15 m south/southwest of the 25-m tower. The eddy covariance fluxes were calculated using standard methods (e.g., Aubinet et al., 2012). An open-path IRGA has the so-called Burba or self-heating correction (Burba et al., 2008; Frank and Massman, 2020); however, for latent heat flux this is a factor of 100 smaller than for CO<sub>2</sub> flux, and is only significant if the latent heat flux is accumulated over a long period (Reverter et al., 2010). Therefore, we did not apply any self-heating correction to subcanopy ET in our study.

Precipitation disrupts the eddy-covariance flux measurements when liquid water accumulates on open-path IRGA optical lenses and sonic transducers (Sect. 2.4). This is especially challenging for open-path IRGAs, such as the one located in the subcanopy. Rather than attempt to gap-fill these missing data, we have excluded them from our study. These missing data due to precipitation primarily occurred during the dWet and wWet days not the wDry and dDry days, which are the focus of our study.

### A2 Turbulence



In order to evaluate turbulence at different heights within the canopy, we calculated a *local* friction velocity as  $(u_*)_z = ((\overline{u'w'})_z^2 + (\overline{v'w'})_z^2)^{0.25}$  where u', v', and w' are the planar-fit wind fluctuations in the streamwise, crosswind, and vertical directions, respectively. Above-canopy (21.5 m)  $(u_*)_z$  is close to the true friction velocity  $u_*$  value (which, theoretically, is constant above the canopy).

### A3 Net radiation

Above-canopy net radiation  $R_{net}$  was measured at 25 m height on the US-NR1 main tower with a 4-component radiometer (Kipp and Zonen, model CNR1). The radiation data are primarily used to show when clouds were present as well as how the radiation conditions on wDry days compare to those on dDry days.

### A4 Surface wetness

The Campbell Scientific model 237 leaf wetness sensor (Campbell Scientific, Inc., 2021) is a horizontally-oriented resistive-grid that measures the presence of liquid water. It is described as a "leaf wetness" sensor, as has been previously reported (e.g., Burns et al., 2015). However, a flat, hard plastic plate does not replicate the energy balance in clumped conifer needles or the ability of those wet needles to resist evaporation; after examining the data in our study (Sect. 3.2) and re-reading the caveats within the sensor manual, we have concluded that "surface wetness" (or simply "wetness") is a more appropriate sensor description. The inability to mimic clumped needles is further exacerbated by the 13.5 m location of the sensor on the main tower, near the upper part of the forest canopy. The output from the sensor has been normalized so that a value of

zero corresponds to dry conditions while a value of one corresponds to completely wet conditions. Values between 0 and 1 correspond to "slightly wet" conditions.

# A5 Tree water content






In the summer of 2017, GS3 water content sensors (Decagon Devices, Inc., 2016; METER Group, Inc., 2019) were installed at breast-height into five different tree boles (2 pine, 2 spruce and 1 fir tree) at the US-NR1 site. These five trees are located about 20-30 m southeast of the main tower (Fig. 1). Holes were drilled in a vertical pattern to allow the three sensor needles to snugly fit into the trunk and then the entire sensor was sealed in place using silicone sealant.

The GS3 sensor uses a 70 MHz oscillating wave (sensed by the probe needles) to measure the dielectric permittivity of the material surrounding the sensor. The GS3 sensor is designed to measure water content in a variety of materials; though they are primarily used in soils (Hilhorst, 2000), they can be adapted to work in soilless media such as manure (Sutitarnnontr et al., 2014) or to monitor the internal water content of living trees (Hao et al., 2013). The GS3 probe outputs three variables: temperature [°C], electrical conductivity [dS m<sup>-1</sup>], and apparent dielectric permittivity  $\varepsilon_a$  [dimensionless]. The range of  $\varepsilon_a$  is between 1 (for air) to 80 (for water). Dielectric permittivity is a measure of how well a material can hold an electrical charge and depends on the wave frequency; liquid water has a high value of  $\varepsilon_a$  because of the polar nature of water molecules (Blanken, 2024). For a 70 MHz signal, the value of  $\varepsilon_a$  of ice is around 3 whereas that of liquid water is closer to 80 (Fletcher, 1970). An example of decreased  $\varepsilon_a$  due to frozen water within the tree bole is shown in Fig. 2d (at around DOY 298).

From the GS3 sensor, apparent dielectric permittivity can be related to volumetric water content (VWC) of the media the GS3 is within by the following expression,

$$VWC = \left(A(\varepsilon_a)^{0.5} - B\right)^C,\tag{A1}$$

where A, B, and C are empirical constants that depend on the media type. However, we did not calibrate our GS3 sensors for the boles they were inserted into. For our particular purpose, it is only important that changes in VWC are proportionally-related to changes in  $\varepsilon_a$ ; therefore, we only consider relative differences in  $\varepsilon_a$  over the diel cycle (or with precipitation state).

Of the five GS3 sensors deployed at US-NR1, the one in the fir tree did not work and will not be used. The other four sensors showed occasional sharp jumps in the dielectric permittivity. Examples of the warm-season time series for years 2018–2022 are shown in in Figs. S14–S18. Some conclusions from these time series: (i) the jumps only occurred during the warm-season, (ii) when one sensor jumped, most of the other ones did too (but not always in the same direction), and (iii) the occurrence of the jumps has been more frequent in recent years (and we have not used the 2023 data because correcting the jumps was too difficult). Though we could not determine the exact cause/reason for the jumps, we deemed them to be nonphysical in nature and likely related to lightning or changes to the power supply. Because we are interested in short-term changes in  $\varepsilon_a$  due to precipitation, we removed the jumps as shown in the lower panels of Figs. S14–S18, prior to the diel cycle analysis. For the purposes of our study the tree water content shows a maximum in the early morning and minimum in the afternoon (Fig. S8c).

This result is consistent with other studies in subalpine forests (Oberleitner et al., 2022), and is caused by transpiration reducing internally-stored water within the tree tissue (Landsberg and Waring, 2017).

# A6 Lidar





Airborne scanning lidar measurements were collected in 2010 with an Optech Gemini Airborne Laser Terrain Mapper deployed on a Piper Chieftan flying at 600 m above the surface over the entire Boulder Creek watershed (Anderson et al., 2012; Broxton et al., 2015). By taking the difference between the ground elevation surface and the lidar cloud data, the trees and gaps in the forest near the US-NR1 main tower can be observed (Burns, 2018). The x- and y-axes in Figure 1 are distances from the US-NR1 main tower; the method of (Li et al., 2012) has been used to identify trees locations where a cluster of trees taller than 12 m just to the south of the gnssB location can be seen.

# 590 Appendix B: Footprint of VOD and water vapor fluxes

The VOD footprint calculation follows Humphrey and Frankenberg (2023), where the footprint radius r was calculated with,

$$r = \frac{h - z_{\text{gnssB}}}{\tan(\theta_{el})}.$$
(B1)

If we assume a canopy height of  $h=13\,\mathrm{m}$ ,  $z_{\mathrm{gnssB}}=1.27\,\mathrm{m}$ , and an elevation angle cutoff  $\theta_{el}=10^\circ$ , the resulting footprint has  $r\approx 66\,\mathrm{m}$ . This footprint would include trees far from gnssB where only the very top of the tree contributes to the VOD measurement (a nearly negligible contribution). A more realistic footprint is found by considering where at least half of the tree is contributing the VOD calculation which is:

$$r = \frac{h/2 - z_{\text{gnssB}}}{\tan(\theta_{el})}.$$
(B2)

Now, we get a value of  $r \approx 30 \,\mathrm{m}$  which is shown as the outer circle in Fig. 1. If we closely compare the tree locations shown in Fig. 1 with the skyplot for the VOD calculation (Fig. S3), we can see that several tall trees southwest of gnssB are making a large contribution to the VOD; we show this in Fig. 1 as a footprint with  $r = 20 \,\mathrm{m}$  where we have removed part of the footprint to the north of gnssB due to the paucity of GNSS orbits in the northern sky. This second line is also to emphasize that the VOD footprint is not a fixed location within the forest, but depends on individual tree heights and the distance from the gnssB antenna. Overall, we estimate the VOD footprint at US-NR1 to be around 2,000 to 3,000 m<sup>2</sup>. The schematic in Fig. S4 provides a picture of how the footprint varies with elevation angle and tree height from the gnssB antenna. As part of manuscript discussion (i.e., Burns et al., 2025a), we confirmed that using  $\theta_{el} = 30^{\circ}$  did not significantly affect the resulting VOD.

In contrast, the above-canopy eddy-covariance fluxes measured at US-NR1 have a flux footprint climatology of around 50,000 m<sup>2</sup>, depending on the wind direction and atmospheric stability (e.g., Chu et al., 2021). A more explicit view of the US-NR1 flux footprint climatology was calculated using the Kljun et al. (2015) model for five different atmospheric stability

- conditions for winds from the west (Fig. S12) and for winds from the east (Fig. S13). The footprint analysis is separated into east and west wind directions because winds at the site are typically either upslope (from east) or downslope (from west) (Burns et al., 2011). For comparison purposes, we include the  $r \approx 30 \, \text{m}$  VOD footprint from Fig. 1 in Figs. S12 and S13 and the legend in each figure lists the frequency of occurrence for each stability condition. For downslope winds with strongly stable conditions, the flux footprint size triples and is on the order of  $140,000 \, \text{m}^2$  (Fig. S12).
- Author contributions. SpB and PB conceived the VOD-portion of the project after seeing a presentation by VH about GNSS-based VOD. SpB and PB obtained the GNSS hardware from UNAVCO and collected the GNSS data. VH processed the GNSS data and calculated VOD. MS and ED provided tree sway motion hardware and setup the tree sway measurements. MS processed the tree sway motion data. DB provided GS3 sensors and setup the tree bole water content measurements. All authors contributed to writing, discussing, and editing the manuscript text.
- Competing interests. The lead author has declared that none of the authors have any competing interests.

Acknowledgements. We thank Kristine Larson (emeritus, University of Colorado) for advice in the early stages of our study and the Earth-Scope Constortium (formerly, UNAVCO) for providing the GNSS hardware; Jim Normandeau was especially responsive to our requests. We also gratefully acknowledge the organizers of the 2021 AmeriFlux Evapotranspiration Workshop (Koong Yi, Kyle Delwiche, Jacob Nelson, and Trevor Keenan) without which this work would not have occurred. The comments from four anonymous reviewers greatly improved the manuscript. The US-NR1 AmeriFlux site has been supported as a core site by the U.S. DOE, Office of Science through the AmeriFlux Management Project (AMP) at Lawrence Berkeley National Laboratory under Award Number 7094866.

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
