# Peer review of "Using GNSS-based vegetation optical depth, tree sway motion, and eddy-covariance to examine evaporation of canopy-intercepted rainfall in a subalpine forest"

_EGUsphere, 2025_

## Referee Comment (RC2)

This manuscript presents the process of canopy evaporation in a subalpine forest at the Niwot Ridge US-NR1 AmeriFlux site. The authors employ a combination of independent measurements that includes GNSS-based VOD, tree sway frequency, and eddy-covariance evapotranspiration, to understand the dynamics of water stored in the forest canopy, particularly following rainfall events. The observations are also compared with results from the Community Land Model CLM4.5. Overall, the paper presents a unique multi-sensor approach and offer valuable insights into canopy water dynamics. Although the research is insightful, I have a few comments that are detailed hereafter which should be addressed before publication.

Major comments:

- The manuscript would benefit from improved readability by restructuring the Introduction section. Some parts included in the Introduction seem more appropriate for the Materials and Methods section.
- GNSS L-band is used to retrieve VOD. Given that L-band is sensitive to water content in woody components (e.g., stems and branches), it would be helpful if the authors could provide a clearer justification or discussion regarding this aspect in the context of their interpretation.
- Page 5, Line 139: Authors mention "Linear interpolation over time was used to convert the hourly data to a 30-min time series". What is the rationale for linearly interpolating hourly data to 30-minute resolution, instead of conducting the analysis directly at the native 1-hour interval? Does this interpolation introduce any artifacts?
- Please clarify the climate classification of the study site.
- Tu & Yang, 2022, Hua et al., 2020 discuss the overestimation of PET/ET particularly in arid and semi-arid environments while Sun et al., 2016 discusses underestimation of ET in cold areas using traditional methods. Could the authors clarify whether ET overestimation/underestimation is relevant for this ecosystem, what are the implications and uncertainties?
- A brief explanation of the detrending process for tree sway frequency in the main text would be beneficial, even if detailed in Raleigh et al. (2022). Further, please clarify relevance of detrending tree sway alone and not others.
- The study relies on 17 wDry days for the VOD and tree sway, while the ET results for wDry days (in Figure 6a,6b) are based on a significantly larger sample size of 176 days from a longer period (2004–2022 vs. 2022–2023 for VOD/sway). To strengthen the paper to explicitly discuss how this disparity might influence the comparison, perhaps a composite of ET from only the 17 wDry days used for VOD/sway could be presented in the supplementary information for a more direct comparison.

Minor comments:

- Page 1, Line 1: Is Interception only due to warm-season precipitation?

- Page 1, Line 9: Can the authors elaborate/clarify on this "changes in internal tree-water content than VOD."?
- First paragraph of introduction section is detailed/explained well but not sufficiently referenced.
- Page 2, Line 22: Authors mention "Evapotranspiration ET is the sum of transpiration with soil and canopy evaporation" this is confusing; but ET = soil evaporation + canopy evaporation + transpiration?

- Throughout the manuscript, there are instances where terms like 'evapotranspiration' are written in full repeatedly after being defined by their abbreviation (e.g., ET). I recommend using the respective abbreviations consistently after the first mention to improve readability and maintain consistency.
- Page 10, Line 210: Authors mention "To ease comparison with other studies" and no references were cited here. I recommend including relevant studies to support the statement.
- Table 1 indicates VOD measurements began in June 2022, but the study period warm season is defined starting in Sept 2022. Please clarify on this.
- Please mention the temporal sampling interval for each observation type and ensure consistency throughout the manuscript.

---

## Author Comment (AC1)

**Reply to Referee #1**

**S. P. Burns et al.**

sean@ucar.edu

Date: May 16, 2025

The comments by Referee 1 are greatly appreciated. We have listed the comments by Referee 1 below in italics, followed by our responses.

*Burns et al. measured warm season ET, VOD and tree sway in Colorado subalpine forest to was to track evaporation of intercepted water. They found that a canopy intercepted water can take a tremendous amount of time to evaporate and thus can contribute to ET at the site especially on dry days after a rain event. This work well describes VOD and tree sway as a method of measuring evaporation of canopy interception.*

This is an accurate summary of our manuscript and the goals of the study.

**Under the category "Comments":**

*To help distinguish these VOD measurements from more common measurements excluding, I suggest to specify in the acronym that this is VOD plus intercepted water. For example, VOD underscore "int" to distinguish from studies where VOD refers to water only within the tissues.*

Though we appreciate the concept of making this distinction, the practical logistics of applying it are a bit difficult, and (in our opinion) could lead to confusion. For example, there are several plots that show VOD in both dry and wet conditions (e.g., Figs. 2 and 3) as well as plots that show VOD in only dry conditions (Fig. 5). So, it would be inaccurate to label the VOD in dry conditions as $VOD_{int}$. We think it is clear that the focus of our study is on how VOD is affected by wet conditions and leave it to the reader to distinguish when the forest surfaces are wet and/or dry.

*line 66: Please clarify what the "small pockets" are referring to.*

The "small pockets" were referring to the tracheids within the xylem that move the water in a conifer tree. In an attempt to clarify this text, we changed the words "small pockets" to "tracheids within the xylem that are on the order of 5 to 80 $\mu$m in diameter and less than 5 mm long". We also added a new reference to Hacke et al. (2015) which contains this information. The sentence has been re-written as,

> "This is because intercepted water forms a water layer which attenuates and reflects microwave signals more than internal water found within the xylem tubular structures; for conifers, tree water is primarily found within xylem tracheids that are on the order of 5 to 80 $\mu$m in diameter and less than 5 mm long (e.g., Tyree and Ewers, 1991; Hacke et al., 2015)."

*line 75: Parentheses missing around GNSS*

The parentheses have been added.

*line 95: The inclusion of the CLM analysis while exciting comes up abruptly here and the importance of this analysis could be integrated sooner. Suggest to reference Burns et al 2018 or specific the concepts from that paper that will be expanded upon here.*

This is a good point. We added some context for the Burns et al. (2018) study. The original paragraph was:

> "Though the focus of our study is on observations, we also wanted to compare the VOD and tree sway observations with land-surface model results. To achieve this, we used the Community Land Model CLM4.5 (e.g., Oleson et al., 2013). This expands on previous work with CLM4.5 at US-NR1 by Burns et al. (2018) to include modeled canopy surface water content (which we expect to be comparable with the VOD and tree sway frequency measurements)."

The revised paragraph is:

> "Though the focus of our study is on observations, we also wanted to compare the VOD and tree sway observations with land-surface model results. In Burns et al. (2018) the Community Land Model CLM4.5 (Oleson et al., 2013) components of the latent heat flux (transpiration, canopy evaporation, and soil evaporation) were compared with the measured ecosystem-scale latent heat flux. Since the VOD and tree sway frequency measurements are related to canopy water content (and thus canopy evaporation), we compared these observations to CLM4.5-modeled canopy surface water content."

We will also consider moving the location of the CLM4.5 comparison within the Introduction.

*line 163: Suggest to replace "use the concept" with "assume"*

We agree and changed "use the concept" to "assume".

*Figure 3, 8: It would be helpful to please clarify in the results why there is not diurnal cycle of VOD where VOD peaks in the morning and decreases into the afternoon even on the dry days. This is common finding in Holtzman et al. 2021 Biogeosciences and Yao et al. 2024 Geophysical Research Letters. Later on VOD does increase due to rain but do the trees not dehydrate when transpiring through the day?*

The finding that VOD had no diel cycle in dDry conditions was a surprise. This is highlighted in Fig. 5 which is only data from the dDry days (so there should be minimal effect of precipitation). In Figs. 5c and 5e both the tree sway frequency data and tree bole moisture sensors show a diel cycle that suggests transpiration effects on the tree water content. Clearly VOD (in Fig. 5d) does not have a diel cycle in these conditions. We address this with the following text at the end of Sect. 3.2:

"There is an important implication from the dDry periods (when most of the ET is transpiration, not evaporation). Though $\varepsilon_a$ had a dDry diel cycle with a clear early-morning maximum and late afternoon minimum, VOD had relatively small variation without any apparent pattern (Fig. 5). This suggests that, at our site, VOD changes were largely controlled by water on the canopy *surfaces*, not the the internal water content of the trees. In contrast, tree sway frequency had a diel pattern on dDry days (Fig. 5c) that was similar to that of $\varepsilon_a$ (Fig. 5e), suggesting that tree sway frequency was more affected by internal tree water content changes than VOD. Ciruzzi and Loheide II (2019) has also shown that tree sway motion is affected by changes in internal tree water content. Other possible factors could be an effect of the temperature diel cycle on tree sway frequency, or too much noise in the VOD measurements to capture a diel cycle in internal tree water content."

In the last sentence we report a few possible reasons that we did not find a clear diel signal in the VOD data during dDry conditions. However, taking a closer look at the diel cycle in Holtzman et al. (2021) (their Fig. 4) and Yao et al. (2024) (their Fig. 2) we note that have diel VOD ranges of 0.85 to 1.1 and 0.62 to 0.7, respectively. Both of these studies are from deciduous forests in the eastern USA. At the US-NR1 subalpine forest, our diel range in VOD for dry conditions is on the order of 0.36 to 0.38 (see Fig. 5d). Therefore, another possible reason for the lack of diel cycle in VOD in dry conditions is the lower water content of the subalpine forest trees compared to deciduous trees in a more humid/wet environment. We will add a note with the ranges of the Holtzman/Yao diel cycle somewhere within our manuscript as a possible reason for the lack of a VOD diel cycle in dry conditions. We welcome any additional comments/ideas by reviewer 1 related to this.

*336-337: Please describe the B0, G1, A1, and F2 cases here or in the methods sections for reference to the reader.*

The meaning of B0, G1, A1, and F2 are described at the end of Sect. 2.3. On line 336-337, we have added a reference back to Sect 2.3 so this information can be easily found.

*419: This is does not seem like a novel finding given your reference to the common method of removing these data in the introduction. Please clarify the novelty here.*

We agree that our study builds on previous suggestions (e.g., Holtzman et al., 2021; Yao et al., 2024) that time periods following precipitation affect VOD. However, in Holtzman et al. (2021), they suggested that the canopy wetness did *not* affect VOD (i.e., Sect. 3.3 in Holtzman, et al. is entitled, "**Canopy interception fails to influence VOD**"). And, in Section 4.4, they write,

"More research is needed to better understand how VOD sensitivity varies between water internal and external to the canopy.".

So, Holtzman et al. (2021) did not make a conclusive statement on this issue. Yao et al. (2024) showed how wet periods affected the canopy (i.e., their Fig. 2c), which they used this as evidence to avoid wet periods. In contrast, we are stating these "wet" periods are valuable for looking at canopy evaporation and specify how long it takes for the canopy to dry out (which is the novel information added by our study). We agree that these subtle points in Holtzman and Yao should be clarified in our manuscript. If there are other studies that have specifically looked at this issue that we have not included within our manuscript, please let us know about them. We will modify our text to better highlight this point.

*477: suggest to replace leaf with needles for clarity.*

Good point. We changed "leaf" to "clumped needles".

**References**

Burns, S. P., Swenson, S. C., Wieder, W. R., Lawrence, D. M., Bonan, G. B., Knowles, J. F., and Blanken, P. D.: A comparison of the diel cycle of modeled and measured latent heat flux during the warm season in a Colorado subalpine forest, J. Adv. Model Earth Sys., 10, 617–651, doi:10.1002/2017MS001248, 2018.

Ciruzzi, D. M. and Loheide II, S. P.: Monitoring tree sway as an indicator of water stress, Geophysical Research Letters, 46, 12 021–12 029, doi:10.1029/2019GL084122, 2019.

Hacke, U. G., Lachenbruch, B., Pittermann, J., Mayr, S., Domec, J., and Schulte, P. J.: The hydraulic architecture of conifers, in: Functional and Ecological Xylem Anatomy, edited by Hacke, U., Springer International Publishing, Cham, Switzerland, pp. 39–75, doi:10.1007/978-3-319-15783-2_2, 2015.

Holtzman, N. M., Anderegg, L. D. L., Kraatz, S., Mavrovic, A., Sonnentag, O., Pappas, C., Cosh, M. H., Langlois, A., Lakhankar, T., Tesser, D., Steiner, N., Colliander, A., Roy, A., and Konings, A. G.: L-band vegetation optical depth as an indicator of plant water potential in a temperate deciduous forest stand, Biogeosciences, 18, 739–753, doi:10.5194/bg-18-739-2021, 2021.

McElrone, A. J., Choat, B., Gambetta, G. A., and Brodersen, C. R.: Water uptake and transport in vascular plants, Nature Education Knowledge, 4, https://www.nature.com/scitable/knowledge/library/water-uptake-and-transport-in-vascular-plants-103016037/, 2013.

Oleson, K. W., Lawrence, D. M., Bonan, G. B., Drewniak, B., Huang, M., Koven, C. D., Levis, S., Li, F., Riley, W. J., Subin, Z. M., Swenson, S., Thornton, P. E., Bozbiyik, A., Fisher, R., Heald, C. L., Kluzek, E., Lamarque, J.-F., Lawrence, P. J., Leung, L. R., Lipscomb, W. Muszala, S. P., Ricciuto, D. M., Sacks, W. J., Sun, Y., Tang, J., and Yang, Z.-L.: Technical description of version 4.5 of the Community Land Model (CLM), Tech. Rep. NCAR/TN-503+STR, NCAR Technical Note, doi:10.5065/D6RR1W7M, 420 pp., 2013.

Tyree, M. T. and Ewers, F. W.: The hydraulic architecture of trees and other woody-plants, New Phytologist, 119, 345–360, doi:10.1111/j.1469-8137.1991.tb00035.x, 1991.

Yao, Y., Humphrey, V., Konings, A. G., Wang, Y., Yin, Y., Holtzman, N., Wood, J. D., Bar-On, Y., and Frankenberg, C.: Investigating diurnal and seasonal cycles of vegetation optical depth retrieved from GNSS signals in a broadleaf forest, Geophysical Research Letters, 51, e2023GL107 121, doi:10.1029/2023GL107121, 2024.

---

## Author Comment (AC2)

**Reply to Referee #2**

**S. P. Burns et al.**

sean@ucar.edu

Date: May 30, 2025

The comments by Referee 2 are greatly appreciated. We have listed the comments by Referee 2 below in italics, followed by our responses. We added numbers to each of the specific comments so it is easier to reference the comments by Referee 2.

*This manuscript presents the process of canopy evaporation in a subalpine forest at the Niwot Ridge US-NR1 AmeriFlux site. The authors employ a combination of independent measurements that includes GNSS-based VOD, tree sway frequency, and eddy-covariance evapotranspiration, to understand the dynamics of water stored in the forest canopy, particularly following rainfall events. The observations are also compared with results from the Community Land Model CLM4.5. Overall, the paper presents a unique multi-sensor approach and offer valuable insights into canopy water dynamics. Although the research is insightful, I have a few comments that are detailed hereafter which should be addressed before publication.*

This is an accurate summary of our manuscript and we appreciate the positive comments about the "valuable insights" our study provides. We address the specific comments below.

**Under the category "Major comments":**

1. *The manuscript would benefit from improved readability by restructuring the Introduction section. Some parts included in the Introduction seem more appropriate for the Materials and Methods section.*

We agree that readability in the Introduction could be improved. This comment would be more helpful if a specific example of what the reviewer is referring to was provided (though some context is given in comment #10 below; please see our reply to comment #10). In an attempt to improve the readability, we moved a paragraph (lines 83–88) to Sect. 2.1 in the Materials and Methods section. We agree that Materials and Methods a more appropriate location for this information. Between moving this text and addressing comment #10 (below) we think the readability of the Introduction has been improved; however, if this is not what you were referring to, please let us know.

2. *GNSS L-band is used to retrieve VOD. Given that L-band is sensitive to water content in woody components (e.g., stems and branches), it would be helpful if the authors could provide a clearer justification or discussion regarding this aspect in the context of their interpretation.*

One of our conclusions is that VOD at US-NR1 is not very sensitive to the internal tree water content. We agree that this could be expanded upon; and our response to a comment from Referee 1 contains additional details. To summarize that response: We noted that VOD at the US-NR1 site site (during dry days) is at around 0.35 which is much lower than VOD in other studies which are

[Figure]

Figure R1: A portion of the the VOD time series shown in Fig. 2b in the discussion manuscript. Here we have included both the raw/hourly and interpolated/30-min data (see legend). Statistics (number of samples $N$, mean, and standard deviation Std Dev) from both sets of VOD data for the period DOY 160-170 are included as a table in the upper-left hand part of the figure.

on the order of 0.6 to 1.1 (e.g., Holtzman et al., 2021; Humphrey and Frankenberg, 2023; Yao et al., 2024). Based on this we postulate that the internal water content of the US-NR1 subalpine trees are low compared to the trees in more humid locations and therefore VOD is not detecting this (lower) water content signal. It could also be that the US-NR1 forest is less dense. A study by Luo et al. (2020) has shown that tree water content varies significantly between different species. We will add additional discussion about this in the revised manuscript, and, at the very least, point out that the US-NR1 VOD in dry conditions is lower than VOD in the other studies (as mentioned above).

3. *Page 5, Line 139: Authors mention "Linear interpolation over time was used to convert the hourly data to a 30-min time series". What is the rationale for linearly interpolating hourly data to 30-minute resolution, instead of conducting the analysis directly at the native 1-hour interval? Does this interpolation introduce any artifacts?*

This is a good question. There are two primary motivations for interpolating the VOD data from hourly to 30-min time periods; (i) all the water vapor flux data and tree sway data are calculated over a 30-min period, and (ii) more clarity in the diel cycle is obtained with as high a sampling frequency (or as short an averaging period) as possible. Rather than downgrading the measurement period of the tree sway and flux data, we decided to linearly interpolate the VOD data.

To check how the linear interpolation might affect the VOD statistics, a 7-day time series of the raw/hourly and interpolated 30-min VOD data is shown in Fig. R1. This figure includes 10-day mean and standard deviation of the respective VOD data, and the differences in mean values (0.379 vs 0.378) are very small (less than 0.5%), whereas differences in the standard deviation (0.060 vs 0.058) are less than 3%.

[Figure]

Figure R2: As in Figure R1, but only showing the first 12 hours of DOY 260.

If we zoom in a bit more and only show half of a day, then we see that the effect of linear interpolation decreases the range of the measured VOD (Fig. R2). This happens because the 30-min periods are centered on 15 and 45 minutes past the hour, while the hourly data are centered on 30-min past the hour. The decrease in the range of the measurements is why the linearly-interpolated data have a slightly smaller standard deviation than the hourly measurements.

As far as we can tell, the linear interpolation does have have any significant impact on our results or conclusions. We further test this by repeating Fig. 3d in the manuscript, but showing both the diel cycle of the hourly VOD data along with the 30-min interpolated VOD data (Fig. R3). There are small differences, but nothing that would change our interpretation of the results.

4. *Please clarify the climate classification of the study site.*

We added additional information in the site description section which describes the US-NR1 climate. Our revised text is:

" The mean annual temperature at US-NR1 is around 2°C. According to the Köppen–Geiger climate classification system (Kottek et al., 2006) the site is type Dfc which corresponds to a cold, snowy/moist continental climate with precipitation spread fairly evenly throughout the year; the long-term mean annual precipitation at the site is around 800 mm and snow typically covers the ground from mid-November until late May (Burns et al., 2015)."

[Figure]

Figure R3: Similar to Fig. 3d in the manuscript, but showing the mean diel cycle for both the hourly and 30-min interpolated VOD data (see legend). See Fig. 3d within the manuscript for additional plot details.

5. *Tu & Yang, 2022, Hua et al., 2020 discuss the overestimation of PET/ET particularly in arid and semi-arid environments while Sun et al., 2016 discusses underestimation of ET in cold areas using traditional methods. Could the authors clarify whether ET overestimation/underestimation is relevant for this ecosystem, what are the implications and uncertainties?*

We considered the following papers brought up by the reviewer: Hua et al. (2020); Sun et al. (2016); Tu and Yang (2022). If any of these are incorrect, please provide the DOI, so we can make sure to find the correct paper.

These papers are primarily concerned with modeling ET. Sun et al. (2016) use modeled ET to look at effects on the air and land-surface temperature. Hua et al. (2020) uses Penman-Monteith to model ET and compare with surface met data and MODIS products. Tu and Yang (2022) focused on potential evaporation using a variety of models and compared these to ET from flux towers; however there are some open questions about net radiation assumptions within the chosen model (e.g., Szilagyi, 2022). While these are all reasonable/useful studies, we used a direct measurement of ET in our study. The studies could be relevant to the CLM4.5 portion of our study, but we examined how the model reacts to precipitation, not the metrics used in those studies. Perhaps these citations would be appropriate for a future study (i.e., that is trying to model absolute ET), but not the current one.

6. *A brief explanation of the detrending process for tree sway frequency in the main text would be beneficial, even if detailed in Raleigh et al. (2022). Further, please clarify relevance of detrending tree sway alone and not others.*

The low-frequency detrending of the tree sway frequency data is described on lines 155–157 and an example is shown in supplemental Fig. S4. The detrending uses a 10-day sliding median filter. The low-frequency detrending was not part of Raleigh et al. (2022). The tree sway data has an obvious low-frequency trend in the time series (i.e., Fig. 2c in the manuscript). As we have shown, the low-frequency detrending did not impact the results (see discussion on lines 245–247 and Fig. 4). None of the other variables showed an obvious trend with time so there was not a need to detrend any other variables (in fact, it was not really needed for tree sway frequency either).

**7.** *The study relies on 17 wDry days for the VOD and tree sway, while the ET results for wDry days (in Figure 6a,6b) are based on a significantly larger sample size of 176 days from a longer period (2004-2022 vs. 2022-2023 for VOD/sway). To strengthen the paper to explicitly discuss how this disparity might influence the comparison, perhaps a composite of ET from only the 17 wDry days used for VOD/sway could be presented in the supplementary information for a more direct comparison.*

It is correct that in Fig. 6a,b, the period of 2004 to 2022 is used for the ET diel cycle. This is discussed as a limitation of our study (lines 388–391) in the discussion manuscript. We note that a composite of the above-canopy ET for the 17 wDry days is shown in Fig. 3b. There are several reasons that a longer period is used for ET in Fig. 6: (i) the subcanopy flux system was not working correctly in Fall 2022 into summer of 2023, and (ii) a smooth ET diel cycle greatly benefits from having many samples/years of data. This can be seen by comparing the above-canopy ET shown in the wDry part of Fig. 3b with that in Fig. 6a. With only 17 ET periods the line in Fig. 3b is a bit jagged whereas the one in Fig. 6a is much smoother. To extract a smooth diel cycle is an even larger challenge in the subcanopy, where we use an open-path IRGA.

For these reasons (primarily item (i) above), we cannot do this suggestion by Referee 2.

**Under the category "Minor comments":**

**8.** *Page 1, Line 1: Is Interception only due to warm-season precipitation?*

Is "Page 1, Line 1" referring to the title which has "canopy-intercepted rainfall" in it? This comment is a bit unclear, but we think you are likely referring to the fact that fog/dew are also a source of intercepted water? If this understanding is correct, this is a good point and we have added text to our list of limitations in our study as,

"Nor did we consider possible effects of fog/dew as a source of intercepted water."

We mentioned that we think fog/dew is generally a small source of intercepted water at our site (see lines 68-69 in our discussion manuscript).

If we mis-understood this comment, clarification would be appreciated!

**9.** *Page 1, Line 9: Can the authors elaborate/clarify on this "changes in internal tree-water content than VOD."?*

This point is discussed at the end of Sect. 3.2 (lines 268–275) and shown in Fig. 5 within the discussion manuscript. This comment is also closely related to comment #2 above. As we replied in comment #2, the low-level of VOD during dry days is something we need to point out more clearly within the manuscript (see comment #2 for additional details).

**10.** *First paragraph of introduction section is detailed/explained well but not sufficiently referenced.*

All the references listed in lines 30–31 are the sources of information for the descriptions prior to that. We agree with Referee 2 that the references are too far removed from each of the descriptions. This accidentally happened as the manuscript was being modified/edited. In the revised manuscript, we have put the references in locations closer to the phenomena being described.

11. *Page 2, Line 22: Authors mention "Evapotranspiration ET is the sum of transpiration with soil and canopy evaporation" this is confusing; but ET = soil evaporation + canopy evaporation + transpiration?*

These were intended to be equivalent statements (are they not stating the same thing?). We have reworded line 22 to be:

> "Evapotranspiration ET is the sum of transpiration with soil evaporation/condensation and canopy evaporation/condensation (e.g., Stoy et al., 2019; Miralles et al., 2020). For our forest site, we assume that evaporation/condensation is primarily evaporation (i.e., transport of water vapor from the soil/canopy surfaces to the atmosphere) instead of condensation of water on these surfaces."

If this was not the intention of your comment, please clarify.

12. *Throughout the manuscript, there are instances where terms like 'evapotranspiration' are written in full repeatedly after being defined by their abbreviation (e.g., ET). I recommend using the respective abbreviations consistently after the first mention to improve readability and maintain consistency.*

We found a few places where this occurred. We have removed those instances in the revised manuscript; however, we have kept "evapotranspiration" written out fully within the abstract, figure captions, and the conclusions.

13. *Page 10, Line 210: Authors mention "To ease comparison with other studies" and no references were cited here. I recommend including relevant studies to support the statement.*

Good point. We have revised this sentence to be:

> To ease comparison with other studies (e.g., Klaassen et al., 1998; Humphrey and Frankenberg, 2023), we have expressed ET and the rate of precipitation in units of mm of $H_2O$ per hour (mm $h^{-1}$).

14. *Table 1 indicates VOD measurements began in June 2022, but the study period warm season is defined starting in Sept 2022. Please clarify on this.*

Thanks for pointing this out. The GNSS measurements started in June 2022, but we initially had both systems side-by-side at the top of the tower (this is described on line 140–141 in the discussion manuscript). However, it is true that VOD can only be measured with both an above-canopy and subcanopy system; therefore, we have revised Table 1 to show the VOD dates as "Aug 2022" rather than June.

15. *Please mention the temporal sampling interval for each observation type and ensure consistency throughout the manuscript.*

On line 210 we wrote, "Unless otherwise noted, ET and all other statistics are calculated over 30-min periods.". We have also added a new column to Table 1 which lists the averaging periods used for each variable.

**References**

Burns, S. P., Blanken, P. D., Turnipseed, A. A., Hu, J., and Monson, R. K.: The influence of warm-season precipitation on the diel cycle of the surface energy balance and carbon dioxide at a Colorado subalpine forest site, Biogeosciences, 12, 7349–7377, doi:10.5194/bg-12-7349-2015, 2015.

Holtzman, N. M., Anderegg, L. D. L., Kraatz, S., Mavrovic, A., Sonnentag, O., Pappas, C., Cosh, M. H., Langlois, A., Lakhankar, T., Tesser, D., Steiner, N., Colliander, A., Roy, A., and Konings, A. G.: L-band vegetation optical depth as an indicator of plant water potential in a temperate deciduous forest stand, Biogeosciences, 18, 739–753, doi:10.5194/bg-18-739-2021, 2021.

Hua, D., Hao, X., Zhang, Y., and Qin, J.: Uncertainty assessment of potential evapotranspiration in arid areas, as estimated by the Penman-Monteith method, Journal of Arid Land, 12, 166–180, doi:10.1007/s40333-020-0093-7, 2020.

Humphrey, V. and Frankenberg, C.: Continuous ground monitoring of vegetation optical depth and water content with GPS signals, Biogeosciences, 20, 1789–1811, doi:10.5194/bg-20-1789-2023, 2023.

Klaassen, W., Bosveld, F., and de Water, E.: Water storage and evaporation as constituents of rainfall interception, Journal Of Hydrology, 212, 36–50, doi:10.1016/S0022-1694(98)00200-5, 1998.

Kottek, M., Grieser, J., Beck, C., Rudolf, B., and Rubel, F.: World Map of the Köppen-Geiger climate classification updated, Meteor. Z., 15, 259–263, doi:10.1127/0941-2948/2006/0130, 2006.

Luo, Z. D., Deng, Z. J., Singha, K., Zhang, X. P., Liu, N., Zhou, Y. F., He, X. G., and Guan, H. D.: Temporal and spatial variation in water content within living tree stems determined by electrical resistivity tomography, Agricultural And Forest Meteorology, 291, doi:10.1016/j.agrformet.2020.108058, 2020.

Miralles, D. G., Brutsaert, W., Dolman, A. J., and Gash, J. H.: On the use of the term "Evapotranspiration", Water Resources Research, 56, e2020WR028 055, doi:10.1029/2020WR028055, 2020.

Raleigh, M. S., Gutmann, E. D., Van Stan, J. T., Burns, S. P., Blanken, P. D., and Small, E. E.: Challenges and capabilities in estimating snow mass intercepted in conifer canopies with tree sway monitoring, Water Resources Research, 58, doi:10.1029/2021WR030972, 2022.

Stoy, P. C., El-Madany, T. S., Fisher, J. B., Gentine, P., Gerken, T., Good, S. P., Klosterhalfen, A., Liu, S., Miralles, D. G., Perez-Priego, O., Rigden, A. J., Skaggs, T. H., Wohlfahrt, G., Anderson, R. G., Coenders-Gerrits, A. M. J., Jung, M., Maes, W. H., Mammarella, I., Mauder, M., Migliavacca, M., Nelson, J. A., Poyatos, R., Reichstein, M., Scott, R. L., and Wolf, S.: Reviews and syntheses: Turning the challenges of partitioning ecosystem evaporation and transpiration into opportunities, Biogeosciences, 16, 3747–3775, doi:10.5194/bg-16-3747-2019, 2019.

Sun, Z., Wang, Q., Batkhishig, O., and Ouyang, Z.: Relationship between Evapotranspiration and Land Surface Temperature under Energy- and Water-Limited Conditions in Dry and Cold Climates, Advances in Meteorology, 2016, 1835 487, doi:10.1155/2016/1835487, 2016.

Szilagyi, J.: Comment on "On the Estimation of Potential Evaporation Under Wet and Dry Conditions" by Z. Tu and Y. Yang, Water Resources Research, 58, e2022WR033 264, doi:10.1029/2022WR033264, 2022.

Tu, Z. and Yang, Y.: On the estimation of potential evaporation under wet and dry conditions, Water Resources Research, 58, e2021WR031 486, doi:10.1029/2021WR031486, 2022.

Yao, Y., Humphrey, V., Konings, A. G., Wang, Y., Yin, Y., Holtzman, N., Wood, J. D., Bar-On, Y., and Frankenberg, C.: Investigating diurnal and seasonal cycles of vegetation optical depth retrieved from GNSS signals in a broadleaf forest, Geophysical Research Letters, 51, e2023GL107121, doi:10.1029/2023GL107121, 2024.

---

## Author Comment (AC3)

**Reply to Referee #3**

**S. P. Burns et al.**

sean@ucar.edu

Date: June 5, 2025

The comments by Referee 3 are greatly appreciated. We have listed the comments by Referee 3 below in italics, followed by our responses. We added numbers to each of the specific comments so it is easier to reference the comments by Referee 3. Coauthor Mark Raleigh processed the tree sway frequency data and coauthor Vincent Humphrey processed the GNSS/VOD data. Both of these coauthors have been away from work since the comments by Referee 3 were posted. In order to keep the discussion moving forward and respond in a timely fashion, the lead author has crafted the responses below. We hope the replies are accurate; however, certain comments might need for future revision after consultation with Mark, Vincent, and the other coauthors.

*This paper compiles VOD, tree sway frequency, and flux tower data for a subalpine forest. Additional data include output from a land-surface model, and field/sensor data related to air temperature, bole temperature, bole apparent dielectric permittivity, wetness, and precipitation. Together, these data are used to suggest that VOD and tree sway frequency capture signals related to canopy evaporation. These measurements offer unique insight into canopy storage dynamics from completely independent observations. I don't have any major suggestions/comments for the authors. The following suggestions are meant to help the reader understand additional context for the tree sway measurements, interpretation of results, and to facilitate reproducibility/follow up studies.*

This is an accurate summary of our manuscript and we appreciate the effort made by Referee 3 in the interpretation of our data/results and to facilitate follow-up studies.

**Under the category "Main Comments":**

*I agree with Referee #2 in that parts of the tree sway methodology can be expanded, even briefly. Suggested expansions:*

We reply below to each specific comment.

1. *A) Line 145-146. If possible, include a brief description of the accelerometer methods, even though they are included in Raleigh 2022. Can go in the appendix, along with the other measurement expanded details if authors see fit.*

This is a good idea and we think it's best to put it in the main text rather than the appendix (which doesn't have any information about the tree sway frequency measurements). We used what is described in Raleigh et al. (2022) and modified the text near lines 145-146 to be:

"Processing of the raw tree sway accelerations [$m\,s^{-2}$] into frequency [Hz] used the method described by Raleigh et al. (2022), except that a 30-minute analysis window was used. A brief summary of the data-processing steps is as follows: (1) spectral analysis of the 12-Hz acceleration data determined a primary frequency of the tree-swaying motion $f_{sway}$ for each 30-min period, (2) a sliding 72-hr mean-filtered $f_{sway}$ time series was calculated, (3) any 30-min $f_{sway}$ outliers relative to the 72-hr mean-filtered data or values with low spectral power (i.e., due to low wind speeds) were removed, and (4) any missing 30-min time periods in the $f_{sway}$ time series were gap-filled using splines. A detailed description of these steps can be found in Sect. 3.2 of Raleigh et al. (2022)."

2. *B) Line 147. Why was a 30-minute window chosen? Were other windows tried but this was the best one for the analysis?*

The short answer is that the eddy-covariance fluxes are calculated over 30-min periods and the US-NR1 site data are typically averaged to 30-min periods for analysis. Depending on the goals of the study, there are situation when examining the high-frequency data is more appropriate (e.g., for spectral analysis). For the US-NR1 site (and many flux sites), 30 minutes is the standard data-sharing and analysis time period. A 30-min period also provides better temporal resolution (compared to hourly periods) when examining the diel cycle. We discuss this in more detail in item #3 of our reply to Referee #2. We did not do a systematic examination of time-window-averaging length for the tree sway data analysis.

3. *C) Line 148. What interpolation and smoothing function(s) were used?*

A MATLAB spline fit of the tree sway freq data was used to gap-fill the low-wind periods. An example is shown in Fig. 4 of Raleigh et al. (2022). We mention the spline interpolation in our modified text (shown in #2 above), but will not go into any more detail within our manuscript. Coauthor Mark Raleigh can provide more specific details about the spline fit after he returns to the office.

4. *D) Line 156-157. How was the value chosen to be 1? Was this done by offsetting the low frequency trend such that the average frequency over the time series was 1? If so, please include this.*

For the detrended tree sway frequency $f_d$ a value of 1 was chosen for two primary reasons: (i) it was not very different than the actual frequency and (ii) it added an "offset" between the detrended and raw data, so that plots (such as Fig. 4) do not have the $f_{sway}$ and $f_d$ data right on top of each other. As you suggested, it was done by removing the low-frequency trend and then adding back in a value of 1. The important take-away message (described on lines 158–159), is that the low-frequency detrending of the tree sway did not significantly impact how diel cycle of tree sway frequency changed with precipitation state (as shown in Fig. 4 and supplemental Fig. S5c).

5. *Similarly, Line 138, can there be a brief inclusion of the data processing for VOD? Even if it is a brief summary, it would be helpful. This can be in the appendix.*

We will include some of the key information about the VOD data processing which is extracted from Humphrey and Frankenberg (2023). We revised the text near line 138 to be:

> "The data processing and calculation of hourly VOD time series followed the procedures in Humphrey and Frankenberg (2023). A brief summary of the VOD data-processing steps is as follows: (1) calculate the GNSS signal attenuation due to the forest by comparing the GNSS signal strength from the forest receiver (gnssB) to the one above the canopy (gnssA), (2) use the signal-strength difference to estimate the forest transmissivity $\gamma$, and (3) calculate an initial estimate of VOD from $\text{VOD} = -\ln(\gamma)\cos(\theta)$. Because the GNSS system samples the forest at irregular temporal intervals and angles, the long-term mean as a function of azimuth and elevation angle is used to improve the precision of the hourly measurements (Humphrey and Frankenberg, 2023)."

6. *Table 1. Is it possible to add the original/raw frequency of measurements for each observation?*

We will add additional details to Table 1 (this was also suggested by Reviewer 2). There are two different considerations—one is the raw sampling frequency and the other is the averaging period (over which mean values or fluxes are calculated). In the revised manuscript, we will include both of these in separate columns in Table 1.

7. *Figure 2. Is the accelerometer associated with Pine 3 or Pine 4? Or are the dielectric permittivity sensor on two different trees? Please clarify.*

Good point. The 3-axis accelerometer is on a spruce tree that is very close to the main tower (within about 2 m of the tower) while Pine 3 and Pine 4 are located about 20-30 m southeast of the main tower. To clarify this, we added the approximate location of Pine 3 and Pine 4 in the sensor description on lines 482–483 (Sect A5).

8. *Line 347-349. Is it possible to expand on this (how these empirical relationships might facilitate changes in the land-surface model)? This seems like an important consideration, however there is nuance to it. In particular for tree sway frequency, the tree sway frequency is proportional to the inverse square root of tree mass, so while the frequency is observed to be linear, this does not mean that the tree mass (interception) is changing linearly. However, there is no mention of how tree sway frequency relates to mass in a mathematical form throughout the manuscript. I suggest adding this here and/or earlier on in the manuscript so if improvements on the land-surface model can begin there's the additional context of how physically sway frequency relates to changing tree mass due to interception.*

We are still thinking about this comment. The relationship between tree sway frequency and mass is shown by Eq. (1) in Raleigh et al. (2022). This is an important comment and we need a bit more time to consider what changes to make to the manuscript based on it. A full reply to this comment will be in our final summary of changes to the manuscript.

9. *This is not exceptionally important, and the authors can ignore: I was trying to determine any sort of relative magnitude differences in responses for the precipitation events presented in table 3 and figure 8 (and 9). I tried to find the lowest precipitation event (3.6 mm, solid cyan line) and the highest precipitation value (29 mm, solid red line). I think I found them in the sway frequency plot and they had the lowest and highest range/variability, which was helpful to see that with a first order approximation that the observations aligned with physical interpretations (more precipitation –> more change in tree sway frequency). I couldn't quite find them in the VOD diagram. Highlighting these two events with bolded lines might help the reader understand the relative sensitivity to single events and strengthen the connections observed between precipitation, evaporation, and VOD or sway frequency.*

We like the idea of highlighting the VOD and tree sway lines in Figs 8/9 for the lowest and highest precipitation values. We will try to add this to the revised manuscript. We should point out that Fig. 10b in the discussion manuscript has the information we believe you are trying to extract for VOD. This plot shows the precipitation amount for each storm vs the 4-hour mean VOD value from 4 to 8 hours after the storm stopped (note: we noticed that there is a mistake in the caption of Fig. 10 where the red dots are related to Fig. 9, not Fig. 8). Though only based on 17 points, Fig. 10b gives a rough idea of the relationship higher VOD values occur with higher precipitation amounts (as one would expect). Since we have many years of tree sway data, we could create a similar plot for tree sway that might reveal the pattern more clearly (based on more points). We will try this and report back in our final summary about changes to the manuscript. If this was not the intention of your comment, please clarify.

10. *Are the authors amenable to including in the acknowledgements or in brief 'open science' section in the appendix or supplemental info where the data and code used throughout the manuscript can be found? This would help future studies reproduce these results and follow up studies that are interested in conducting similar experiments elsewhere.*

Thanks for bringing this up. Most of the US-NR1 30-min flux/met data are already available via AmeriFlux (https://ameriflux.lbl.gov/sites/siteinfo/US-NR1) which is listed under the "Assets" tab in the discussion paper.

For the final paper, we are planning to create an ESS-DIVE archive for the raw GNSS data, as well as the processed tree sway and flux data, and other data from our manuscript. The planned ESS-DIVE archive will have a format very similar a 2020 ESS-DIVE archive that is listed in the Assets tab:

> Burns, S.P., P.D. Blanken, and R.K. Monson, 2020: *Data, Photographs, Videos, and Information for the Niwot Ridge Subalpine Forest (US-NR1) AmeriFlux site.* AmeriFlux Management Project, ESS-DIVE Dataset,
> https://doi.org/10.15485/1671825

The goal is to have the GNSS ESS-DIVE archive created and included in the revised manuscript under the paper Assets tab. We need to have further discussion about including the data-processing code within the ESS-DIVE archive (or perhaps made available elsewhere).

**References**

Humphrey, V. and Frankenberg, C.: Continuous ground monitoring of vegetation optical depth and water content with GPS signals, Biogeosciences, 20, 1789–1811, doi:10.5194/bg-20-1789-2023, 2023.

Raleigh, M. S., Gutmann, E. D., Van Stan, J. T., Burns, S. P., Blanken, P. D., and Small, E. E.: Challenges and capabilities in estimating snow mass intercepted in conifer canopies with tree sway monitoring, Water Resources Research, 58, doi:10.1029/2021WR030972, 2022.

---

## Author Comment (AC4)

**Reply to Referee #4**

**S. P. Burns et al.**

sean@ucar.edu

Date: June 6, 2025

The comments by Referee 4 are greatly appreciated. We have listed the comments by Referee 4 below in italics, followed by our responses.

*This research investigates promising and novel techniques to predict rainfall interception and, thus indirectly, canopy evaporation. The authors use L-band microwave active microwave attenuation data (vegetation optical depth) from a GNSS doublet and tree sway data measured by a an accelerometer placed on the trunk of a tree canopy. The study is set in an subalpine, high elevation needleleaf forest in in Colorado/USA. The authors demonstrate the ability of both proxies to correlate with onset and drydown of precipiation events, evapotranspiration and modeled interception storage from different land surface model (CLM4.5) parametrizations. The data sets and analysis presented by the authors allow for the conclusion that these techniques are promising tools to measure interception storage and that they hold potential to supplement/validate land surface models that are known to have high uncertainties in interception fluxes the their parametrization of the canopy, and uncertainties in EC water flux measurements during rain events. This study is of great quality. However, the authors should address the comments below before publication.*

This is an accurate summary of our study and we appreciate the positive comments that our study is of "great quality". The comments by Referee 4 focus on the footprint analysis and details about the VOD measurements. To reply in a timely manner, we only address two of the specific comments below. The other comments will be addressed in our complete responses to all reviewer comments that will be included in the paper revisions.

**Under the category "Major comment":**

*Please elaborate on the robustness of tree sway motion being able to represent interception storage without the need to account for wind speed as a possible confounding factor. In this context, it would be valuable to find sway motion data as a function of wind speed—e.g. in fig. 10 and at least in one of the plot over time—to clarify on this relationship and include this missing piece of information.*

Based on mechanical theory, tree sway frequency $f_{sway}$ acts like a damped harmonic oscillator; therefore, is does not depend on wind speed. This is highlighted in Sect.3 of Raleigh et al. (2022) who show that $f_{sway}$ is described by:

$$f_{sway} \propto \frac{1}{2\pi} \left( \frac{K}{m} \right)^{0.5},$$

(1)

where $K$ is the flexural rigidity and $m$ is the mass of the tree. As precipitation accumulates on the

tree leaves and branches, $m$ changes which alters $f_{sway}$. We will further address this aspect of our study more explicitly in our revised manuscript.

**Under the category "Minor comments":**

Comments 1–4 will be addressed at a later time.

*5. 433/4: The size of the EC footprint has not been explicitly mentioned in the text. Also, which footprint size of VOD as your referring to in this statement? To make a statement about the footprint size (a very relevant discussion) the referee suggests to state that although the footprint sizes between all technique partly or greatly differed, the good correlations could be found etc.*

We thank Referee 4 for noticing this shortcoming. Shortly after making the submission, the lead author realized that though we make reference the ET flux footprint, we did not include any specific details about it. We appreciated that this oversight was noticed by Referee 4, and will include the following information in the revised manuscript. First, we will cite Chu et al. (2021) who show a footprint climatology and suggest that the US-NR1 footprint has a size of around $500\,\mathrm{m}^2$, but depends on wind direction and atmospheric stability. A more explicit view of the US-NR1 footprint is shown below for five different atmospheric stability conditions in Fig. R1 for winds from the west, and in Fig. R2 for winds from the east. For comparison purposes, we include the larger VOD footprint from Fig. 1 on the same plot. The tower footprints have been calculated using the simple footprint model of (Kljun et al., 2015). For the revised manuscript, we plan to include similar plots in the supplemental figures and improve the details about the ET flux footprint within the text. One important difference between the VOD and flux footprints is that the VOD footprint does not vary with wind direction or atmospheric stability.

[Figure]

Figure R1: Contours of the 80% flux footprint are shown for winds from the west for different stability classes (SU, strongly unstable; WU, weakly unstable; NN, near-neutral; WS, weakly stable; SS, strongly stable). Footprints are calculated based on Kljun et al. (2015) and shown as distance [meters] from the main US-NR1 flux tower. The larger VOD footprint from Fig. 1 is shown as a black circle.

[Figure]

Figure R2: As in Fig. R1, but for winds from the east.

**References**

Chu, H., Luo, X., Ouyang, Z., Chan, W. S., Dengel, S., Biraud, S. C., Torn, M. S., Metzger, S., Kumar, J., Arain, M. A., Arkebauer, T. J., Baldocchi, D., Bernacchi, C., Billesbach, D., Black, T. A., Blanken, P. D., Bohrer, G., Bracho, R., Brown, S., Brunsell, N. A., Chen, J., Chen, X., Clark, K., Desai, A. R., Duman, T., Durden, D., Fares, S., Forbrich, I., Gamon, J. A., Gough, C. M., Griffis, T., Helbig, M., Hollinger, D., Humphreys, E., Ikawa, H., Iwata, H., Ju, Y., Knowles, J. F., Knox, S. H., Kobayashi, H., Kolb, T., Law, B., Lee, X., Litvak, M., Liu, H., Munger, J. W., Noormets, A., Novick, K., Oberbauer, S. F., Oechel, W., Oikawa, P., Papuga, S. A., Pendall, E., Prajapati, P., Prueger, J., Quinton, W. L., Richardson, A. D., Russell, E. S., Scott, R. L., Starr, G., Staebler, R., Stoy, P. C., Stuart-Haëntjens, E., Sonnentag, O., Sullivan, R. C., Suyker, A., Ueyama, M., Vargas, R., Wood, J. D., and Zona, D.: Representativeness of eddy-covariance flux footprints for areas surrounding AmeriFlux sites, Agricultural and Forest Meteorology, 301-302, 108 350, doi:10.1016/j.agrformet.2021.108350, 2021.

Kljun, N., Calanca, P., Rotach, M. W., and Schmid, H. P.: A simple two-dimensional parameterisation for Flux Footprint Prediction (FFP), Geosci. Model Dev., 8, 3695–3713, doi:10.5194/gmd-8-3695-2015, 2015.

Raleigh, M. S., Gutmann, E. D., Van Stan, J. T., Burns, S. P., Blanken, P. D., and Small, E. E.: Challenges and capabilities in estimating snow mass intercepted in conifer canopies with tree sway monitoring, Water Resources Research, 58, doi:10.1029/2021WR030972, 2022.

---

## Author Comment (AC5)

**Reply to Referee #4**

**S. P. Burns et al.**

sean@ucar.edu

Date: June 27, 2025

The comments by Referee 4 are greatly appreciated. We have listed the comments by Referee 4 below in italics, followed by our responses. For completeness, we reply to all of Referee 4's comments and have refined/improved the previous replies; therefore, these comments supersede anything that differs from our initial reply to Referee 4 on 6 June 2025.

*This research investigates promising and novel techniques to predict rainfall interception and, thus indirectly, canopy evaporation. The authors use L-band microwave active microwave attenuation data (vegetation optical depth) from a GNSS doublet and tree sway data measured by a an accelerometer placed on the trunk of a tree canopy. The study is set in an subalpine, high elevation needleleaf forest in in Colorado/USA. The authors demonstrate the ability of both proxies to correlate with onset and drydown of precipitation events, evapotranspiration and modeled interception storage from different land surface model (CLM4.5) parameterizations. The data sets and analysis presented by the authors allow for the conclusion that these techniques are promising tools to measure interception storage and that they hold potential to supplement/validate land surface models that are known to have high uncertainties in interception fluxes the their parameterization of the canopy, and uncertainties in EC water flux measurements during rain events. This study is of great quality. However, the authors should address the comments below before publication.*

This is an accurate summary of our study and we appreciate the positive comments that our study is of "great quality". The specific comments by Referee 4 are replied to below.

**Under the category "Major comment":**

*Please elaborate on the robustness of tree sway motion being able to represent interception storage without the need to account for wind speed as a possible confounding factor. In this context, it would be valuable to find sway motion data as a function of wind speed—e.g. in fig. 10 and at least in one of the plot over time—to clarify on this relationship and include this missing piece of information.*

Based on mechanical theory, the natural sway frequency $f_{sway}$ of a conifer tree acts like a damped harmonic oscillator; therefore, it does not depend on wind speed. This is highlighted in Sect.3 of Raleigh et al. (2022) as well as many other studies (e.g., Moore and Maguire, 2004; Van Emmerik et al., 2017; Jackson et al., 2021), who show that $f_{sway}$ is described by the cantilever model:

$$f_{sway} \propto \frac{1}{2\pi} \left( \frac{K}{m} \right)^{0.5},$$

(1)

where $K$ is the flexural rigidity of the tree and $m$ is the mass of the tree, including the branches and

[Figure]

Figure R1: Similar to Fig. 10 in the discussion manuscript, but using detrended tree sway frequency $f_d$ rather than VOD. In **(a)**, the relationship between above-canopy mean horizontal wind speed WS and total precipitation amount from the wet day preceding the wDry day is shown. We have multiplied $f_d$ by $-1$ to create patterns consistent with VOD (as in Fig. 10). The solid black points are the 30-min mean values from the warm season periods for years 2016 to 2023, the red points are the mean values calculated between 4 and 8 hours after precipitation ended. See Fig. 10 for additional details.

leaves/needles. As precipitation accumulates on the tree leaves and branches, $m$ changes which alters $f_{sway}$. Therefore, $f_{sway}$ is primarily a structural property of the tree that depends on mass (tree biomass + water in/on the tree), elasticity (which varies with tree temperature, thermal state, and water content), and tree geometry (tree height and DBH). There are other models which are appropriate for other tree types (e.g., simple pendulum for broadleaf trees), but across model types (cantilever and pendulum) tree frequency is not predicted by wind speed.

As alluded to by Referee 4, this technique requires some minimal level of wind speed to generate tree sway. This was mentioned on lines 400-401, as one of the limitations of the method. The lack of a dependence of tree sway on wind speed is discussed in detail in the supplemental material of Raleigh et al. (2022) (see their Fig.S5) as well as the other papers listed above. The mean wind speed impacts the amplitude of oscillations, rather than the frequency (which is what is used in our study). As explained in Raleigh et al. (2022), periods with absolute calm-wind conditions were gap-filled by interpolation. With this said, we agree with Referee 4 that it is worth creating something like Fig. 10, but using tree sway frequency rather than VOD (especially since we have 6 years of tree sway freq data). This new figure is shown in Fig. R1, and many of the relationships between tree sway frequency and other variables (VPD, wind speed, turbulence) are similar to those with VOD. In Fig. R1d, the relationship between tree sway frequency and wind speed is shown and it can be seen that the periods with lower tree sway freq for WS less than 5 m/s correspond to wetter periods (ie, the red dots). The other take-away from Fig. R1d, is that higher winds (and turbulence levels) are less conducive to changes in tree sway frequency, this is either because the rainwater is mechanically blown/shaken off the tree or it does not rain as often in windy conditions (as shown in Fig. R1a). We intend to include Fig. R1 in the revised manuscript with additional discussion about it.

**Under the category "Minor comments":**

*1. Fig 1*

*1.1 Please report the the inner circle radius (r=20m) as the authors have done for the outer circle*

*1.2 Please point the reader of fig. 1's caption to what the different footprint circles represent to better understand the results, i.e. what is the main take-away from the inner circle radius (apart from GNSS paucity visualization).*

Because comments 1.1 and 1.2 are related, we answer them together. The meaning of the inner footprint was not clear in our submitted manuscript. There are several points and clarifications that we need to add to the footprint discussion in Appendix B. First, we need to clarify that no clear-cut boundary on the VOD footprint location exists. The exact contribution of individual trees to the VOD measurement will depend on the height of the tree and the proximity of the tree to the GNSS antenna. The inner footprint was an attempt to show that trees closer to the subcanopy GNSS antenna contribute more to the VOD flux. In Fig. R2, we have attempted to show this schematically from a side-view perspective of the forest. The tree shading in Fig. R2 shows how the trees closer to the subcanopy GNSS antenna contribute more to the VOD measurement. We intend to include Fig. R2 in the supplemental material of the revised manuscript and improve the text in Appendix B.

*1.3 Appendix B, line 527: Please clarify the role of r=20 in this study. Did the authors clip the radius so the "several tall trees" are not included in the footprint?*

The inner and outer VOD footprints shown in Fig. 1 are purely conceptual and no trees were removed from the data-processing based on these footprints. A better representation of the areas excluded in the data processing are shown in the skyplot (Fig. S3) where regions were excluded based on the lightning dissipator locations (not based on the trees location or the footprint radius).

[Figure]

Figure R2: Schematic of how the VOD footprint varies with tree height and distance from the subcanopy GNSS antenna. The trees with darker shading indicate a larger contribution to the VOD footprint.

*2. Appendix B & Fig. S3 (GNSS sky view):*

*2.1 Please show the elevation angle and cutoff to clarify which parts of the canopy will effectively be used for VOD calculation, especially elevation=10°*

The schematic in Fig. R2 shows an elevation angle of 10 degrees and which parts of the forest will be used. As discussed above, we plan to include this schematic and additional discussion about it in the revised manuscript.

*2.2 Please clarify the rationale behind clipping out another area in NE, close to the northern GNSS gap*

There are two lightning dissipators at the top of the tower, so two locations in the skyview are excluded (the lightning dissipators are described in Sect. 2.2.1, lines 130–131 of the discussion manuscript). To clarify what the lightning dissipators look like, photos of them are in Fig. R3.

*2.3. You use a very low cutoff elevation angle of 10°. Looking on fig. S3 – assuming the outer two circles being roughly within θ in (10, 30] – only very low VOD can be found that do not display any pattern expected from forest attenuation and possibly fail to represent true forest VOD. Under this light, please explain why the authors used a cutoff=10°.*

In principle, the lower the elevation angle, the more homogeneous the VOD should be because it is sampling across a larger number of trees in all directions. Thus, with a high enough canopy, the values at lower elevation angles should converge towards the average VOD at the site, which seems to be the case here (it goes to around 0.4 it seems). Because values from lower elevation angles mainly sample the upper half of the trees, it is true that including them could bias the VOD estimate somewhat, especially if the canopy density is very heterogeneous in the vertical direction. On the other hand, including these low elevations (10–30 deg.) increases the representativity (larger footprint) and the sample size (number of raw measurements), which reduces noise in the VOD time series. It is of course a trade-off, which this study does not aim to explore at this stage.

*2.4. Since the Lambert-Beer angle correction assumes a homogeneous canopy and ignores multipath scattering, any losses observed may be due to scattering caused by multiple layers of vegetation, causing Lambert-Beer to break at low angles. Hence, the referee suggests using a higher cutoff elevation angle (∼30 deg.) or would value a discussion why low VOD at lower angles will not affect the overall results. Consider page 12 in Camps et al. (2020) about this question: "Note, however, that only at high elevation angles (elevation angle > 67.5°) is the single scattering albedo correlated with the NDVI, and at lower elevation angles, the presence of multiple scattering makes the tau-omega model [all zeroth order assumption, incl. lambert-beer, ∼the referee] more likely to be invalid."*

Thank you for bringing the work by Camps et al. (2020) to our attention. See answer to comment 2.3 above for related details; For our study, the fact that the low elevation values converge to the mean VOD tends to suggest that these values are not so systematically biased that they would need to be removed. We would have acted differently if low elevation values had a different behavior. Though we don't expect the VOD to change much, we agree with Referee 4 that the effect of using a different elevation angle should be further explored. We will recalculate VOD with an elevation angle of 30 degrees to see how it changes the VOD values. This will take a bit of time to do, so we do not have results to share right now, but we plan to include this information in our revised manuscript.

*3. 135: Which GNSS frequency is used, please indicate the frequency(ies) in section 2.2.1 since GNSS VOD offers a range of bands to choose from.*

Good point. On line 136, the general L-band frequency range (1000-2000 MHz) is described. The specific GNSS frequency used was 1575 MHz which we added to the text near line 136.

*4. The authors detrend sway motion to alleviate effects of temperature and vegetation water content on short-term changes. However, VOD is also affected by long-term changes in biomass, and vegetation water content. Why did the authors not consider detrending VOD, especially since a trend is visible in fig. 2? This is worth noting in 2.2.1.*

Any low-frequency trend in VOD appears to be much smaller than that of tree sway frequency (i.e., compare the VOD time series in Fig. 2b to that of tree sway frequency in Fig. 2c). Since removing the low-frequency trend in tree sway frequency did not affect the results, we have assumed that removing any low-frequency trend in VOD will also have a minimal effect on the results. We have added text about not removing any low-frequency trend in VOD to Sect. 3.1, where the time series are discussed.

*5. 433/4: The size of the EC footprint has not been explicitly mentioned in the text. Also, which footprint size of VOD as your referring to in this statement? To make a statement about the footprint size (a very relevant discussion) the referee suggests to state that although the footprint sizes between all technique partly or greatly differed, the good correlations could be found etc.*

We thank Referee 4 for noticing this shortcoming. Shortly after making the submission, the lead author realized that though we make reference the ET flux footprint, we did not include any specific details about it. We appreciate that this oversight was noticed by Referee 4, and will include the following information in the revised manuscript. First, we will cite Chu et al. (2021) who show a footprint climatology and suggest that the US-NR1 footprint has a size of around $500\,m^2$, but depends on wind direction and atmospheric stability. A more explicit view of the US-NR1 flux footprint climatology is shown below for five different atmospheric stability conditions for winds from the west (Fig. R4) and for winds from the east (Fig. R5). The data are separated into east and west wind directions because winds at the site are typically either upslope (from east) or downslope (from west). For comparison purposes, we include the larger VOD footprint from Fig. 1 on the same plot. The tower footprints have been calculated using the simple footprint model of (Kljun et al., 2015). For the revised manuscript, we plan to include Figs. R4 and R5 within the supplemental material

and add a description of the ET flux footprint to Appendix B. One important difference between the VOD and flux footprints that we will emphasize: the flux footprint location varies with wind direction and atmospheric stability whereas the VOD footprint is unaffected by these variables.

[Figure]

[Figure]

Figure R3: Photos of the lightning dissipators at the top of the US-NR1 flux tower. The right-side photo shows the gnssA antenna on the southwest corner (left/front corner in photo) of the tower and the lightning dissipator on the southeast corner (right corner in photo) of the tower. There is 2nd lightning dissipator on the northwest corner that is barely visible in the right photo, but is the one shown in the left-side photo. The dissipators extend about 2 m above the top of the tower.

[Figure]

Figure R4: Climatology of the footprint region from which 80% of the 21.5 m turbulent scalar flux originates are shown for winds from the west for different stability classes (SU, strongly unstable; WU, weakly unstable; NN, near-neutral; WS, weakly stable; SS, strongly stable). These are US-NR1 data from July for years 1999–2023 where the number of 30-min samples within each stability category are shown by $N$ in the legend. Footprints are calculated based on Kljun et al. (2015) and shown as distance [meters] from the main US-NR1 flux tower. The larger VOD footprint from Fig. 1 is shown as a black circle.

[Figure]

Figure R5: As in Fig. R4, but for winds from the east.

**References**

Camps, A., Alonso-Arroyo, A., Park, H., Onrubia, R., Pascual, D., and Querol, J.: L-Band vegetation optical depth estimation using transmitted GNSS signals: Application to GNSS-reflectometry and positioning, Remote Sensing, 12, doi:10.3390/rs12152352, 2020.

Chu, H., Luo, X., Ouyang, Z., Chan, W. S., Dengel, S., Biraud, S. C., Torn, M. S., Metzger, S., Kumar, J., Arain, M. A., Arkebauer, T. J., Baldocchi, D., Bernacchi, C., Billesbach, D., Black, T. A., Blanken, P. D., Bohrer, G., Bracho, R., Brown, S., Brunsell, N. A., Chen, J., Chen, X., Clark, K., Desai, A. R., Duman, T., Durden, D., Fares, S., Forbrich, I., Gamon, J. A., Gough, C. M., Griffis, T., Helbig, M., Hollinger, D., Humphreys, E., Ikawa, H., Iwata, H., Ju, Y., Knowles, J. F., Knox, S. H., Kobayashi, H., Kolb, T., Law, B., Lee, X., Litvak, M., Liu, H., Munger, J. W., Noormets, A., Novick, K., Oberbauer, S. F., Oechel, W., Oikawa, P., Papuga, S. A., Pendall, E., Prajapati, P., Prueger, J., Quinton, W. L., Richardson, A. D., Russell, E. S., Scott, R. L., Starr, G., Staebler, R., Stoy, P. C., Stuart-Haëntjens, E., Sonnentag, O., Sullivan, R. C., Suyker, A., Ueyama, M., Vargas, R., Wood, J. D., and Zona, D.: Representativeness of eddy-covariance flux footprints for areas surrounding AmeriFlux sites, Agricultural and Forest Meteorology, 301-302, 108 350, doi:10.1016/j.agrformet.2021.108350, 2021.

Jackson, T. D., Sethi, S., Dellwik, E., Angelou, N., Bunce, A., van Emmerik, T., Duperat, M., Ruel, J.-C., Wellpott, A., Van Bloem, S., Achim, A., Kane, B., Ciruzzi, D. M., Loheide II, S. P., James, K., Burcham, D., Moore, J., Schindler, D., Kolbe, S., Wiegmann, K., Rudnicki, M., Lieffers, V. J., Selker, J., Gougherty, A. V., Newson, T., Koeser, A., Miesbauer, J., Samelson, R., Wagner, J., Ambrose, A. R., Detter, A., Rust, S., Coomes, D., and Gardiner, B.: The motion of trees in the wind: A data synthesis, Biogeosciences, 18, 4059–4072, doi:10.5194/bg-18-4059-2021, 2021.

Kljun, N., Calanca, P., Rotach, M. W., and Schmid, H. P.: A simple two-dimensional parameterisation for Flux Footprint Prediction (FFP), Geosci. Model Dev., 8, 3695–3713, doi:10.5194/gmd-8-3695-2015, 2015.

Moore, J. R. and Maguire, D. A.: Natural sway frequencies and damping ratios of trees: Concepts, review and synthesis of previous studies, Trees, 18, 195–203, doi:10.1007/s00468-003-0295-6, 2004.

Raleigh, M. S., Gutmann, E. D., Van Stan, J. T., Burns, S. P., Blanken, P. D., and Small, E. E.: Challenges and capabilities in estimating snow mass intercepted in conifer canopies with tree sway monitoring, Water Resources Research, 58, doi:10.1029/2021WR030972, 2022.

Van Emmerik, T., Steele-Dunne, S., Hut, R., Gentine, P., Guerin, M., Oliveira, R. S., Wagner, J., Selker, J., and Van de Giesen, N.: Measuring Tree Properties and Responses Using Low-Cost Accelerometers, Sensors, 17, doi:10.3390/s17051098, 2017.

---

## Author Comment (AC6)

**List of Revisions to egusphere-2025-1755**

**S. P. Burns et al.**

sean@ucar.edu

Date: June 27, 2025

Listed below is a summary of the major planned and/or completed revisions to manuscript egusphere-2025-1755 submitted for publication in the EGU journal *Biogeosciences*. These revisions are primarily based on comments made by four Referees. Here, we summarize the significant/major revisions; additional manuscript changes are described in our point-by-point responses to the reviewer comments available on the manuscript discussion webpage. If the editor agrees with our proposed revisions, then we will submit our revised manuscript.

1. Both Referees #1 and #2 commented on the lack of a clear diel cycle in VOD during dry conditions. As part of our replies to these comments, we realized that VOD at the US-NR1 site (during dry days) is on the order of 0.35–0.4 which is much lower than VOD in other studies which are on the order of 0.6–1.1 (e.g., Holtzman et al., 2021; Humphrey and Frankenberg, 2023; Yao et al., 2024). In order to properly compare the VOD between these sites, the forest characteristics are needed. While we most likely won't be able to fully determine the cause of the low US-NR1 VOD, we will add additional discussion about this in the revised manuscript, and, at the very least, point out that the US-NR1 VOD in dry conditions is lower than VOD in the other studies (see our replies to Referees #1 and #2 for additional details).

2. Based on suggestions from Referees #1 and #2, we moved some information from the Introduction into the "Materials and Methods" section, and re-wrote the part of the Introduction where the comparison with CLM4.5 is described.

3. Based on comments from Referees #2 and #3, we have included information about the sampling frequency of each variable in Table 1 and fixed the time period that VOD was measured.

4. Based on comments from Referees #2 and #3, we have expanded our description of the data processing steps for both VOD (Sect. 2.2.1) and tree sway frequency (Sect. 2.2.2).

5. Based on comments from Referees #3 and #4, we have examined the effect of precipitation amount and wind speed on the tree sway frequency (similar to Fig. 10 in the discussion manuscript). We intend to add a figure similar to Fig. 10, but using tree sway frequency as the dependent variable. See Fig. R1 in our reply to Referee #4 for details.

6. Based on comments from Referee #4, we have clarified the meaning of the smaller VOD footprint shown in Fig. 1 and added a side-view schematic of the VOD footprint (see Fig. R2 in our reply to Referee #4). In our revised manuscript, we will: (1) improve our discussion of the VOD footprint (and add Fig. R2 to the supplemental material), (2) expand/improve the text in Appendix B to include a discussion about the eddy-covariance ET flux footprint, (3) contrast the ET and VOD footprints, and (4) add the ET flux footprint figures (shown as Figs. R4 and R5 in our reply to Referee #4) to the supplemental material.

7. Based on comments from Referee #4, we will re-calculate VOD using an elevation angle of 30 degrees and check that it does not significantly modify the US-NR1 VOD values.

8. We will create an ESS-DIVE archive to share the raw GNSS data used in our study, as well as other data. We were planning to do this; it was also suggested by Referee #3 (item 10).

9. In the Conclusions section of the revised manuscript, we will improve the text readability.

**References**

Holtzman, N. M., Anderegg, L. D. L., Kraatz, S., Mavrovic, A., Sonnentag, O., Pappas, C., Cosh, M. H., Langlois, A., Lakhankar, T., Tesser, D., Steiner, N., Colliander, A., Roy, A., and Konings, A. G.: L-band vegetation optical depth as an indicator of plant water potential in a temperate deciduous forest stand, Biogeosciences, 18, 739–753, doi:10.5194/bg-18-739-2021, 2021.

Humphrey, V. and Frankenberg, C.: Continuous ground monitoring of vegetation optical depth and water content with GPS signals, Biogeosciences, 20, 1789–1811, doi:10.5194/bg-20-1789-2023, 2023.

Yao, Y., Humphrey, V., Konings, A. G., Wang, Y., Yin, Y., Holtzman, N., Wood, J. D., Bar-On, Y., and Frankenberg, C.: Investigating diurnal and seasonal cycles of vegetation optical depth retrieved from GNSS signals in a broadleaf forest, Geophysical Research Letters, 51, e2023GL107121, doi:10.1029/2023GL107121, 2024.

---

## Author Response (AR1)

**Mesoscale & Microscale Meteorology Laboratory**

P.O. Box 3000, Boulder, CO 80307-3000 USA • P: (303) 497-8934 • F: (303) 497-8171

Sean P. Burns • sean@ucar.edu

August 15, 2025

Dr. Andrew Feldman Biogeosciencs Associate Editor Copernicus Publications editorial@copernicus.org

Dear Dr. Feldman,

Thank you for the assessment of "minor revisions" needed for our manuscript (egusphere-2025-1755) entitled, "Using GNSS-based vegetation optical depth, tree sway motion, and eddy-covariance to examine evaporation of canopy-intercepted rainfall in a subalpine forest" by myself, Vincent Humphrey, Ethan Gutmann, Mark Raleigh, Dave Bowling, and Peter Blanken. We are encouraged by the positive reviews and our revised manuscript has been re-submitted electronically via the Copernicus website for publication as a research article in the EGU journal *Biogeosciences*.

The re-submitted PDF was created using a recent EGU latex style file and includes the tables and figures embedded within the main text, followed by the supplemental information. We have also uploaded a list of the changes we made to our manuscript, our replies to your comments and all the referee comments, and a PDF of the manuscript text which highlights in color where changes within the text have been made. If our proposed changes are not acceptable to you or the reviewers, please let us know and we will make further revisions following your advice. If a decision is made to accept the article for publication, then we will upload the latex file used to generate the PDF and all the individual figure files (as PDFs).

If there are any questions or problems with our re-submission please don't hesitate to contact me.

Sincerely,

Sean P. Burns

**Comment on "Using GNSS-based vegetation optical depth, tree sway motion, and eddy-covariance to examine evaporation of canopy-intercepted rainfall in a subalpine forest" by S. P. Burns et al.**

**List of Revisions to egusphere-2025-1755**

**S. P. Burns et al.**

sean@ucar.edu

Date: August 15, 2025

Listed below is a summary of the major changes to manuscript egusphere-2025-1755 submitted for publication in the EGU journal *Biogeosciences*. These revisions are primarily based on comments made by the four Referees and associate editor. In this list we summarize the significant/major revisions; additional manuscript changes are described in our point-by-point responses to the editor and reviewer comments which are attached following the below list.

- 1. Both Referees #1 and #2 commented on the lack of a clear diel cycle in VOD during dry conditions. As part of our replies to these comments, we realized that VOD at the US-NR1 site (during dry days) is on the order of 0.35–0.4 which is much lower than VOD in other studies which are on the order of 0.6–1.1 (e.g., Holtzman et al., 2021; Humphrey and Frankenberg, 2023; Yao et al., 2024). In order to properly compare VOD between these sites, the forest characteristics are needed. While we cannot fully determine the reason for the low US-NR1 VOD, we have added additional discussion about this in the revised manuscript, and point out that the US-NR1 VOD in dry conditions is lower than VOD in the other studies (see our replies to Referees #1 and #2 for additional details).
- 2. Based on suggestions from Referees #1 and #2, we moved some information from the Introduction into the "Materials and Methods" section, and re-wrote the part of the Introduction where the comparison with CLM4.5 is described.
- 3. Based on comments from the associate editor and Referee #1, we moved the figure that shows how the CLM4.5 data vary for each precipitation state (Fig. S11 in the discussion paper) into the main text (Fig. 8 in the revised manuscript). This was done to better highlight the CLM4.5 results within our paper. Further discussion on the CLM4.5 comparison is within our reply to the associate editor and Referee #1.
- 4. Based on comments from Referees #2 and #3, we have included information about the sampling frequency of each variable in Table 1 and fixed the time period that VOD was measured.
- 5. Based on comment #9 from Referee #3, we have highlighted the time series with the minimum and maximum precipitation amounts in Figs. 8 and 9 in the discussion paper (Figs. 9 and 10 in the revised manuscript). We agree with Referee #3 that adding these highlights nicely shows the min and max ends of the precipitation spectrum.
- 6. Based on comments from Referees #2 and #3, we have expanded our description of the data processing steps for both VOD (Sect. 2.2.1) and tree sway frequency (Sect. 2.2.2).

- 7. Based on comments from the associate editor and Referees #3 and #4, we have examined the effect of precipitation amount and wind speed on the tree sway frequency (similar to Fig. 10 in the discussion manuscript). We added a new figure similar to Fig. 10 (Fig. 12 in the revised manuscript), but using tree sway frequency as the dependent variable. This figure is also shown as Fig. R1 and discussed in our reply to Referee #4.
- 8. Based on comments from Referee #4, we have clarified the meaning of the smaller VOD footprint shown in Fig. 1 and added a side-view schematic of the VOD footprint (see Fig. S4 in the revised manuscript and Fig. R2 in our reply to Referee #4). In our revised manuscript, we have: (1) improved our discussion of the VOD footprint (and added Fig. S4 to the supplemental material), (2) expanded/improved the text in Appendix B to include a discussion about the eddy-covariance ET flux footprint, (3) contrasted the ET and VOD footprints, and (4) added the ET flux footprint figures as Figs. S12 and S13 in the revisited manuscript (these are also discussed in our reply to Referee #4 and shown as Figs. R5 and R6 in that reply).
- 9. Based on comments from Referee #4, we have re-calculated VOD using an elevation angle of 30 degrees and confirmed that it does not significantly modify the US-NR1 VOD values (see our reply to Referee #4 for details).
- 10. As part of our original plan (and suggested by Referee #3, #10), we have created a new ESS-DIVE archive that includes all the raw GNSS data used in our study (a 40 Gb dataset), as well as all other data and software used within our study. The new ESS-DIVE archive is:

Burns S. P., Humphrey V., Raleigh M. S., Bowling D. R., Gutmann E. D., and Blanken P. D. (2025): GNSS-based Vegetation Optical Depth (VOD), Tree Sway, and Evapotranspiration data from the Niwot Ridge Subalpine Forest (US-NR1) AmeriFlux site. AmeriFlux Management Project. Dataset. doi:10.15485/2574352

This DOI is currently "reserved" but not active (as of 15 August). We need to finish adding data to it and then have it reviewed by ESS-DIVE. We expect to have this completed by the end of August 2025.

Two other "Assets" we will add to the article are github pages that have the software used for tree sway data processing and the VOD calculation:

Raleigh M. S., Code to Process Tree Sway Frequency from Accelerometer Data, https://github.com/truewind/accelerometer\_tree\_sway/, last access: 14 August 2025.

Humphrey, V., Python Toolkit for Deriving Vegetation Optical Depth (VOD) from Pairs of GNSS Receivers, https://github.com/vincenthumphrey/gnssvod, last access: 14 August 2025.

- 11. In the Conclusions section of the revised manuscript, we have attempted to improve the text readability.
- 12. Additional references added to the manuscript are listed below (see "New References in Manuscript"). At the end of this document we attached a PDF which shows changes to the text using latexdiff (this is suggested in the "Manuscript preparation guidelines for authors" section on the BG website). Removed text is shown in red, added text is in blue.

**References**

[revised manuscript text omitted]

**Reply to Associate Editor, Andrew Feldman**

**S. P. Burns et al.**

sean@ucar.edu

Date: August 15, 2025

The comments by the Associate Editor are greatly appreciated. We have listed the comments by the Associate Editor below in italics, followed by our responses.

We've received four reviews from subject-matter experts on this topic. All four find high value in the manuscript. Most edits are requests for clarification, and I don't disagree with any. However, I want to point out a few common areas to focus your revisions.

We appreciate this comment-all four reviews have been very useful in improving the quality of our manuscript.

The main objective is relating tree sway to the intercepted water volume as captured by GNSS VOD. The evaporation estimation of intercepted water and the CLM analysis seem to be abruptly discussed and not clearly motivated in the introduction. What exactly do we learn from the evaporation estimates and the CLM analysis? Some clearer purpose statements can help readers understand the objectives as well as temper expectations. Additionally, tying the purpose of the evaporation estimates and CLM points more closely to the tree sway-VOD GNSS connection would be helpful throughout the results and discussion.

There have been discussions among the co-authors about the inclusion of the CLM4.5 data within our study; as lead author, I felt strongly that it was important to leave this in the study. In my opinion, the CLM4.5 comparison provides an example of how land-surface models might be lacking with regard to canopy evaporation. To bring this point out more clearly, we moved the figure that shows how the CLM4.5 data vary for each precipitation state (Fig. S11 in the discussion paper) into the main text (Fig. 8 in the revised manuscript). We also modified the text in the Conclusions to better highlight how canopy evaporation with CLM4.5 behaved more like our flat-plate wetness sensor than evaporation from the canopy.

I also agree with Reviewer 4 that it seemed like a large assumption to state that wind speed was not needed in relating interception storage to tree sway frequency (line 148). More motivation is required here.

The relationship between wind speed and tree sway has been well-documented in many studies. See our reply to the "Major Comment" by Reviewer #4 for additional details.

A few other notable points to pay attention to include Reviewers 2 and 3 asking for more detailed methodology descriptions in some cases (tree sway, accelerometer, interpolation methods).

We have added additional text to the revised manuscript with details about the data processing for VOD and tree sway. See our replies to Reviewer 3 (item #1 and item #5) for additional details.

Reviewer 3 also recommended a more physics-based description of the connection between GNSS VOD and tree sway (connection to water mass). Several reviewers ask questions about the detrending methods.

This has been added to the revised manuscript; details are in our reply to Reviewer 3 as well as the "Major Comment" by Reviewer 4.

A final note of my own: I thought that some claims could have been more quantitative throughout. For example, a claim was made about internal tree water content versus VOD on tree sway in lines 269-273, which appeared to rely on visual comparison in Figure 5. This could probably be argued in a more quantitative way.

We agree that portions of our paper are more "qualitative" than "quantitative" (for example, the claim that you note). However, we also feel that portions of our study are quantitative. For example, providing the linear fits of how VOD and tree sway frequency change with time (in Fig. 6) and the linear fit for VOD vs tree sway (in Fig. 4). This quantitative information can be used to compare/contrast our results to future studies.

Overall, while there are many comments, there are not fatal flaws raised with the argumentation and, therefore, the manuscript requires minor revisions.

Thank you for this assessment.

**Reply to Referee #1**

**S. P. Burns et al.**

sean@ucar.edu

Date: August 15, 2025

The comments by Referee 1 are greatly appreciated. We have listed the comments by Referee 1 below in italics, followed by our responses. We added numbers to each of the specific comments so it is easier to reference the comments by Referee 1.

Burns et al. measured warm season ET, VOD and tree sway in Colorado subalpine forest to was to track evaporation of intercepted water. They found that a canopy intercepted water can take a tremendous amount of time to evaporate and thus can contribute to ET at the site especially on dry days after a rain event. This work well describes VOD and tree sway as a method of measuring evaporation of canopy interception.

This is an accurate summary of our manuscript and the goals of the study.

**Under the category "Comments":**

1. To help distinguish these VOD measurements from more common measurements excluding, I suggest to specify in the acronym that this is VOD plus intercepted water. For example, VOD underscore "int" to distinguish from studies where VOD refers to water only within the tissues.

Though we appreciate the concept of making this distinction, the practical logistics of applying it are a bit difficult, and (in our opinion) could lead to confusion. For example, there are several plots that show VOD in both dry and wet conditions (e.g., Figs. 2 and 3) as well as plots that show VOD in only dry conditions (Fig. 5). So, it would be inaccurate to label the VOD in dry conditions as  $VOD_{int}$ . We think it is clear that the focus of our study is on how VOD is affected by wet conditions and leave it to the reader to distinguish when the forest surfaces are wet and/or dry.

2. line 66: Please clarify what the "small pockets" are referring to.

The "small pockets" were referring to the tracheids within the xylem that move the water in a conifer tree. In an attempt to clarify this text, we changed the words "small pockets" to "tracheids within the xylem that are on the order of 5 to 80  $\mu$ m in diameter and less than 5 mm long". We also added a new reference to Hacke et al. (2015) which contains this information. The sentence has been re-written as,

"This is because intercepted water forms a water layer which attenuates and reflects microwave signals more than internal water found within the xylem tubular structures; for conifers, tree water is primarily found within xylem tracheids that are on the order of 5 to 80  $\mu$ m in diameter and less than 5 mm long (e.g., Tyree and Ewers, 1991; Hacke et al., 2015)."

3. line 75: Parentheses missing around GNSS

The parentheses have been added.

4. line 95: The inclusion of the CLM analysis while exciting comes up abruptly here and the importance of this analysis could be integrated sooner. Suggest to reference Burns et al 2018 or specific the concepts from that paper that will be expanded upon here.

This is a good point. We added some context for the Burns et al. (2018) study. The original paragraph was:

"Though the focus of our study is on observations, we also wanted to compare the VOD and tree sway observations with land-surface model results. To achieve this, we used the Community Land Model CLM4.5 (e.g., Oleson et al., 2013). This expands on previous work with CLM4.5 at US-NR1 by Burns et al. (2018) to include modeled canopy surface water content (which we expect to be comparable with the VOD and tree sway frequency measurements)."

**The revised paragraph is:**

"Though the focus of our study is on observations, we also wanted to compare the VOD and tree sway observations with land-surface model results. In Burns et al. (2018) the Community Land Model CLM4.5 (Oleson et al., 2013) components of the latent heat flux (transpiration, canopy evaporation, and soil evaporation) were compared with the measured ecosystem-scale latent heat flux. Since the VOD and tree sway frequency measurements are related to canopy water content (and thus canopy evaporation), we compared these observations to CLM4.5-modeled canopy surface water content."

5. line 163: Suggest to replace "use the concept" with "assume"

We agree and changed "use the concept" to "assume".

6. Figure 3, 8: It would be helpful to please clarify in the results why there is not diurnal cycle of VOD where VOD peaks in the morning and decreases into the afternoon even on the dry days. This is common finding in Holtzman et al. 2021 Biogeosciences and Yao et al. 2024 Geophysical Research Letters. Later on VOD does increase due to rain but do the trees not dehydrate when transpiring through the day?

The finding that VOD had no diel cycle in dDry conditions was a surprise. This is highlighted in Fig. 5 which is only data from the dDry days (so there should be minimal effect of precipitation). In Figs. 5c and 5e both the tree sway frequency data and tree bole moisture sensors show a diel cycle that suggests transpiration effects on the tree water content. Clearly VOD (in Fig. 5d) does not have a diel cycle in these conditions. In the revised manuscript, we address this with the following text at the end of Sect. 3.2:

"There is an important implication from the dDry periods (when most of the ET is transpiration, not evaporation). Though  $\varepsilon_a$  had a dDry diel cycle with a clear early-morning maximum and late afternoon minimum, VOD had relatively small variation without any apparent pattern (Fig. 5). This suggests that, at our site, VOD changes were largely controlled by water on the canopy *surfaces*, not the the internal water content of the trees. In contrast, tree sway frequency had a diel pattern on dDry days (Fig. 5c) that was similar to that of  $\varepsilon_a$  (Fig. 5e), suggesting that tree sway frequency was more affected by internal tree water content changes than VOD. Ciruzzi and Loheide II (2019) has also shown that tree sway motion followed changes in internal tree water content.

The lack of a clear diel pattern in VOD during dDry conditions was a surprising result because previous studies (e.g., Holtzman et al., 2021; Humphrey and Frankenberg, 2023; Yao et al., 2024) have shown a VOD diel pattern that they related to the changes in internal water content of the forest/trees. As we looked closer at this, we realized that US-NR1 VOD in dDry conditions was much lower than VOD from the other sites (both the mean value and the diel range in dry conditions). For example, the US-NR1 dDry VOD diel range is 0.36 to 0.38 (Fig. 5d) whereas the VOD diel range in the study by Holtzman et al. (2021) was on the order of 0.85 to 1.1 (their Fig. 4) and that of Yao et al. (2024) had a range of 0.62 to 0.65 (their Fig. 2). Both of these studies are from deciduous forests in the eastern USA. Though we cannot definitively explain the reason for low VOD at the US-NR1 site, we offer a few possible explanations: (i) the US-NR1 forest has a lower tree density, (ii) the internal water content of the coniferous US-NR1 trees is lower and stored differently within the tree bole than broadleaf trees in the more humid locations (Hacke et al., 2015; Luo et al., 2020), or (iii) there is too much noise in the VOD measurements to properly capture the true diel cycle in internal tree water content during dDry conditions (which is when the VOD signal is weakest)."

This issue is further discussed in our reply to Referee #2 (item #2).

7. 336-337: Please describe the B0, G1, A1, and F2 cases here or in the methods sections for reference to the reader.

The meaning of B0, G1, A1, and F2 are described at the end of Sect. 2.3. In Sect. 3.4 of the revised manuscript, we have added a reference back to Sect 2.3 so this information can be easily found.

8. 419: This is does not seem like a novel finding given your reference to the common method of removing these data in the introduction. Please clarify the novelty here.

We agree that our study builds on previous suggestions (e.g., Holtzman et al., 2021; Yao et al., 2024) that time periods following precipitation affect VOD. However, in Holtzman et al. (2021), they suggested that the canopy wetness did *not* affect VOD (i.e., Sect. 3.3 in Holtzman, et al. is entitled, "Canopy interception fails to influence VOD"). And, in Section 4.4, they write,

"More research is needed to better understand how VOD sensitivity varies between water internal and external to the canopy.".

So, Holtzman et al. (2021) did not make a conclusive statement on this issue. Yao et al. (2024) showed how wet periods affected the canopy (i.e., their Fig. 2c), which they used this as evidence to avoid wet periods. In contrast, we are stating these "wet" periods are valuable for looking at canopy evaporation and specify how long it takes for the canopy to dry out (which is the novel information added by our study). We agree that these subtle points in Holtzman and Yao should be clarified in our manuscript. If there are other studies that have specifically looked at this issue that we have not included within our manuscript, please let us know about them. We have modified the text in the conclusions to better highlight this point.

9. 477: suggest to replace leaf with needles for clarity.

Good point. We changed "leaf" to "clumped needles".


One of our conclusions is that VOD at US-NR1 is not very sensitive to the internal tree water content. We agree that this could be expanded upon; our responses to Referee 1 (items #6 and #8) contain additional details and summarize the changes made in the revised manuscript.

Figure R1: A portion of the the VOD time series shown in Fig. 2b in the discussion manuscript. Here we have included both the raw/hourly and interpolated/30-min data (see legend). Statistics (number of samples N, mean, and standard deviation Std Dev) from both sets of VOD data for the period DOY 160-170 are included as a table in the upper-left hand part of the figure.

3. Page 5, Line 139: Authors mention "Linear interpolation over time was used to convert the hourly data to a 30-min time series". What is the rationale for linearly interpolating hourly data to 30-minute resolution, instead of conducting the analysis directly at the native 1-hour interval? Does this interpolation introduce any artifacts?

This is a good question. There are two primary motivations for interpolating the VOD data from hourly to 30-min time periods; (i) all the water vapor flux data and tree sway data are calculated over a 30-min period, and (ii) more clarity in the diel cycle is obtained with as high a sampling frequency (or as short an averaging period) as possible. Rather than downgrading the measurement period of the tree sway and flux data, we decided to linearly interpolate the VOD data.

To check how the linear interpolation might affect the VOD statistics, a 7-day time series of the raw/hourly and interpolated 30-min VOD data is shown in Fig. R1. This figure includes 10-day mean and standard deviation of the respective VOD data, and the differences in mean values (0.379 vs 0.378) are very small (less than 0.5%), whereas differences in the standard deviation (0.060 vs 0.058) are less than 3%.

If we zoom in a bit more and only show half of a day, then we see that the effect of linear interpolation decreases the range of the measured VOD (Fig. R2). This happens because the 30-min periods are centered on 15 and 45 minutes past the hour, while the hourly data are centered on 30-min past the hour. The decrease in the range of the measurements is why the linearly-interpolated data have a slightly smaller standard deviation than the hourly measurements.

As far as we can tell, the linear interpolation does have have any significant impact on our results or conclusions. We further test this by repeating Fig. 3d in the manuscript, but showing both the diel cycle of the hourly VOD data along with the 30-min interpolated VOD data (Fig. R3). There are small differences, but nothing that would change our interpretation of the results.

Figure R2: As in Figure R1, but only showing the first 12 hours of DOY 260.

**4. Please clarify the climate classification of the study site.**

We added additional information in the site description section which describes the US-NR1 climate. In the revised manuscript, the new text is:

"The mean annual temperature at US-NR1 is around 2°C. According to the Köppen–Geiger climate classification system (Kottek et al., 2006) the site is type Dfc which corresponds to a cold, snowy/moist continental climate with precipitation spread fairly evenly throughout the year; the long-term mean annual precipitation at the site is around 800 mm and snow typically covers the ground from mid-November until late May (Burns et al., 2015)."

5. Tu & Yang, 2022, Hua et al., 2020 discuss the overestimation of PET/ET particularly in arid and semi-arid environments while Sun et al., 2016 discusses underestimation of ET in cold areas using traditional methods. Could the authors clarify whether ET overestimation/underestimation is relevant for this ecosystem, what are the implications and uncertainties?

We considered the following papers brought up by the reviewer: Hua et al. (2020); Sun et al. (2016); Tu and Yang (2022). If any of these are incorrect, please provide the DOI, so we can make sure to find the correct paper.

These papers are primarily concerned with modeling ET. Sun et al. (2016) use modeled ET to look at effects on the air and land-surface temperature. Hua et al. (2020) uses Penman-Monteith to model ET and compare with surface met data and MODIS products. Tu and Yang (2022) focused on potential evaporation using a variety of models and compared these to ET from flux towers; however there are some open questions about net radiation assumptions within the chosen model (e.g., Szilagyi, 2022). While these are all reasonable/useful studies, we used a direct measurement of ET in our study. The studies could be relevant to the CLM4.5 portion of our study, but we examined how the model reacts to precipitation, not the metrics used in those studies. Perhaps these citations would be appropriate for a future study (i.e., that is trying to model absolute ET), but not the current one.

Figure R3: Similar to Fig. 3d in the manuscript, but showing the mean diel cycle for both the hourly and 30-min interpolated VOD data (see legend). See Fig. 3d within the manuscript for additional plot details.

6. A brief explanation of the detrending process for tree sway frequency in the main text would be beneficial, even if detailed in Raleigh et al. (2022). Further, please clarify relevance of detrending tree sway alone and not others.

The low-frequency detrending of the tree sway frequency data is described on lines 155–157 and an example is shown in supplemental Fig. S4 (in the discussion paper). The detrending uses a 10-day sliding median filter. The low-frequency detrending was not part of Raleigh et al. (2022). The tree sway data has an obvious low-frequency trend in the time series (i.e., Fig. 2c in the manuscript). As we have shown, the low-frequency detrending did not impact the results (see discussion on lines 245–247 in the discussion paper and Fig. 4). None of the other variables showed an obvious trend with time so there was not a need to detrend any other variables (in fact, it was not really needed for tree sway frequency either).

7. The study relies on 17 wDry days for the VOD and tree sway, while the ET results for wDry days (in Figure 6a,6b) are based on a significantly larger sample size of 176 days from a longer period (2004-2022 vs. 2022-2023 for VOD/sway). To strengthen the paper to explicitly discuss how this disparity might influence the comparison, perhaps a composite of ET from only the 17 wDry days used for VOD/sway could be presented in the supplementary information for a more direct comparison.

It is correct that in Fig. 6a,b, the period of 2004 to 2022 is used for the ET diel cycle. This is discussed as a limitation of our study (lines 388–391) in the discussion manuscript. We note that a composite of the above-canopy ET for the 17 wDry days is shown in Fig. 3b. There are several reasons that a longer period is used for ET in Fig. 6: (i) the subcanopy flux system was not working correctly in Fall 2022 into summer of 2023, and (ii) a smooth ET diel cycle greatly benefits from having many samples/years of data. This can be seen by comparing the above-canopy ET shown in the wDry part of Fig. 3b with that in Fig. 6a. With only 17 ET periods the line in Fig. 3b is a bit jagged whereas the one in Fig. 6a is much smoother. To extract a smooth diel cycle is an even larger challenge in the subcanopy, where we use an open-path IRGA.

For these reasons (primarily item (i) above), we cannot do this suggestion by Referee 2.

**Under the category "Minor comments":**

8. Page 1, Line 1: Is Interception only due to warm-season precipitation?

Is "Page 1, Line 1" referring to the title which has "canopy-intercepted rainfall" in it? This comment is a bit unclear, but we think you are likely referring to the fact that fog/dew are also a source of intercepted water? If this understanding is correct, this is a good point and we have added text to our list of limitations in our study as,

"Nor did we consider possible effects of fog/dew as a source of intercepted water."

We mentioned that we think fog/dew is generally a small source of intercepted water at our site (see lines 68-69 in our discussion manuscript).

If we mis-understood this comment, clarification would be appreciated!

9. Page 1, Line 9: Can the authors elaborate/clarify on this "changes in internal tree-water content than VOD."?

This point is discussed at the end of Sect. 3.2 (lines 268–275) and shown in Fig. 5 within the discussion manuscript. This comment is also closely related to comment #2 above. As we replied in comment #2, the low-level of VOD during dry days is something we need to point out more clearly within the manuscript (see comment #2 for additional details).

10. First paragraph of introduction section is detailed/explained well but not sufficiently referenced.

All the references listed in lines 30–31 are the sources of information for the descriptions prior to that. We agree with Referee 2 that the references are too far removed from each of the descriptions. This accidentally happened as the manuscript was being modified/edited. In the revised manuscript, we have put the references in locations closer to the phenomena being described.

11. Page 2, Line 22: Authors mention "Evapotranspiration ET is the sum of transpiration with soil and canopy evaporation" this is confusing; but ET = soil evaporation + canopy evaporation + transpiration?

These were intended to be equivalent statements (are they not stating the same thing?). In the revised manuscript, we have re-written this text; see the new text at the end of the first paragraph in the Introduction.

If this was not the intention of your comment, please clarify.

12. Throughout the manuscript, there are instances where terms like 'evapotranspiration' are written in full repeatedly after being defined by their abbreviation (e.g., ET). I recommend using the respective abbreviations consistently after the first mention to improve readability and maintain consistency.

We found a few places where this occurred. We have removed those instances in the revised manuscript; however, we have kept "evapotranspiration" written out fully within the abstract, figure captions, and the conclusions.

13. Page 10, Line 210: Authors mention "To ease comparison with other studies" and no references were cited here. I recommend including relevant studies to support the statement.

Good point. We have revised this sentence to be:

To ease comparison with other studies (e.g., Klaassen et al., 1998; Humphrey and Frankenberg, 2023), we have expressed ET and the rate of precipitation in units of mm of  $H_2O$  per hour (mm  $h^{-1}$ ).

14. Table 1 indicates VOD measurements began in June 2022, but the study period warm season is defined starting in Sept 2022. Please clarify on this.

Thanks for pointing this out. The GNSS measurements started in June 2022, but we initially had both systems side-by-side at the top of the tower (this is described on line 140–141 in the discussion manuscript). However, it is true that VOD can only be measured with both an above-canopy and subcanopy system; therefore, we have revised Table 1 to show the VOD dates as "Aug 2022" rather than June.

15. Please mention the temporal sampling interval for each observation type and ensure consistency throughout the manuscript.

On line 210 we wrote, "Unless otherwise noted, ET and all other statistics are calculated over 30-min periods.". We have also added a new column to Table 1 which lists the averaging periods used for each variable.


sean@ucar.edu

Date: August 15, 2025

The comments by Referee 3 are greatly appreciated. We have listed the comments by Referee 3 below in italics, followed by our responses. We added numbers to each of the specific comments so it is easier to reference the comments by Referee 3.

This paper compiles VOD, tree sway frequency, and flux tower data for a subalpine forest. Additional data include output from a land-surface model, and field/sensor data related to air temperature, bole temperature, bole apparent dielectric permittivity, wetness, and precipitation. Together, these data are used to suggest that VOD and tree sway frequency capture signals related to canopy evaporation. These measurements offer unique insight into canopy storage dynamics from completely independent observations. I don't have any major suggestions/comments for the authors. The following suggestions are meant to help the reader understand additional context for the tree sway measurements, interpretation of results, and to facilitate reproducibility/follow up studies.

This is an accurate summary of our manuscript and we appreciate the effort made by Referee 3 in the interpretation of our data/results and to facilitate follow-up studies.

**Under the category "Main Comments":**

I agree with Referee #2 in that parts of the tree sway methodology can be expanded, even briefly. Suggested expansions:

We reply below to each specific comment.

1. A) Line 145-146. If possible, include a brief description of the accelerometer methods, even though they are included in Raleigh 2022. Can go in the appendix, along with the other measurement expanded details if authors see fit.

This is a good idea and we decided to put it in the main text rather than the appendix (which doesn't have any information about the tree sway frequency measurements). We used what is described in Raleigh et al. (2022) and the revised text in Sect. 2.2.2 is:

"Processing of the raw tree sway accelerations [m s-2] into frequency [Hz] used the method described by Raleigh et al. (2022), except that a 30-minute analysis window was used. A brief summary of the data-processing steps is as follows: (1) spectral analysis of the 12-Hz acceleration data to determine a primary frequency of the tree-swaying motion  $f_{sway}$  for each 30-min period, (2) calculate a sliding 72-hr mean-filtered  $f_{sway}$  time series, (3) remove any 30-min  $f_{sway}$  outliers relative to the 72-hr mean-filtered data or values with low spectral power (i.e., due to low wind speeds), and (4) gap-fill any missing or removed 30-min time periods in the  $f_{sway}$  time series using splines. A detailed description of these steps can be found in Sect. 3.2 of Raleigh et al. (2022)."

2. B) Line 147. Why was a 30-minute window chosen? Were other windows tried but this was the best one for the analysis?

The short answer is that the eddy-covariance fluxes are calculated over 30-min periods and the US-NR1 site data are typically averaged to 30-min periods for analysis. Depending on the goals of the study, there are situation when examining the high-frequency data is more appropriate (e.g., for spectral analysis). For the US-NR1 site (and many flux sites), 30 minutes is the standard data-sharing and analysis time period. A 30-min period also provides better temporal resolution (compared to hourly periods) when examining the diel cycle. We discuss this in more detail in item #3 of our reply to Referee #2. We did not do a systematic examination of time-window-averaging length for the tree sway data analysis.

3. C) Line 148. What interpolation and smoothing function(s) were used?

A MATLAB spline fit of the tree sway freq data was used to gap-fill the low-wind periods. This comes from the MATLAB "Curve Fitting" toolbox includes a smoothing parameter which varies from 0 (very smooth) to 1 (straight line interpolation between points). With experimentation, Mark set the smoothing parameter to 0.99, as that seemed to capture the short-term variations in sway frequency over different seasons. An example is shown in Fig. 4 of Raleigh et al. (2022). We mention the spline interpolation in our modified text (shown in #2 above), but will not go into any more detail within our manuscript.

4. D) Line 156-157. How was the value chosen to be 1? Was this done by offsetting the low frequency trend such that the average frequency over the time series was 1? If so, please include this.

For the detrended tree sway frequency  $f_d$  a value of 1 was chosen for two primary reasons: (i) it was not very different than the actual frequency and (ii) it added an "offset" between the detrended and raw data, so that plots (such as Fig. 4) do not have the  $f_{sway}$  and  $f_d$  data right on top of each other. As you suggested, it was done by removing the low-frequency trend and then adding back in a value of 1. The important take-away message (described on lines 158–159 in the discussion paper), is that the low-frequency detrending of the tree sway did not significantly impact how diel cycle of tree sway frequency changed with precipitation state (as shown in Fig. 4 and supplemental Fig. S5c of the discussion paper).

5. Similarly, Line 138, can there be a brief inclusion of the data processing for VOD? Even if it is a brief summary, it would be helpful. This can be in the appendix.

We included the key information about the VOD data processing which was extracted from Humphrey and Frankenberg (2023). The revised text in Sect. 2.2.1 is:

"The data processing and calculation of hourly VOD time series used the L1 band (1575 MHz) and followed the procedures in Humphrey and Frankenberg (2023). A brief summary of the VOD data-processing steps is as follows: (1) calculate the GNSS signal attenuation due to the forest by comparing the GNSS signal strength from the forest receiver (gnssB) to the one above the canopy (gnssA), (2) use the signal-strength difference to estimate the forest transmissivity  $\gamma$ , and (3) calculated an initial estimate of VOD from VOD =  $-\ln(\gamma)\cos(\theta)$  (where  $\gamma$  and  $\theta$  are the canopy transmissivity and incidence angle, respectively). Because the GNSS samples the forest at irregular temporal intervals and angles, the long-term mean as a function of azimuth and elevation angle is used to improve the precision of the hourly measurements (Humphrey and Frankenberg, 2023)."

6. Table 1. Is it possible to add the original/raw frequency of measurements for each observation?

We added additional details to Table 1 (this was also suggested by Reviewer 2). There are two different considerations—one is the raw sampling frequency and the other is the averaging period (over which mean values or fluxes are calculated). In the revised manuscript, we have included the raw sampling frequency in a new column in Table 1.

7. Figure 2. Is the accelerometer associated with Pine 3 or Pine 4? Or are the dielectric permittivity sensor on two different trees? Please clarify.

Good point. The 3-axis accelerometer is on a spruce tree that is very close to the main tower (within about 2 m of the tower) while Pine 3 and Pine 4 are located about 20-30 m southeast of the main tower. To clarify this, we added the approximate location of Pine 3 and Pine 4 in the sensor description in Sect A5.

8. Line 347-349. Is it possible to expand on this (how these empirical relationships might facilitate changes in the land-surface model)? This seems like an important consideration, however there is nuance to it. In particular for tree sway frequency, the tree sway frequency is proportional to the inverse square root of tree mass, so while the frequency is observed to be linear, this does not mean that the tree mass (interception) is changing linearly. However, there is no mention of how tree sway frequency relates to mass in a mathematical form throughout the manuscript. I suggest adding this here and/or earlier on in the manuscript so if improvements on the land-surface model can begin there's the additional context of how physically sway frequency relates to changing tree mass due to interception.

This is a good point. In Sect. 2.2.2 of the revised manuscript we added Eq. (2) which shows how the mass within the canopy relates to the tree sway frequency. We also added additional text explaining this relationship. This information was taken from Raleigh et al. (2022), but it is good to point it out here. Please see our reply to the "Major comment" by Referee #4 for more specific details.

9. This is not exceptionally important, and the authors can ignore: I was trying to determine any sort of relative magnitude differences in responses for the precipitation events presented in table 3 and figure 8 (and 9). I tried to find the lowest precipitation event (3.6 mm, solid cyan line) and the highest precipitation value (29 mm, solid red line). I think I found them in the sway frequency plot and they had the lowest and highest range/variability, which was helpful to see that with a first order approximation that the observations aligned with physical interpretations (more precipitation -> more change in tree sway frequency). I couldn't quite find them in the VOD diagram. Highlighting these two events with bolded lines might help the reader understand the relative sensitivity to single events and strengthen the connections observed between precipitation, evaporation, and VOD or sway frequency.

We like the idea of highlighting the VOD and tree sway lines in Figs. 8/9 for the lowest and highest precipitation values. We added this feature to Figs. 9/10 in the revised manuscript. We should point out that Fig. 10b in the discussion manuscript (Fig. 11b in the revised manuscript) has the information we believe you are trying to extract for VOD. This plot shows the precipitation amount for each storm vs the 4-hour mean VOD value from 4 to 8 hours after the storm stopped (note: we noticed that there is a mistake in the caption of Fig. 10 in the discussion paper where the red dots are related to Fig. 9, not Fig. 8; this has been fixed in the revised manuscript). Though based on only 17 points, Fig. 10b gives a rough idea of the relationship higher VOD values occur with higher precipitation amounts (as one would expect). Since we have many years of tree sway data, we created a similar plot for tree sway (Fig. 12 in the revised manuscript) that reveals very similar patterns, but has more data.

10. Are the authors amenable to including in the acknowledgements or in brief 'open science' section in the appendix or supplemental info where the data and code used throughout the manuscript can be found? This would help future studies reproduce these results and follow up studies that are interested in conducting similar experiments elsewhere.

Thanks for bringing this up. Most of the US-NR1 30-min flux/met data are already available via AmeriFlux (https://ameriflux.lbl.gov/sites/siteinfo/US-NR1) which is listed under the "Assets" tab in the discussion paper.

For the final paper, we are in the act of creating an ESS-DIVE archive for the raw GNSS data (a 40 Gb dataset), as well as the processed tree sway and flux data, and all other data used within our manuscript. The ESS-DIVE archive has a reserved DOI and the citation will be:

Burns, S.P., V. Humphrey, M. S. Raleigh, D. R. Bowling, E. D. Gutmann, and P.D. Blanken, 2025: *GNSS-based Vegetation Optical Depth (VOD), Tree Sway, and Evapotranspiration data from the Niwot Ridge Subalpine Forest (US-NR1) AmeriFlux site*, AmeriFlux Management Project, ESS-DIVE Dataset,

```
https://doi.org/10.15485/2574352
```

At this point, we have reserved the DOI for this ESS-DIVE archive, but it might not be active/available (our goal is to have the DOI active/available when the paper is published). This ESS-DIVE archive will have a format very similar the 2020 ESS-DIVE archive that is listed in the Assets tab of the discussion paper:

Burns, S.P., P.D. Blanken, and R.K. Monson, 2020: *Data, Photographs, Videos, and Information for the Niwot Ridge Subalpine Forest (US-NR1) AmeriFlux site.* AmeriFlux Management Project, ESS-DIVE Dataset,

```
https://doi.org/10.15485/1671825
```

In the final paper, we will also add the following links to the Assets tab which are the github websites that contain the tree sway and VOD data-processing software:

```
Raleigh M. S., Code to Process Tree Sway Frequency from Accelerometer Data, https://github.com/truewind/accelerometer_tree_sway/, last access: 14 August 2025.
```

Humphrey, V., Python Toolkit for Deriving Vegetation Optical Depth (VOD) from Pairs of GNSS Receivers, https://github.com/vincenthumphrey/gnssvod, last access: 14 August 2025.


sean@ucar.edu

Date: August 15, 2025

The comments by Referee 4 are greatly appreciated. We have listed the comments by Referee 4 below in italics, followed by our responses.

This research investigates promising and novel techniques to predict rainfall interception and, thus indirectly, canopy evaporation. The authors use L-band microwave active microwave attenuation data (vegetation optical depth) from a GNSS doublet and tree sway data measured by a an accelerometer placed on the trunk of a tree canopy. The study is set in an subalpine, high elevation needleleaf forest in in Colorado/USA. The authors demonstrate the ability of both proxies to correlate with onset and drydown of precipitation events, evapotranspiration and modeled interception storage from different land surface model (CLM4.5) parameterizations. The data sets and analysis presented by the authors allow for the conclusion that these techniques are promising tools to measure interception storage and that they hold potential to supplement/validate land surface models that are known to have high uncertainties in interception fluxes the their parameterization of the canopy, and uncertainties in EC water flux measurements during rain events. This study is of great quality. However, the authors should address the comments below before publication.

This is an accurate summary of our study and we appreciate the positive comments that our study is of "great quality". The specific comments by Referee 4 are replied to below.

**Under the category "Major comment":**

Please elaborate on the robustness of tree sway motion being able to represent interception storage without the need to account for wind speed as a possible confounding factor. In this context, it would be valuable to find sway motion data as a function of wind speed—e.g. in fig. 10 and at least in one of the plot over time—to clarify on this relationship and include this missing piece of information.

Based on mechanical theory, the natural sway frequency  $f_{sway}$  of a conifer tree acts like a damped harmonic oscillator; therefore, it does not depend on wind speed. This is highlighted in Sect.3 of Raleigh et al. (2022) as well as many other studies (e.g., Moore and Maguire, 2004; Van Emmerik et al., 2017; Jackson et al., 2021), who show that  $f_{sway}$  is described by the cantilever model:

$$f_{sway} \propto \frac{1}{2\pi} \left(\frac{K}{m}\right)^{0.5},$$
 (1)

where K is the flexural rigidity of the tree and m is the mass of the tree, including the branches and leaves/needles. As precipitation accumulates on the tree leaves and branches, m changes which alters  $f_{sway}$ . Therefore,  $f_{sway}$  is primarily a structural property of the tree that depends on mass (tree biomass + water in/on the tree), elasticity (which varies with tree temperature, thermal state, and water content), and tree geometry (tree height and DBH). There are other models which are appropriate for other tree types (e.g., simple pendulum for broadleaf trees), but across model types (cantilever and pendulum) tree frequency is not predicted by wind speed.

Figure R1: Similar to Fig. 10 in the discussion manuscript, but using detrended tree sway frequency  $f_d$  rather than VOD. In (a), the relationship between above-canopy mean horizontal wind speed WS and total precipitation amount from the wet day preceding the wDry day is shown. We have multiplied  $f_d$  by -1 to create patterns consistent with VOD (as in Fig. 10). The solid black points are the 30-min mean values from the warm season periods for years 2016 to 2023, the red points are the mean values calculated between 4 and 8 hours after precipitation ended. See Fig. 10 for additional details.

Contribution of individual trees to the overall GNSS-VOD signal

Figure R2: Schematic of how the VOD footprint varies with tree height and distance from the subcanopy GNSS antenna. The trees with darker shading indicate a larger contribution to the VOD footprint.

As alluded to by Referee 4, this technique requires some minimal level of wind speed to generate tree sway. This was mentioned on lines 400-401 in the discussion paper, as one of the limitations of the method. The lack of a dependence of tree sway on wind speed is discussed in detail in the supplemental material of Raleigh et al. (2022) (see their Fig.S5) as well as the other papers listed above. The mean wind speed impacts the amplitude of oscillations, rather than the frequency (which is what is used in our study). As explained in Raleigh et al. (2022), periods with absolute calm-wind conditions were gap-filled by interpolation. With this said, we agree with Referee 4 that it is worth creating something like Fig. 10, but using tree sway frequency rather than VOD (especially since we have 6 years of tree sway freq data). This new figure is shown in Fig. R1, and many of the relationships between tree sway frequency and other variables (VPD, wind speed, turbulence) are similar to those with VOD. In Fig. R1d, the relationship between tree sway frequency and wind speed is shown and it can be seen that the periods with lower tree sway freq for WS less than 5 m/s correspond to wetter periods (ie, the red dots). The other take-away from Fig. R1d, is that higher winds (and turbulence levels) are less conducive to changes in tree sway frequency, this is either because the rainwater is mechanically blown/shaken off the tree or it does not rain as often in windy conditions (as shown in Fig. R1a). Fig. R1 has been included as Fig. 12 in the revised manuscript with additional discussion about it.

**Under the category "Minor comments":**

- 1. Fig 1
- 1.1 Please report the the inner circle radius (r=20m) as the authors have done for the outer circle
- 1.2 Please point the reader of fig. 1's caption to what the different footprint circles represent to better understand the results, i.e. what is the main take-away from the inner circle radius (apart from GNSS paucity visualization).

Because comments 1.1 and 1.2 are related, we answer them together. The meaning of the inner footprint was not clear in the discussion manuscript. There are several points and clarifications that we have added to the footprint discussion in Appendix B.

First, we clarified that no clear-cut boundary on the VOD footprint location exists. The exact contribution of individual trees to the VOD measurement will depend on the height of the tree and the proximity of the tree to the GNSS antenna. The inner footprint was an attempt to show that trees closer to the subcanopy GNSS antenna contribute more to the VOD flux. In Fig. R2, we have attempted to show this schematically from a side-view perspective of the forest. The tree shading in Fig. R2 shows how the trees closer to the subcanopy GNSS antenna contribute more to the VOD measurement. We have included Fig. R2 as Fig. S4 in the supplemental material of the revised manuscript and improved the text in Appendix B.

Figure R3: Time Series of Vegetation Optical Depth (VOD) calculated using an elevation angle of 10 and 30 deg (see legend).

1.3 Appendix B, line 527: Please clarify the role of r=20 in this study. Did the authors clip the radius so the "several tall trees" are not included in the footprint?

The inner and outer VOD footprints shown in Fig. 1 are purely conceptual and no trees were removed from the data-processing based on these footprints. A better representation of the areas excluded in the data processing are shown in the skyplot (Fig. S3) where regions were excluded based on the lightning dissipator locations (not based on the trees location or the footprint radius).

- 2. Appendix B & Fig. S3 (GNSS sky view):
- 2.1 Please show the elevation angle and cutoff to clarify which parts of the canopy will effectively be used for VOD calculation, especially elevation= $10^{\circ}$

The schematic in Fig. R2 shows an elevation angle of 10 degrees and which parts of the forest will be used. As explained above, we included this schematic and additional discussion about it in the revised manuscript.

2.2 Please clarify the rationale behind clipping out another area in NE, close to the northern GNSS gap

There are two lightning dissipators at the top of the tower, so two locations in the skyview are excluded (the lightning dissipators are described in Sect. 2.2.1, lines 130–131 of the discussion manuscript). To clarify what the lightning dissipators look like, photos of them are in Fig. R4.

2.3. You use a very low cutoff elevation angle of  $10^{\circ}$ . Looking on fig. S3 – assuming the outer two circles being roughly within  $\theta$  in (10, 30] – only very low VOD can be found that do not display any pattern expected from forest attenuation and possibly fail to represent true forest VOD. Under this light, please explain why the authors used a cutoff= $10^{\circ}$ .

In principle, the lower the elevation angle, the more homogeneous the VOD should be because it is sampling across a larger number of trees in all directions. Thus, with a high enough canopy, the values at lower elevation angles should converge towards the average VOD at the site, which seems to be the case here (goes to around 0.4). Because values from lower elevation angles mainly sample the upper half of the trees, it could bias the VOD estimate somewhat, especially if the canopy density is very heterogeneous in the vertical direction. On the other hand, including these low elevations (10–30 deg.) increases the representativity (larger footprint) and the sample size (number of raw measurements), which reduces noise in the VOD time series. It is of course a trade-off, which this study does not aim to explore at this stage.

2.4. Since the Lambert-Beer angle correction assumes a homogeneous canopy and ignores multipath scattering, any losses observed may be due to scattering caused by multiple layers of vegetation, causing Lambert-Beer to break at low angles. Hence, the referee suggests using a higher cutoff elevation angle (~30 deg.) or would value a discussion why low VOD at lower angles will not affect the overall results. Consider page 12 in Camps et al. (2020) about this question: "Note, however, that only at high elevation angles (elevation angle > 67.5°) is the single scattering albedo correlated with the NDVI, and at lower elevation angles, the presence of multiple scattering makes the tau-omega model [all zeroth order assumption, incl. lambert-beer, ~the referee] more likely to be invalid."

Thank you for bringing the work by Camps et al. (2020) to our attention. See answer to comment 2.3 above for related details; For our study, the fact that the low elevation values converge to the mean VOD tends to suggest that these values are not so systematically biased that they would need to be removed. We would have acted differently if low elevation values had a different behavior. We agree with Referee 4 that the effect of using a different elevation angle should be further explored. We recalculated VOD with an elevation angle of 30 degrees found that it did not have a large effect on the results (Fig. R3).

3. 135: Which GNSS frequency is used, please indicate the frequency(ies) in section 2.2.1 since GNSS VOD offers a range of bands to choose from.

Good point. The specific GNSS frequency used was 1575 MHz which we added to the text in Sect. 2.2.1 of the revised manuscript.

4. The authors detrend sway motion to alleviate effects of temperature and vegetation water content on short-term changes. However, VOD is also affected by long-term changes in biomass, and vegetation water content. Why did the authors not consider detrending VOD, especially since a trend is visible in fig. 2? This is worth noting in 2.2.1.

Any low-frequency trend in VOD appears to be much smaller than that of tree sway frequency (i.e., compare the VOD time series in Fig. 2b to that of tree sway frequency in Fig. 2c). Since removing the low-frequency trend in tree sway frequency did not affect the results, we have assumed that removing any low-frequency trend in VOD will also have a minimal effect on the results. We have added text about not removing any low-frequency trend in VOD to Sect. 3.1, where the time series are discussed.

5. 433/4: The size of the EC footprint has not been explicitly mentioned in the text. Also, which footprint size of VOD as your referring to in this statement? To make a statement about the footprint size (a very relevant discussion) the referee suggests to state that although the footprint sizes between all technique partly or greatly differed, the good correlations could be found etc.

We thank Referee 4 for noticing this shortcoming. Shortly after making the submission, the lead author realized that though we make reference the ET flux footprint, we did not include any specific details about it. We appreciate that this oversight was noticed by Referee 4, and have included the following information in the revised manuscript. First, we cite Chu et al. (2021) who show a footprint climatology and suggest that the US-NR1 footprint has a size of around 500 m2. A more explicit view of the US-NR1 flux footprint climatology is shown below for five different atmospheric stability conditions for winds from the west (Fig. R5) and for winds from the east (Fig. R6). The data are separated into east and west wind directions because winds at the site are typically either upslope (from east) or downslope (from west). For comparison purposes, we include the larger VOD footprint from Fig. 1 on the same plot. The tower footprints have been calculated using the simple footprint model of (Kljun et al., 2015). In the revised manuscript, Figs. R5 and R6 have been added to the supplemental material and a description of the ET flux footprint is in Appendix B.

Figure R4: Photos of the lightning dissipators at the top of the US-NR1 flux tower. The right-side photo shows the gnssA antenna on the southwest corner (left/front corner in photo) of the tower and the lightning dissipator on the southeast corner (right corner in photo) of the tower. There is 2nd lightning dissipator on the northwest corner that is barely visible in the right photo, but is the one shown in the left-side photo. The dissipators extend about 2 m above the top of the tower.

Figure R5: Climatology of the footprint region from which 80% of the 21.5 m turbulent scalar flux originates are shown for winds from the west for different stability classes (SU, strongly unstable; WU, weakly unstable; NN, near-neutral; WS, weakly stable; SS, strongly stable). These are US-NR1 data from July for years 1999–2023 where the number of 30-min samples within each stability category are shown by N in the legend. Footprints are calculated based on Kljun et al. (2015) and shown as distance [meters] from the main US-NR1 flux tower. The larger VOD footprint from Fig. 1 is shown as a black circle.

Figure R6: As in Fig. R5, but for winds from the east.

---

## Author Response (AR2)

**Mesoscale & Microscale Meteorology Laboratory**

P.O. Box 3000, Boulder, CO 80307-3000 USA • P: (303) 497-8934 • F: (303) 497-8171

Sean P. Burns • sean@ucar.edu

September 3, 2025

Dr. Andrew Feldman Biogeosciencs Associate Editor Copernicus Publications editorial@copernicus.org

Dear Dr. Feldman,

Thank you for accepting our manuscript (egusphere-2025-1755) entitled, "Using GNSS-based vegetation optical depth, tree sway motion, and eddy-covariance to examine evaporation of canopy-intercepted rainfall in a subalpine forest" by myself, Vincent Humphrey, Ethan Gutmann, Mark Raleigh, Dave Bowling, and Peter Blanken for publication in the EGU journal *Biogeosciences*.

As suggested, I checked the figures for color-blindness and they seemed reasonable. I did not think it necessary to reload the point-by-point replies to the reviewers. I made a few minor corrections to the text and checked the references so they agree with the required formatting (journal titles, etc.). Therefore, I have uploaded new PDF versions of the text and supplement as well as the latex file (with the text, references, tables, and figure captions) and individual PDF figures as a ZIP archive.

Finally, the ESS-DIVE archive cited within the paper is currently under review by ESS-DIVE:

Burns S. P., Humphrey V., Raleigh M. S., Bowling D. R., Gutmann E. D., and Blanken P. D. (2025): GNSS-based Vegetation Optical Depth, Tree Sway, and Evapotranspiration data from the Niwot Ridge Subalpine Forest (US-NR1) AmeriFlux site. ESS-DIVE [Dataset]. http://dx.doi.org/10.15485/2574352

The process for final approval could take about a week, after which time the DOI listed above should be active and available. If there are any questions or problems with our re-submission please don't hesitate to contact me.

Sincerely,

Sea Buns

Sean P. Burns